# Integrating endogenous TurboID and data-independent acquisition mass spectrometry for in vivo proximity labeling

David S Fay (ID)[1,3]✉, Boopathi Balasubramaniam (ID)[1,3], Sean M Harrington[1,2] & Philip T Edeen[1]

## Abstract

Proximity labeling has emerged as a powerful approach for identifying protein–protein interaction networks within living systems, particularly those involving weak or transient associations. Here, we present a comprehensive revised proximity labeling workflow, integrating TurboID labeling of endogenously expressed fusion proteins and data-independent acquisition (DIA) mass spectrometry (MS). We benchmark this pipeline with a study of five conserved *Caenorhabditis elegans* proteins—NEKL-2, NEKL-3, MLT-2, MLT-3, and MLT-4— that form two NEKL–MLT kinase–scaffold subcomplexes involved in membrane trafficking and actin regulation. Profiling of NEKL–MLT interactomes across 23 experiments validated our approach through the identification of known NEKL–MLT binding partners and conserved *nekl-mlt* genetic interactors, including the discovery of several novel functional interactors. Importantly, inclusion of methodological variations, stringent controls, and filtering strategies enhanced sensitivity and reproducibility, defining a set of intuitive quantitative metrics for routine assessment of experimental quality. We show that DIA-based interactome workflows produce physiologically relevant findings, even in the presence of experimental noise and variability across biological replicates. Our study underscores the utility of DIA mass spectrometry in proximity labeling applications and highlights the value of incorporating internal controls, quantitative metrics, and biological validation to enhance confidence in candidate interactors.

**Keywords** Proximity Labeling; NIMA-related Kinases; *C. elegans*; TurboID
**Subject Categories** Cell Adhesion, Polarity & Cytoskeleton; Methods & Resources; Proteomics

## Introduction

The detection of protein–protein interactions, either through focused studies or high-throughput screens, has been an area of intensive investigation for nearly four decades. Open-ended approaches such as yeast two-hybrid screens (Bruckner et al, 2009; Fields and Song, 1989; Maple and Moller, 2007), affinity purification–mass spectrometry (MS) (Dunham et al, 2012; McLafferty, 2011; Rozanova et al, 2021), and phage display (Palzkill et al, 1998; Smith, 2019; Sundell and Ivarsson, 2014) have collectively contributed to the discovery of protein networks in a wide range of systems. Although each of these approaches has its unique strengths, a common drawback is that none are designed to detect weak or transient interactions within living intact organisms. By addressing this limitation, proximity labeling has emerged as a powerful alternative for identifying protein interactions and subcellular protein networks (Beganton et al, 2019; Bosch et al, 2021; Guo et al, 2023; Lobingier et al, 2017; Roux et al, 2012; Trinkle-Mulcahy, 2019).

In proximity labeling, a protein of interest (the bait) is linked to an enzyme that catalyzes the covalent tagging of neighboring proteins with a distinct chemical label (Beganton et al, 2019; Guo et al, 2023; Qin et al, 2021; Trinkle-Mulcahy, 2019). The labeling radius of proximity enzymes varies depending on the enzyme, substrate, and cellular environment, but is generally estimated at 10–30 nm, comparable to the combined length of several midsized globular proteins (Cho et al, 2020b; May et al, 2020; Roux et al, 2012; Sears et al, 2019; Wu et al, 2024; Yang et al, 2024). After cell lysis, the chemical tag is used to trap and purify bait-labeled proteins, which are subsequently identified using liquid chromatography–tandem MS (LC-MS/MS). One of the widely used approaches, termed BioID, uses biotin (vitamin B7), a small and ubiquitous molecule, as the covalent tag (Bosch et al, 2021; Guo et al, 2023; Roux et al, 2012). Biotin, when activated by a biotin ligase in an ATP-dependent fashion, forms a covalent bond with the ε-amino group of surface-exposed lysine residues (Cronan, 2024). Due to the strong affinity between biotin and streptavidin ($K_d$, $\sim 10^{-14}$ M) (Chaiet and Wolf, 1964; Weber et al, 1989), biotin-labeled proteins can be isolated with high specificity using

[1]Department of Molecular Biology, College of Agriculture, Life Sciences and Natural Resources, University of Wyoming, 1000 E. University Ave., Laramie, WY, USA. [2]INBRE Data Science Core, University of Wyoming, Laramie, WY 82071, USA. [3]These authors contributed equally as first authors: David S Fay, Boopathi Balasubramaniam.
✉E-mail: davidfay@uwyo.edu

streptavidin-coated beads, which is followed by tryptic digestion and LC-MS/MS.

Several recent innovations have expanded the utility, flexibility, and precision of proximity labeling, especially in applications involving metazoan organisms. These advances include the development of highly active and promiscuous biotin ligases, such as TurboID, which function efficiently at temperatures compatible with the growth of model organisms such as *Caenorhabditis elegans*, *Drosophila*, and zebrafish (Branon et al, 2018; Kim et al, 2016; Kubitz et al, 2022). Furthermore, the advent of CRISPR-mediated genome editing has enabled the expression of bait–TurboID fusion proteins at endogenous levels in whole organisms (Branon et al, 2018; Guo et al, 2023; Ma and Liu, 2015). Additional enhancements include the ability to tether freely diffusible TurboID to fluorescently tagged proteins using nanobodies that bind with high affinity to the fluorescent tag (Holzer et al, 2022; Kim et al, 2023; Xiong et al, 2021) and the development of split-TurboID and light-activatable biotin ligases, both of which may provide greater temporal and spatial control over ligase activity (Cho et al, 2020a; Cho et al, 2020b; Qu et al, 2025; Shafraz et al, 2023).

Improvements in MS and analytical workflows, especially recent advances in data-independent acquisition (DIA) techniques, also have the potential to enhance the outcomes of proximity labeling (Androniciuc et al, 2024; Frohlich et al, 2022; Gömöryová et al, 2025; Hay et al, 2023; Li et al, 2021b; Pino and Schilling, 2021; Salovska et al, 2021; Zhong et al, 2024). DIA and traditional "shotgun" mass spectrometry, also known as data-dependent acquisition (DDA), differ in key experimental steps including the selection of precursor ions for fragmentation, and also in the downstream computational approaches used to identify and quantify proteins (Fernandez-Costa et al, 2020; Gillet et al, 2012; Lou and Shui, 2024; Zhang et al, 2023). As compared with DDA, DIA provides several advantages. These include greater reproducibility across experiments, increased depth of proteomic coverage, improved detection of low-abundance proteins, and an expanded dynamic range for quantitative analysis (Frohlich et al, 2022; Li et al, 2021b; Lu et al, 2021; Muller et al, 2019; Yu et al, 2023; Zhong et al, 2024). Nevertheless, among the publicly available studies using proximity labeling to identify protein interactomes in *C. elegans*, only one—conducted by our group—used DIA, and this was limited to a single bait protein (Artan et al, 2021; Artan et al, 2022; Binti et al, 2024b; Holzer et al, 2022; Kutzner et al, 2024; Nikonorova et al, 2025; Price et al, 2021; Sanchez and Feldman, 2021; Shi et al, 2025). Likewise, among the more than 20 proximity labeling studies conducted in *Drosophila*, only a few have used DIA methods (Androniciuc et al, 2024; Gömöryová et al, 2025).

In this study, we present our approaches for identifying the proximate interactors of five conserved *C. elegans* proteins: NEKL-2 (human NEK8/9), NEKL-3 (human NEK6/7), MLT-2 (human ANKS6), MLT-3 (human ANKS3), and MLT-4 (human INVS). NEKL-2 and NEKL-3 are members of the NIMA family of Ser/Thr protein kinases, whereas the MLTs are ankyrin repeat proteins that serve as signaling scaffolds and are essential for the proper subcellular localization of the NEKLs (Czarnecki et al, 2015; Fry et al, 2017; Hoff et al, 2013; Lazetic and Fay, 2017a; Moniz et al, 2011; Parker et al, 2007; Ramachandran et al, 2015; Yochem et al, 2015). Our previous studies have shown that the *C. elegans* NEKL–MLT proteins assemble into two primary subcomplexes—NEKL-3–MLT-3 and NEKL-2–MLT-2–MLT-4—that exhibit

largely distinct subcellular localization patterns, with limited spatial overlap (Fig. 1A). (Joseph et al, 2023; Lazetic and Fay, 2017a). Physical association between the two subcomplexes is supported by experimental evidence from mammalian systems demonstrating that ANKS3 can bind ANKS6 (Leettola et al, 2014; Yakulov et al, 2015) and that NEK9 binds to and phosphorylates NEK6 and NEK7 (Belham et al, 2003; de Souza et al, 2014; Richards et al, 2009; Vaz Meirelles et al, 2010). Moreover, AlphaFold modeling of the *C. elegans* NEKL–MLT proteins supports both protein–protein interactions within each subcomplex as well as more limited interactions between the two subcomplexes (Figs. 1B and EV1A–E).

In *C. elegans*, the NEKL–MLTs function in the epidermis to collectively regulate multiple aspects of membrane trafficking and actin cytoskeletal regulation, functions that appear to be conserved in mammals (Joseph et al, 2023; Joseph et al, 2020; Lazetic and Fay, 2017a; Lazetic et al, 2018; Liberali et al, 2014; Modica et al, 2025; Pelkmans et al, 2005; Yochem et al, 2015). Reduced activity of individual NEKL and MLT proteins leads to larval arrest caused by defects in molting, a process that requires apical extracellular matrix remodeling and is dependent on both membrane trafficking and cytoskeletal reorganization (Lazetic and Fay, 2017b; Sundaram and Pujol, 2024). In addition, conserved endocytic regulators have been identified as genetic suppressors of the membrane-trafficking and molting defects observed in *nekl* mutants (Joseph et al, 2018; Joseph et al, 2020; Lazetic et al, 2018; Milne et al, 2025; Reimann et al, 2025). Nevertheless, the interaction partners and substrates through which the NEKL–MLTs mediate these functions remain largely unknown.

Here, we describe our experimental strategies for carrying out proximity labeling on five members of the *C. elegans* NEKL–MLT complex using a DIA LC-MS/MS pipeline. We present intuitive methods for analyzing, evaluating, and filtering datasets, and compare outcomes between independent experiments, including methodological variations. Importantly, our study reports the first comprehensive identification of proteins that may directly or indirectly interact with the NEKL–MLTs, including potential transient interactors and substrates, and performs functional validation studies on several of the identified targets.

## Results and discussion

### Overview of the study

A central goal of this paper is to describe experimental and analytical approaches for carrying out proximity labeling in *C. elegans*, which in our case focused on five conserved NEKL–MLT family members (NEKL-2, NEKL-3, MLT-2, MLT-3, and MLT-4; Fig. 1A). In doing so we hope to provide readers with practical guidance, insights into experimental design, and benchmarks for evaluating the success of proximity labeling experiments in systems such as *C. elegans*. Detailed follow-up studies on the identified candidates, including their potential physical and functional interactions with the NEKL–MLTs, will be reported elsewhere.

For this study, we used eight CRISPR-generated strains that express endogenously tagged fusions of NEKL–MLT (bait) proteins with TurboID. These included C-terminal fusions with each of the five proteins, as well as N-terminal fusions with three of them (NEKL-2, NEKL-3, and MLT-3; Appendix Supplementary

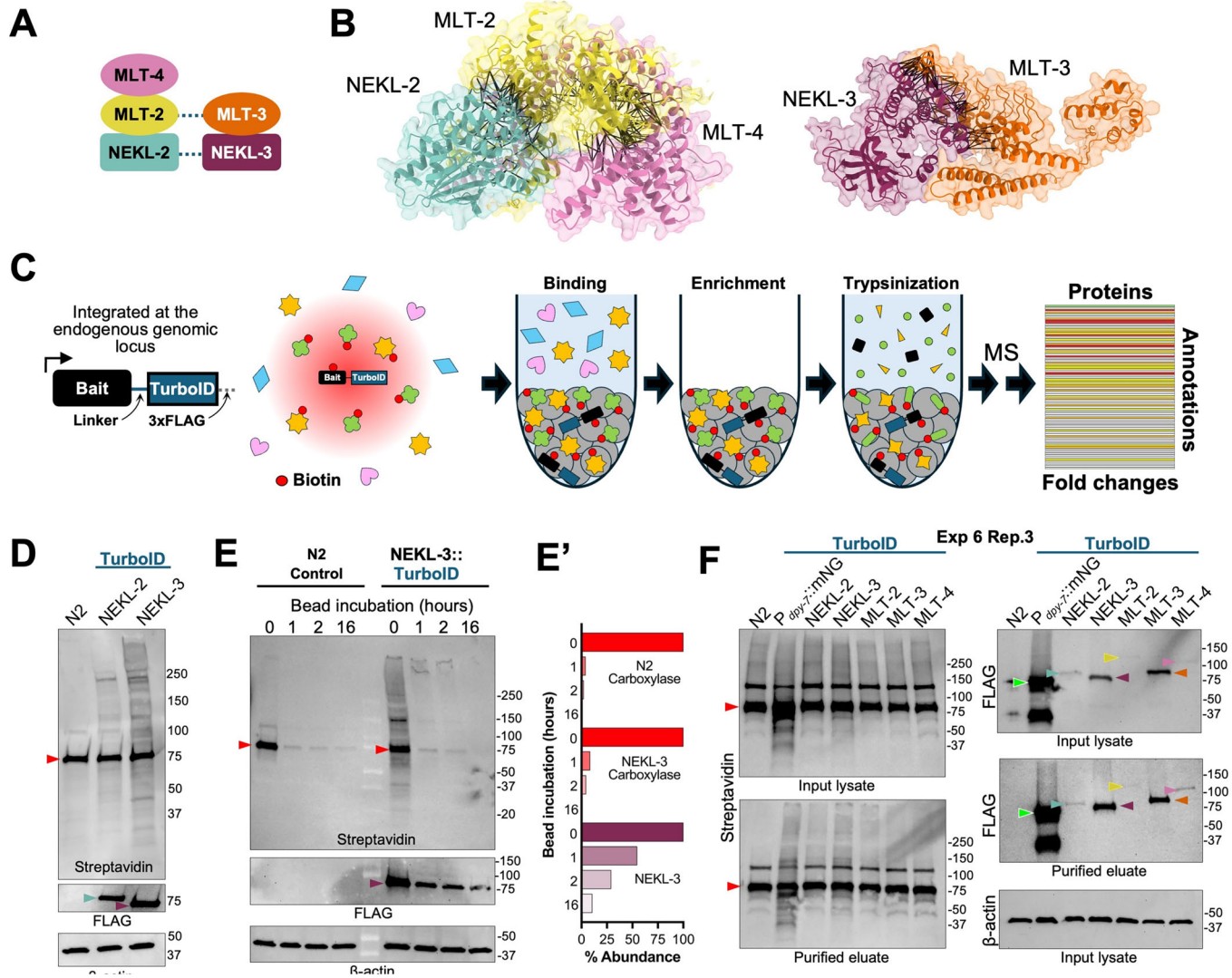

**Figure 1. Overview of the methodology.**

(A) Schematic of the two NEKL–MLT protein subcomplexes (i.e., MLT-4–MLT-2–NEKL-2 and MLT-3–NEKL-3); dashed lines indicate putative contacts between the subcomplexes. (B) AlphaFold3 modeling and ChimeraX rendering of predicted MLT-4–MLT-2–NEKL-2 and MLT-3–NEKL-3 subcomplexes. Proteins are represented as backbone ribbon diagrams with partially transparent surfaces. Black lines between proteins indicate high-confidence Predicted Aligned Error (PAE) pseudobonds (PAE ≤ 6 Å; interatomic-distance ≤ 5 Å). For clarity, highly unstructured portions of the peptides (B-factor ≤30) were hidden. For additional information, see Fig. EV1. (C) Diagram of the proximity labeling experimental steps, including a schematic of a C-terminally tagged bait protein containing an unstructured linker, TurboID, and 3xFLAG-tag. Proteins within the red shaded area, indicated by colored shapes, are biotinylated by the bait–TurboID fusion in vivo. After lysis, biotinylated proteins are captured and enriched on streptavidin beads via washing, followed by trypsin digestion. Released peptide fragments are identified by LC-MS/MS, followed by filtering and annotation of the datasets. (D–F) Western blot analyses of input lysates and purified eluates (after enrichment on beads) from the indicated strains to detect biotinylated proteins (streptavidin), bait–TurboID fusion proteins (FLAG), and β-actin. Arrowheads in western blots are color-coded to correspond to the NEKL–MLT bait proteins in (A). Red triangles correspond to the major biotinylated carboxylase species; bright green triangles with white outline correspond to the P*dpy-7*::mNG::TurboID control. (D) Lysates of N2, NEKL-2::TurboID, and NEKL-3::TurboID showing faint biotinylated protein species in the bait–TurboID lanes. (E) Lysates of N2 and NEKL-3::TurboID showing the strong depletion of biotinylated proteins within one hour of incubation with streptavidin beads. (E′) Shows quantification of the indicated bands from (E). Also note the progressive depletion of NEKL-3::TurboID (FLAG) from the lysate. (F) Representative TurboID experimental blot (Experiment 6, Replicate 3) showing input lysates and purified eluates. Note differences in the levels of the bait–TurboID proteins, as well as the strongly expressed P*dpy-7*::mNG::TurboID control. The full-length mNG::TurboID protein has an expected molecular weight of ~68 kDa; the lower band is an apparent breakdown product. Full blots are available in Appendix Fig. S1. Source data are available online for this figure.

Methods S1). The viability and lack of overt phenotypes in all eight strains suggest that the fusion proteins retain at least partial functionality, as strong loss-of-function mutations in *nekl–mlt* genes are known to cause larval lethality (Lazetic and Fay, 2017a; Yochem et al, 2015).

Eight independent experiments (i.e., biological replicates; Exp 1–Exp 8) were carried out over 16 months using different combinations of bait–TurboID fusions, along with control strains. Experiments were carried out using DIA–MS pipelines except for Exp 5, which used a DDA–MS pipeline. As described below, Exp 1,

2, 4, and 6–8 were carried out using our "standard" experimental pipeline, whereas Exp 3 was carried out in strains depleted of endogenously biotinylated proteins (see below and "Methods and protocols"). In addition, Exp 7 was carried out with strains that were exposed to auxin, a treatment not expected to alter the normal biotinylation profiles of (otherwise) wild-type strains (see below for details).

For most experiments, we carried out three technical replicates per genotype or strain (the remaining experiments included four technical replicates). LC-MS/MS was carried out by the IDeA National Resource for Quantitative Proteomics at the University of Arkansas Medical Center. Spreadsheets containing unprocessed, filtered, labeled, and overlapping data and analysis for Exp 1–Exp 8 are contained in Datasets EV1–8; Datasets EV9–14 include additional analyses described below.

## Outline of the experimental approach

A schematic of the experimental flow is shown in Fig. 1C. To detect a wide range of targets, we carried out studies using mixed-stage populations of C. elegans. Worms were grown at 25 °C to maximize TurboID enzymatic activity (Branon et al, 2018), using NGM medium containing OP50 Escherichia coli without exogenous biotin supplementation. As described in the Introduction, the covalent modification of proteins by biotin at lysine residues on proximal (prey/target) proteins allows for their facile capture and enrichment on streptavidin beads. Following a series of washes, the streptavidin beads were kept frozen until on-bead digestion (trypsinization) was carried out ("Methods and protocols"). We note that on-bead digestion has been reported to preferentially release non-biotinylated peptides from streptavidin beads, while biotinylated peptides remain bound and are consequently underrepresented or absent in downstream LC-MS/MS analyses (Fig. 1C) (Cheah and Yamada, 2017; Cirri et al, 2024; Kim et al, 2018; Li et al, 2021a; Rich et al, 2024; Schiapparelli et al, 2014; Shin et al, 2024). Following on-bead digestion, LC-MS/MS was performed on the released peptides using a DIA pipeline and data were searched against the UniProt C. elegans database (Exp 1–4, 6–8; Appendix Supplementary Methods S2) (UniProt, 2025).

Figure 1D shows an example western blot used to detect biotinylated proteins from the lysates of mixed-stage wild-type (N2; no-TurboID), NEKL-2::TurboID, and NEKL-3::TurboID strains. Full-sized western blots for all our experiments are available in Appendix Fig. S1. The major ~75-kDa band in all three lanes, as well as several of the fainter bands, corresponds to isoforms of four endogenously biotinylated carboxylases present in wild-type worms (PYC-1, PCCA-1, MCCC-1, and POD-2) and may serve as an internal loading control (Artan et al, 2022). In sample lysates containing the bait–TurboID fusions, additional faint bands may also be detected, which primarily represent de novo biotinylation products produced by TurboID (Fig. 1D). We note that the detection and resolution of TurboID-specific biotinylation products by western blots were variable and not necessarily predictive of the quality of the corresponding proteomic data. Moreover, the detection of biotinylated proteins may be difficult if the bait–TurboID fusion is expressed at low levels or in only a subset of cells used to generate the lysate. FLAG tags at the C terminus (Exp 1–7) or hemagglutinin (HA) tags at the N terminus (Exp 8) of TurboID allowed for the detection of the TurboID fusions by

western blotting, whereas actin served as an independent loading control for crude lysates (Fig. 1D).

Consistent with previous reports, we found that a 1-hour incubation of worm lysate with streptavidin beads was sufficient to bind most biotinylated proteins from the lysate (Fig. 1E) (Artan et al, 2021; Branon et al, 2018). Conversely, longer incubation times may result in sample degradation and/or increased non-specific binding of non-biotinylated proteins to the beads. The observed progressive depletion of NEKL-3::TurboID from the lysate during the 16-hour time course as determined by FLAG immunoblotting (Fig. 1E,E') may reflect ongoing cis (self) biotinylation of NEKL-3::TurboID and its subsequent retention on streptavidin beads. This observation raises a potential point of concern, namely that ongoing biotinylation within the lysate during bead incubations could lead to the non-physiological tagging of proteins, with longer incubation times exacerbating such artifacts.

Figure 1F shows an example western blot (Exp 6, replicate 3) in which C-terminal TurboID fusions consisting of all five NEKL–MLT complex members were tested. This blot includes a non-bait-specific TurboID control strain (mNeonGreen::TurboID), which, like the NEKLs and MLTs, is expressed specifically in the epidermis (via the dpy-7 promoter), albeit at substantially higher levels (also see below). Moreover, the expression levels of individual NEKL–MLT TurboID fusions varied considerably, with NEKL-3 and MLT-3 typically being expressed at the highest levels, NEKL-2 and MLT-4 being expressed at intermediate levels, and MLT-2 being expressed at the lowest levels (Fig. 1D,F; Appendix Fig. S1). The expression levels observed on western blots were consistent with the relative levels previously observed when imaging endogenously tagged fluorescent reporters for NEKL–MLT proteins in live cells (Joseph et al, 2023; Lazetic and Fay, 2017a). As described below, low expression levels of a bait may limit the ability to detect proximal targets.

## Identification of other complex members by proximity labeling baits

One expectation of proximity labeling is that bait–TurboID fusions are likely to show high levels of cis biotinylation and should therefore be highly enriched in streptavidin pulldowns versus N2 controls (Fig. 2A). Likewise, known binding partners of the bait proteins should also be enriched via trans biotinylation (Fig. 2A), although this may depend on several factors including their physical distance, expression levels, and extent of association with the bait in vivo. Importantly, we observed both cis and trans biotinylation of NEKL and MLT proteins in all our experiments, albeit to varying degrees (Figs. 2B, 3A–C, and EV2; Datasets EV1–8).

In the case of NEKL-3::TurboID and MLT-3::TurboID, we observed enrichment values versus N2 that were as high as 200–400-fold resulting from cis biotinylation (Figs. 2B, 3A,B, and EV2). Moreover, trans biotinylation of MLT-3 by NEKL-3::TurboID and of NEKL-3 by MLT-3::TurboID led to enrichments ranging from 11- to 65-fold (Figs. 2B and 3A,B). This result is consistent with our previous findings showing that NEKL-3 and MLT-3 bind directly to each other and exhibit strong co-localization in vivo (Lazetic and Fay, 2017a). NEKL-2::TurboID, MLT-2::TurboID, and MLT-4::TurboID also showed consistent but lower levels of cis biotinylation (6.1- to 19-fold), which may be due

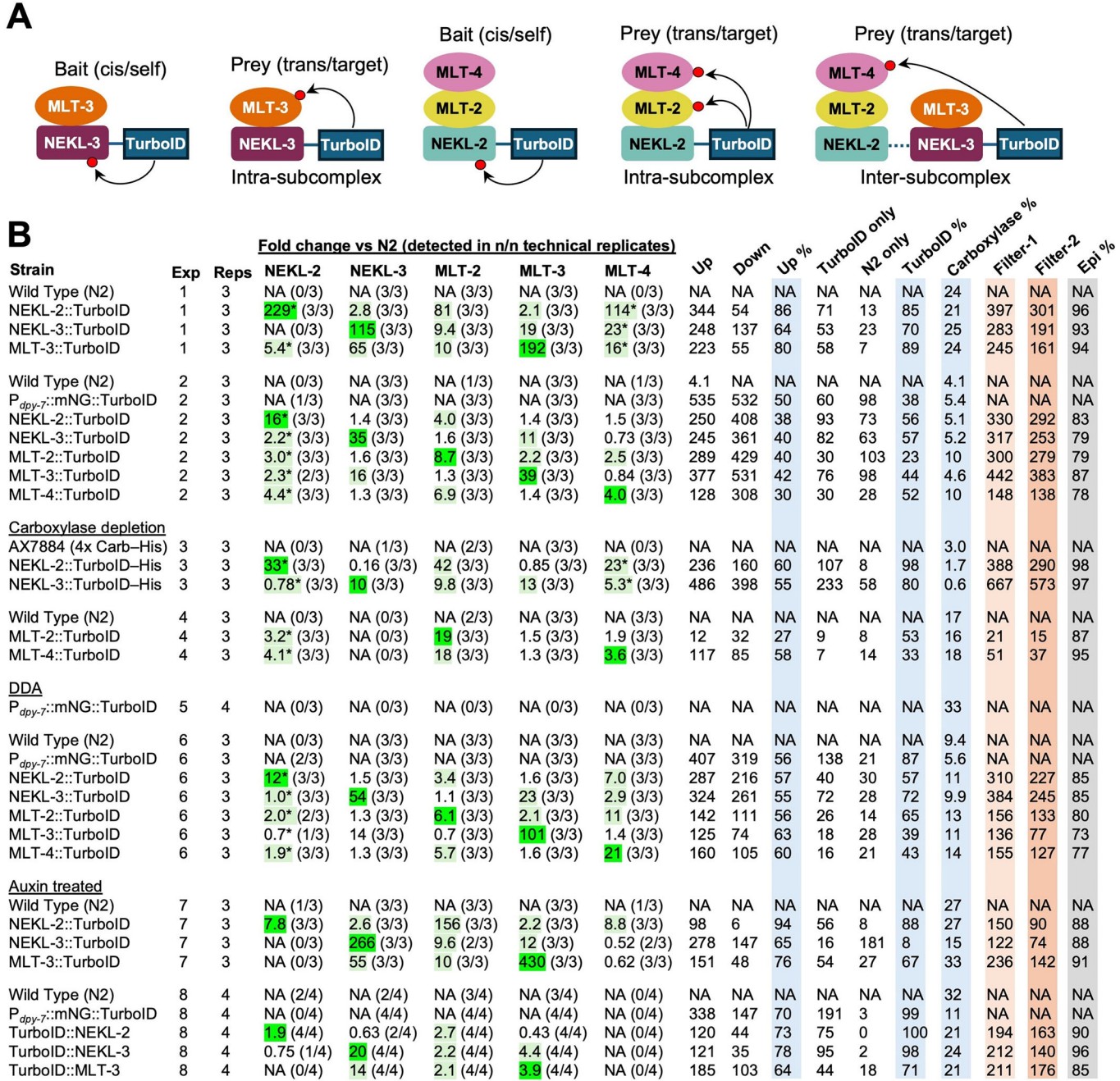

**Figure 2. Summary of results.**

(A) Schematic of biotinylation outcomes for bait (cis/self) and prey (trans/target) proteins within the NEKL–MLT subcomplexes. Red circles indicate biotin. (B) Table of key metrics associated with each experiment. reg regulation, Exp experiment number, Reps. number of technical replicates, NA not applicable, Epi epidermal, DDA data-dependent acquisition. Asterisks indicate estimated fold changes for bait or prey proteins that were not detected in the N2 samples (Dataset EV9). MS experiments were carried out using DIA methods except for Exp 5. Dark green highlighting indicates cis fold changes corresponding to the bait protein; light green highlighting indicates prey proteins that passed Filter-2 (see below and Fig. 5). Light blue highlighting indicates the three main calculated experimental parameters. Supporting information and calculations are available in Datasets EV1–9. Metrics shown on the top row are defined in the text and Fig. 4A. Source data are available online for this figure.

to their lower expression levels and to sensitivity limitations of the assay (Figs. 1E, 2B and 3A,B; Appendix Fig. S1). Nevertheless, these bait–TurboID samples generally identified their subcomplex binding partners, attesting to the robustness of the approach (Figs. 2B and 3C; Datasets EV1–8).

Although the highest level of enrichment by trans biotinylation occurred for components within each of the two NEKL–MLT subcomplexes, we also observed trans biotinylation across the two subcomplexes (Figs. 2A,B, 3A,B, and EV2). For example, in Exp 1, MLT-3::TurboID identified NEKL-2, MLT-2, and MLT-4, whereas

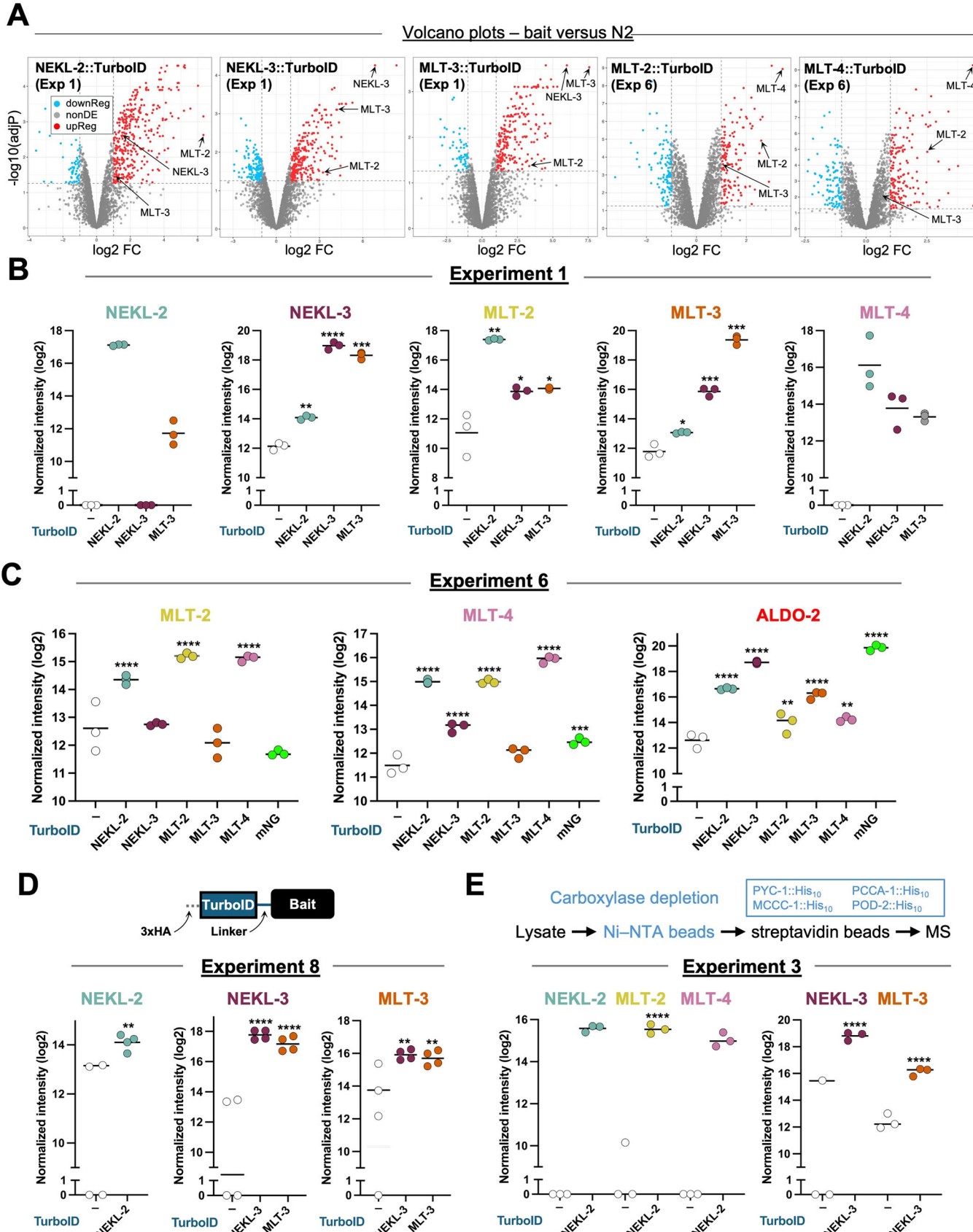

**Figure 3.  Selected data from proximity labeling experiments.**

(A) Volcano plots of bait–TurboID strains versus the N2 (no–TurboID) control, indicating the $\log_2$ fold change (FC) versus $-\log_{10}$ adjusted $P$ value (adjP). Red and blue dots indicate proteins that were up and down, respectively, in the bait strain relative to the N2 strain; select identified bait and prey proteins are indicated by arrows. NonDE not differentially expressed. (B–E) Intensity plots ($\log_2$ scale) showing the abundance of the specified protein with each of the bait–TurboID strains for the indicated experiments (B, Exp 1; C, Exp 6; D, Exp 8; E, Exp 3); bait–TurboID strains are indicated below the x axis. Individual dots correspond to technical replicates. Adjusted $P$ values were calculated using Spectronaut software and are based on comparisons between the bait–TurboID and the N2 control; ****$P \leq 0.0001$, ***$P \leq 0.001$, **$P \leq 0.01$, *$P \leq 0.05$. See S2 Information for details. We note that values of zero indicate that the protein was not detected for a given sample. Protein quantity values on the y axes were calculated using Spectronaut software and indicate relative amounts (intensity values) within each experiment but are not easily comparable between experiments. Data on intensities, fold changes, and $P$ values are available in Datasets EV1, 3, 6, 8. (D) Includes a diagram of the N-terminal TurboID fusions. (E) Includes a flow chart of the carboxylase depletion steps described in the text. Source data are available online for this figure.

NEKL-2::TurboID identified NEKL-3 and MLT-3. This finding is consistent with a lower but detectable level of co-localization between components of the two subcomplexes as well as data indicating interactions between the two subcomplexes in humans and AlphaFold predictions for the *C. elegans* proteins (Fig. EV1A–E) (Belham et al, 2003; de Souza et al, 2014; Joseph et al, 2023; Lazetic and Fay, 2017a; Leettola et al, 2014; Richards et al, 2009; Vaz Meirelles et al, 2010; Yakulov et al, 2015).

In several cases, the complete absence of a bait or prey protein in all technical replicates of the N2 control sample made it impossible to assign a precise fold change for the bait–TurboID sample or to calculate a $P$ value (Figs. 2B and 3B,E). For Fig. 2B, we therefore estimated conservative fold changes (indicated by asterisks), as described in the "Methods and protocols" and Dataset EV9 (tab 1). Notably, our estimates showed a similar trend of enrichment such that members within the same subcomplex exhibited higher levels of trans biotinylation than did members from different subcomplexes (Figs. 2A,B, and 3A,B). Taken together, these findings indicate that our proximity labeling experiments were successful in identifying known NEKL–MLT binding partners.

## General metrics for assessing experimental outcomes

As described above, identification of the bait protein and its known (prey) binding partners, along with their degree of enrichment, provides one means of assessing the relative success of a given experiment. Here, we discuss several additional metrics that may be useful for evaluating experimental quality and examine their correlation with each other and with the enrichment of bait and prey proteins. These metrics may be particularly valuable when little is known about the binding partners of the bait protein of interest.

Volcano plots are useful for evaluating the overall outcomes of proximity labeling experiments, as they visually convey the number of differentially regulated proteins, the direction and magnitude of fold changes, and the associated $P$ values (Fig. 3A). In addition, successful experiments are expected to identify a greater number of proteins in the bait–TurboID samples as compared with the no-TurboID N2 controls, resulting in an asymmetric distribution of data points. This pattern is exemplified in Fig. 3A, which shows volcano plots for NEKL-2::TurboID (Exp 1), NEKL-3::TurboID (Exp 1), MLT-3::TurboID (Exp 1), MLT-2::TurboID (Exp 6), and MLT-4::TurboID (Exp 6). Similar asymmetries were observed in most other experiments, although the degree of asymmetry varied substantially between different bait–TurboID fusion strains and across biological replicates (Fig. EV3). In general, proteins located in the upper right quadrant of the volcano plots are of greatest

interest, consistent with the presence of the bait and known subcomplex members in this region (Fig. 3A).

Figure 2B lists the number of proteins that were enriched in the bait–TurboID samples for each experiment based on a ≥twofold change (versus N2) and an adjusted $P \leq 0.05$; we refer to these as "up" proteins (Datasets EV1–8). Likewise, Fig. 2B shows the number of proteins that were reduced in the bait–TurboID samples versus the N2 samples, which we refer to as "down" proteins. Note that if a protein is reduced in the bait–TurboID sample, this is identical to stating that it is enriched in the N2 control. We calculated the percentage of bait-enriched proteins (Up %) for each experimental sample relative to the total number of up and down proteins, which provides a straightforward quantitative summary of the volcano plot data (Figs. 2B and 4A). Overall, 21/26 TurboID samples showed a higher proportion of up proteins relative to N2, although this percentage ranged from 94% to as low as 27%. Nevertheless, even when the Up % was well below 50%, the bait protein and one or more of its known NEKL–MLT binding partners were still detected, indicating that informative data were present in all experimental samples (Fig. 2B; Datasets EV1–8). A low Up % could indicate a failure to efficiently enrich for biotinylated proteins, which may occur because of insufficient bead washing. Alternatively, a low Up % may be a consequence of low expression levels of the bait protein or weak activity of the TurboID enzyme. These factors, combined with the inherent background present in all studies, may account for the counter-intuitive enrichment of some proteins in the no-TurboID (N2) control samples. We note that we did not observe a consistent pattern of overlap among the proteins enriched in the N2 samples across a tested subset of experiments, suggesting that this class of proteins may arise largely at random and is variable between experiments (Fig. EV4A–C).

Because the volcano plots are based on fold change relative to N2, many proteins of interest may be absent from these plots if they were not detected in the N2 samples. This is exemplified by NEKL-2 and MLT-4 in Exp 1, which were both absent from the N2 samples (Fig. 3B). To capture this category of targets, we identified proteins that were missing in all technical replicates of the N2 sample but were present in all replicates of the bait–TurboID samples; we refer to these as "TurboID-only" proteins (Figs. 2B and 4A; Datasets EV1–8). Likewise, we identified "N2-only" proteins based on their presence in the N2 samples only. As expected, the number of TurboID-only proteins was greater than the number of N2-only proteins in most TurboID experimental samples (19/26; Fig. 2B). We next calculated the percentage of TurboID-only proteins for each experimental sample relative to the total number of Turbo ID-only and N2-only proteins (TurboID

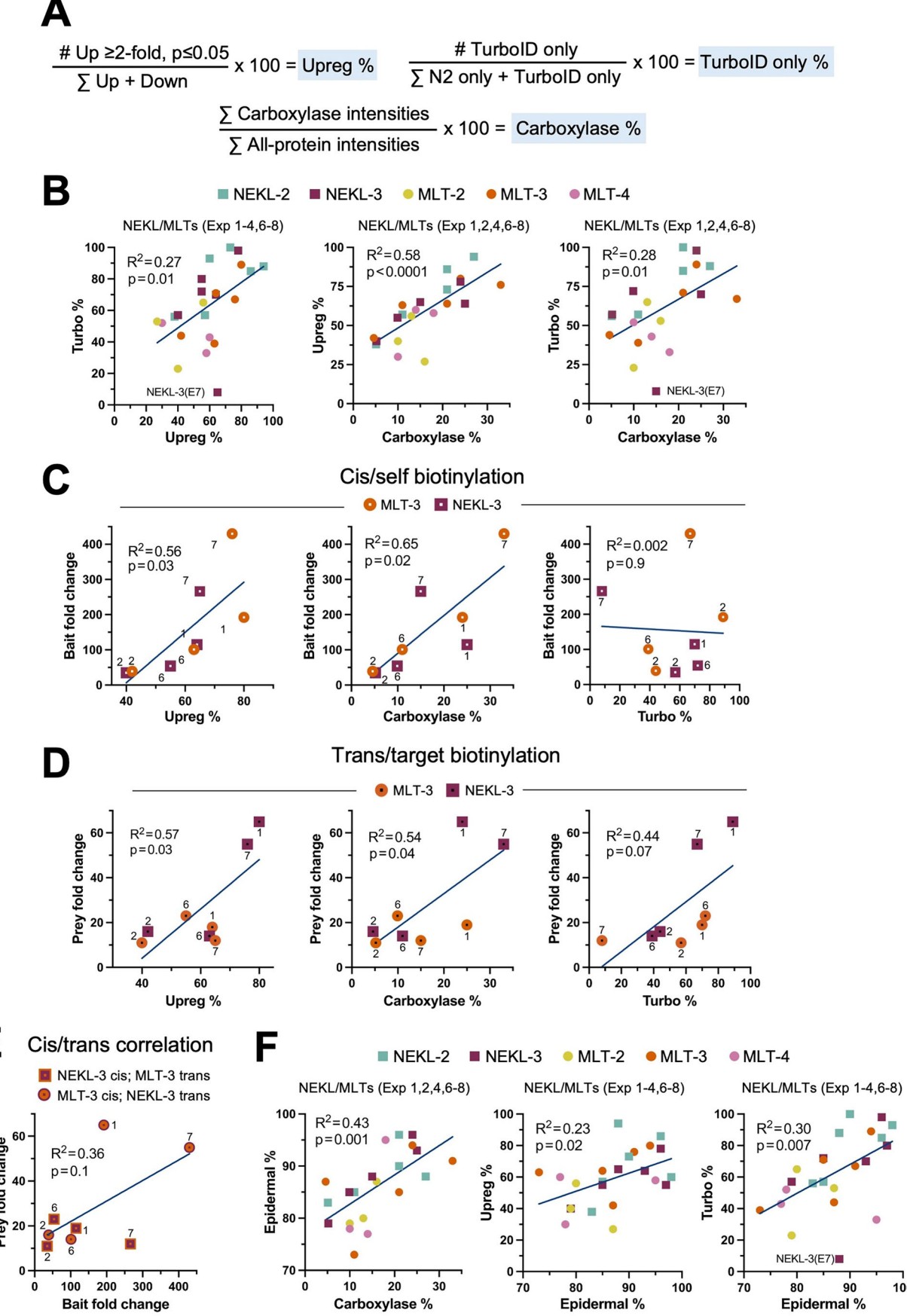

**Figure 4. Correlation of experimental metrics.**

(**A**) Simple equations defining the three main experimental parameters shown in Fig. 2B; Up %, Carboxylase %, and TurboID %. (**B–F**) Scatter plots showing simple linear regression and statistics. Coefficient of determination ($R^2$) values were based on a simple linear regression model and calculated using Prism software. Data points in (**C–E**) indicating bait or prey proteins are displayed at the top of each plot; numbers adjacent to the data points indicate the experiment. (**B**) Scatter plots from the indicated experiments examining the correlation between the parameters defined in (**A**). NEKL-3 Exp 7, which showed high variability between technical replicates, is indicated in (**B, F**); also see Fig. 5. (**C**) Scatter plots of NEKL-3::TurboID and MLT-3::TurboID bait (cis) fold changes versus the parameters defined in (**A**). (**D**) Scatter plots of NEKL-3 and MLT-3 (trans) fold changes in MLT-3::TurboID and NEKL-3::TurboID strains, respectively, versus the parameters defined in (**A**). (**E**) Scatter plot comparing cis and trans fold changes within individual experiments for NEKL-3::TurboID and MLT-3::TurboID strains. (**F**) Plots from the indicated experiments examining the correlation between the parameters defined in (**A**) and the percentage of epidermally expressed proteins (Epidermal %). Source data are available online for this figure.

%; Fig. 4A) and found that TurboID % positively and significantly correlated with Up % (Figs. 2B and 4B; Dataset EV9). As such, a high TurboID % may serve as an additional indicator of experimental success. Nevertheless, up proteins outnumbered TurboID-only proteins by a ratio of ~4:1 across all experiments (Fig. 2B; Datasets EV1–8). Collectively, our findings suggest that the TurboID-only category may include some of the most physiologically relevant proteins in our datasets, and these proteins were therefore included in our downstream analyses.

Our observation that experimental outcomes varied considerably across biological replicates for individual strains prompted us to investigate potential underlying sources of this variability (also see below). We reasoned that if samples entering the mass spectrometer are highly enriched for biotinylated proteins and contain minimal levels of non-biotinylated (non-specific or "sticky") proteins, then the relative abundance of the four endogenously biotinylated carboxylases should be correspondingly high in the total sample (Carboxylase %; Fig. 4A). Carrying out these calculations, we found that Carboxylase % in Exp 1, 2, 4, and 6–8 ranged from 4.1% to 33%, indicating substantial variability in this parameter (Fig. 2B; Dataset EV9, tab 2). Exp 3 was not included in this analysis because, as described below, these lysates underwent a carboxylase depletion step prior to incubation with streptavidin beads.

Carboxylase % positively and significantly correlated with both Up % and TurboID % (Fig. 4B), indicating that these metrics may be innately interconnected. We speculate that a low Carboxylase % may indicate a higher proportion of contaminating non-biotinylated proteins in the sample, thereby increasing background noise and reducing both sensitivity and overall data quality. This may occur because of insufficient washing of the streptavidin beads or may be due to the properties of individual lysates or reagents. Regardless, the proportion of the endogenously biotinylated carboxylases may provide another useful metric for assessing experimental quality. We note, however, that a reduced Carboxylase % may be anticipated if the bait–TurboID protein is expressed at high levels, such that its biotinylated products make up a substantial proportion of the total biotinylated protein pool in the sample. In fact, we previously observed this phenomenon with PTPN-22::TurboID (Binti et al, 2024b), as well as in the current study with the mNG::TurboID control strain (Figs. 1F and 2B; Appendix Fig. S1).

Lastly, we sought to determine how the three parameters described above (Up %, TurboID %, and Carboxylase %) correlated with the extent of bait and prey enrichment in our experiments. For these comparisons we focused on NEKL-3 and MLT-3 (Exp 1, 2, 6, 7), because both proteins were consistently detected in the NEKL-3::TurboID and MLT-3::TurboID samples, as well as in the N2

control. The enrichment of bait–TurboID proteins via cis biotinylation showed a positive and significant correlation with both Up % and Carboxylase %, but not with TurboID % (Fig. 4C; Dataset EV9), possibly due to limited sampling. For NEKL-3 and MLT-3 trans biotinylation by MLT-3::TurboID and NEKL-3::TurboID, respectively, we observed positive correlations with all three metrics, with the strongest associations seen for Up % and Carboxylase % (Fig. 4D). We also note a weak but positive correlation between the fold changes of cis–bait and trans–prey proteins within individual experiments, although based on our small dataset this did not reach statistical significance (Fig. 4E). Together, our findings highlight several straightforward metrics for assessing the potential success or quality of proximity labeling studies, which may be particularly useful in situations where the bait protein has no known binding partners. At the same time, our results highlight several key challenges inherent to proximity labeling studies, including experimental variability, background noise, and difficulties in detecting the targets of low-abundance bait proteins.

## Observations regarding technical replicates

A critical requirement for any successful proteomics study includes a strong correlation among technical replicates. We defined technical replicates as samples prepared from worms plated on the same day and grown under identical conditions but separated prior to lysis and streptavidin-based enrichment. Figure 3 shows data for several genes from Exp 1, 3, 6, and 8, most notably the NEKLs and MLTs themselves. This small subset reveals relatively close agreements among the technical replicates but also some variability, which may occur more frequently with lower-abundance proteins that are closer to the limits of detection.

To examine technical replicates more systematically, we assessed both the extent of protein overlap and the degree of expression-level correlation across multiple experiments. Figure 5A shows protein overlaps for the three technical replicates from N2, NEKL-2::TurboID, NEKL-3::TurboID, and MLT-3::TurboID for Exp 1, along with N2 and NEKL-3::TurboID for Exp 2 (also see Dataset EV10). Approximately 3700 unique proteins were identified for each strain, with 84–92% overlap occurring among each of the three technical replicates. As discussed below, a similarly high level of overlap was observed in most of our other experiments. We conclude that in most cases our technical replicates identified strongly overlapping subsets of proteins within a given strain.

Scatterplot analyses also indicated reasonably strong correlations in the abundance levels of proteins between the technical replicates for each strain in Exp 1 (Fig. 5B; Dataset EV10). In the case of N2 and NEKL-2::TurboID, $R^2$ values ≥0.89 were observed

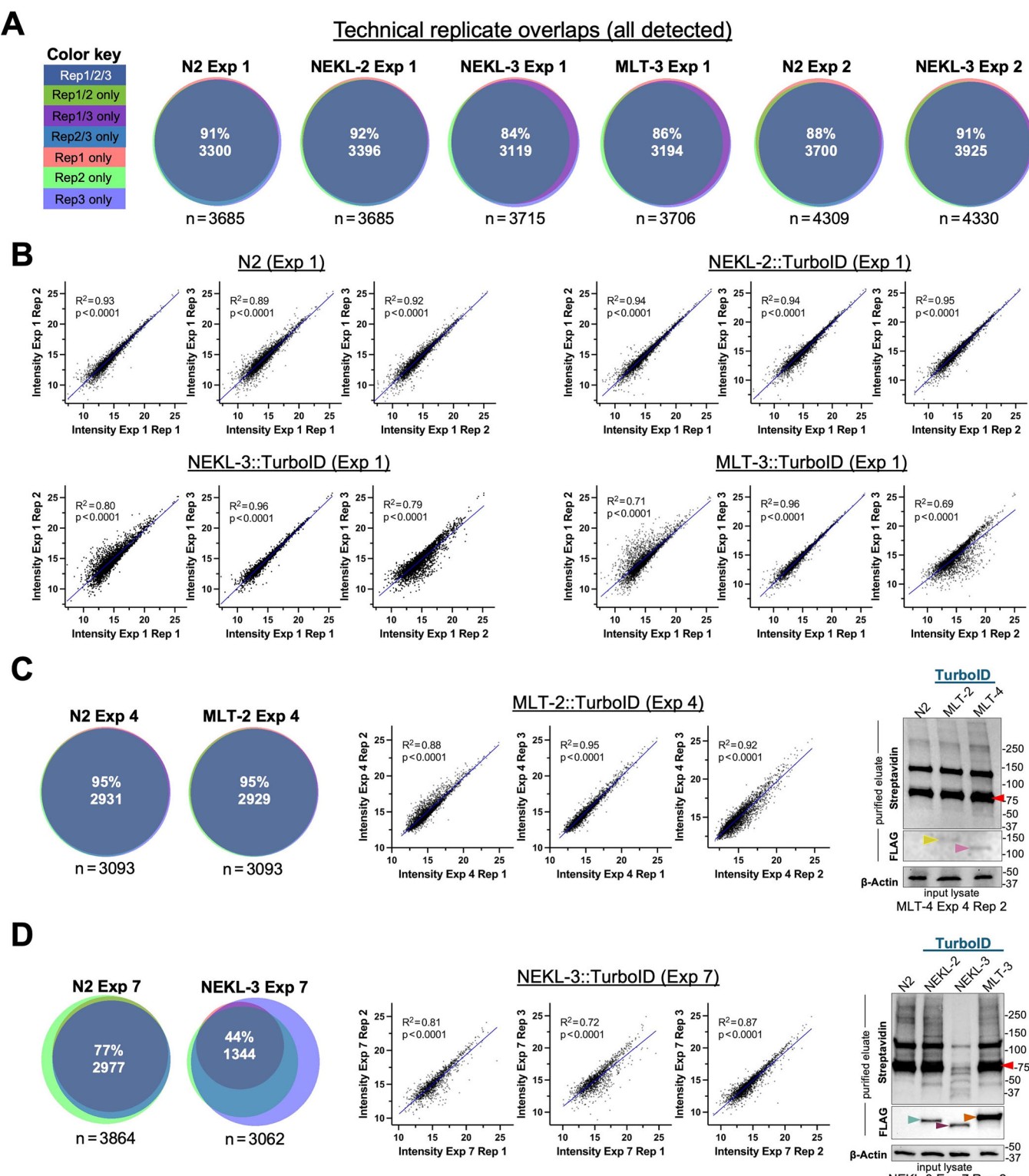

when we tested all three replicate pairs (1 vs 2, 1 vs 3, and 2 vs 3) including $R^2$ values as high as 0.95 (Fig. 5B). NEKL-3::TurboID and MLT-3::TurboID exhibited a wider range of $R^2$ values (0.69–0.96), indicating some variability arising from one of the technical replicates from each strain (Fig. 5B). Despite this variability,

however, the data from NEKL-3::TurboID and MLT-3::TurboID were among the strongest in our collective dataset, based in part on the parameters discussed above (Figs. 2B and 3B). We further analyzed the NEKL-3::TurboID and MLT-3::TurboID datasets using the subset of proteins that was up ≥twofold (irrespective of

Figure 5. Evaluation of technical replicates.

(A, C, D) Proportional Venn diagrams showing the extent of overlap in the proteins detected by technical replicates from the indicated strains in (A) Exp 1 and 2, (C) Exp 4, and (D) Exp 7; *n* indicates the total number of unique proteins. Color key for (C, D) is shown in (A). (B–D) Scatter plots showing correlations in protein abundance (intensities) between replicate pairs from the indicated experiments; dots correspond to individual proteins detected in both replicates. $R^2$ values were calculated using simple linear regression. (B) Scatter plots of technical replicates from N2, NEKL-2::TurboID, NEKL-3::TurboID, and MLT-3::TurboID of Exp 1. (C) (left) Overlaps for technical replicates from N2 and MLT-2::TurboID of Exp 4. (Center) Scatter plots of MLT-2::TurboID (Exp 4) technical replicates. (Right) Western blots corresponding to Exp 4, technical replicate 2 showing biotinylation (streptavidin; eluate), bait proteins (FLAG; eluate), and loading control (β-actin; lysate). Colored arrowheads correspond to MLT-4::TurboID (pink), MLT-2::TurboID (yellow), and the major carboxylase species (red). Note that MLT-2::TurboID is expressed at levels close to the limits of detection. (D) (left) Overlaps for technical replicates from N2 and NEKL-3::TurboID Exp 7. Note the relative lack of overlap for NEKL-3::TurboID due to reduced numbers of proteins identified in replicates 1 and 2. (Center) Scatter plots of NEKL-3::TurboID (Exp 4) technical replicates. (Right) Western blots corresponding to Exp 7, technical replicate 2. Colored arrowheads correspond to NEKL-2::TurboID (teal), NEKL-3::TurboID (purple), MLT-3::TurboID (orange), and the major carboxylase species (red). Note reduced levels of signal in the NEKL-3 eluate. Raw data are available in Dataset EV10. Source data are available online for this figure.

*P* value) to see whether these would show stronger correlations, but no such trend was observed (Fig. EV5A,B; Dataset EV10). In contrast, as might be expected, we did observe stronger correlations between NEKL-3::TurboID and MLT-3::TurboID technical replicates when we filtered for proteins that were up (≥twofold) and had an adjusted $P \le 0.05$ (Fig. EV5A,B; Dataset EV10). We conclude that as few as three technical replicates may be adequate for many studies and that this experimental pipeline is sufficiently robust to compensate for relatively minor variability between technical replicates.

Several additional observations regarding technical replicates are worth noting. Exp 2 showed very strong correspondence between technical replicates (N2 and NEKL-3::TurboID; Figs. 5A and EV5C; Dataset EV10) but produced only weak enrichment for the bait and prey proteins (Fig. 2A). Notably, these samples had low Up % and Carboxylase % values, suggesting a relatively weak enrichment of biotinylated peptides (Figs. 2B and 4). Similarly, Exp 4 showed close correspondence between technical replicates but identified very few MLT-2 proximal interactors (Figs. 2B and EV5D; Dataset EV10). Here, the MLT-2::TurboID sample also exhibited a very low Up % value, which may be due, in part, to the low expression levels of MLT-2::TurboID (Figs. 1E and 5C; Appendix Fig. S1). From these examples, we conclude that a tight correspondence between technical replicates alone is not a strong predictor of experimental success. One instructive case, however, is shown for NEKL-3::TurboID (Exp 7), which had only 44% overlap among the three replicates (Fig. 5D; Dataset EV10). As might be expected, this led to fewer differentially expressed proteins being identified in this experiment than in the four other NEKL-3::TurboID experiments (Fig. 2B). This anomaly can be traced to issues with our sample preparation, most notably the eluates from technical replicates 1 and 2 (Fig. 5D; Appendix Fig S1). Nevertheless, we note that the reduced subset of overlapping proteins identified in this experiment still showed reasonably strong correlations in protein abundance among the technical replicates (Figs. 5D and EV5E; Dataset EV10). To summarize, although a relatively close agreement among technical replicates is clearly critical for experimental success, it offers limited utility as a standalone predictor.

## Strategies for the identification of false positives

Here, we outline approaches for evaluating whether proteins identified by proximity labeling represent true proximal interactors or are more likely to be false positives. As noted above, some false positives may arise due to random chance or sample variability, such as when non-biotinylated proteins adhere to streptavidin beads. Although not highly systematic, such background noise can hinder the detection of true positives. Another class of false positives, one that more stringent washes to remove non-biotinylated proteins would not address, consists of proteins that are biotinylated in a TurboID-dependent manner but are not bait-specific (Fig. 6A). Such "promiscuously" biotinylated proteins may be relatively abundant and contain accessible surface lysine residues. Moreover, the enrichment for this class of false positives is non-random and thus reproducible.

To identify non-specific biotinylated targets in our datasets, we employed an established control strategy in which a non-targeted TurboID strain was analyzed in parallel with our bait–TurboID strains. (Artan et al, 2021; Artan and de Bono, 2022; Jiang et al, 2025; Grismer et al, 2024; Hollstein et al, 2022; Holzer et al, 2022; Moreira et al, 2023; Nikonorova et al, 2025). Specifically, we used a strain that expresses a non-localized mNeonGreen–TurboID fusion protein under the control of the *dpy-7* epidermal-specific promoter (mNG::TurboID) (Artan et al, 2021). Although expressed at substantially higher levels than the NEKLs and MLTs (Fig. 1E), this fusion protein would nevertheless be expected to biotinylate many of the same non-bait-specific targets as the NEKL–MLT TurboID fusions. In total, we carried out three independent experiments (Exp 2, Exp 6, and Exp 8) comparing mNG::TurboID to N2 using our standard DIA–MS pipeline along with a single experiment using DDA methods (Figs. 2B, 6B, and EV6A,B; Dataset EV11). As shown in Fig. 2B, two of our DIA mNG::TurboID experiments (Exp 6 and Exp 8) had higher Up %, TurboID %, and Carboxylase % values than those associated with Exp 2, suggesting higher-quality data. Moreover, based on linear regression, the fold changes of proteins (vs N2) correlated more strongly between Exp 6 and Exp 8 than between either of these experiments and Exp 2 or Exp 5 (Figs. 6C and EV5C,D; Dataset EV11). For these reasons, we carried out annotation and subtraction steps using the combined mNG::TurboID DIA data from Exp 6 and Exp 8.

From Exp 6 and Exp 8, we identified 378 proteins that were present in all technical replicates of both samples and showed significant enrichment (adjusted $P \le 0.05$) in the mNG::TurboID sample versus N2. Of these, 212 had mean fold changes of 2–132 (N2$^{6/8}$ G$^{6/8}$; Fig. 6D,E; Dataset EV11). Another 35 proteins showed ≥twofold enrichment (and adjusted $P \le 0.05$) in Exp 6 but were absent from the N2 samples in Exp 8 (N2$^6$ G$^{6/8}$), whereas 14 proteins showed ≥2-fold enrichment (and adjusted $P \le 0.05$) in Exp

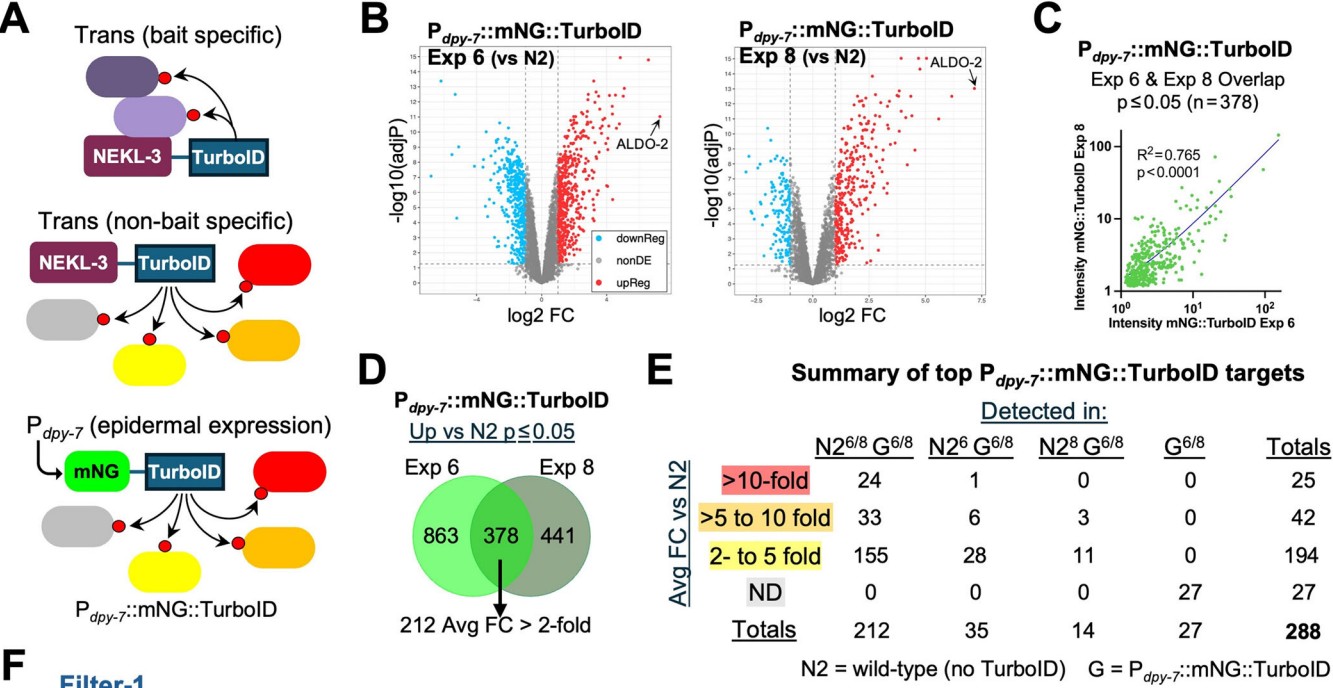

**B**

$P_{dpy-7}$::mNG::TurboID
Exp 6 (vs N2)

$P_{dpy-7}$::mNG::TurboID
Exp 8 (vs N2)

*y-axis:* -log10(adjP)
*x-axis:* log2 FC

ALDO-2

downReg
nonDE
upReg

**C**

$P_{dpy-7}$::mNG::TurboID
Exp 6 & Exp 8 Overlap
$p \leq 0.05$ (n = 378)

$R^2 = 0.765$
$p < 0.0001$

*y-axis:* Intensity mNG::TurboID Exp 8
*x-axis:* Intensity mNG::TurboID Exp 6

**D**

$P_{dpy-7}$::mNG::TurboID
Up vs N2 $p \leq 0.05$

Exp 6    Exp 8

863    378    441

212 Avg FC > 2-fold

**E**

Summary of top $P_{dpy-7}$::mNG::TurboID targets

Detected in:

| Avg FC vs N2 | N2[6/8] G[6/8] | N2[6] G[6/8] | N2[8] G[6/8] | G[6/8] | Totals |
|---|---|---|---|---|---|
| >10-fold | 24 | 1 | 0 | 0 | 25 |
| >5 to 10 fold | 33 | 6 | 3 | 0 | 42 |
| 2- to 5 fold | 155 | 28 | 11 | 0 | 194 |
| ND | 0 | 0 | 0 | 27 | 27 |
| Totals | 212 | 35 | 14 | 27 | **288** |

N2 = wild-type (no TurboID)    G = $P_{dpy-7}$::mNG::TurboID

**F**

**Filter-1**

- Detected in all bait (NEKL–MLT) TurboID replicates (e.g., 3/3)
- ≥2-Fold upregulation in bait–TurboID vs N2
- Adjusted $p \leq 0.05$

**OR**

- Detected in all bait–TurboID replicates
- Not detected in any N2 replicates (e.g., 0/3)

**Filter-2**

- Remove all proteins with $P_{dpy-7}$::mNG::TurboID vs N2 fold change of >5
- Calculate ratio of bait–TurboID fold change to $P_{dpy-7}$::mNG::TurboID fold change
- Remove all proteins with bait–TurboID to $P_{dpy-7}$::mNG::TurboID ratio of <1.5

**G**

- 🟩 NEKL–MLT (ND in $P_{dpy-7}$::mNG::TurboID)
- ⬜ $P_{dpy-7}$::mNG::TurboID vs N2 Up <2-fold or ND in $P_{dpy-7}$
- ⬛ Detected in $P_{dpy-7}$::mNG:: TurboID; ND in N2
- 🟨 $P_{dpy-7}$::mNG::TurboID vs N2 Up 2- to 5-fold
- 🟧 $P_{dpy-7}$::mNG::TurboID vs N2 Up >5 to 10-fold
- 🟥 $P_{dpy-7}$::mNG::TurboID vs N2 Up >10-fold

**H**

Individual    Overlaps    Unique

Filter-1

*y-axis:* Number of proteins

NEKL-2  NEKL-3  MLT-3    All 3  NEKL-2 NEKL-3  NEKL-2 MLT-3  NEKL-3 MLT-3    NEKL-2  NEKL-3  MLT-3

Filter-2

*y-axis:* Number of proteins

NEKL-2  NEKL-3  MLT-3    All 3  NEKL-2 NEKL-3  NEKL-2 MLT-3  NEKL-3 MLT-3    NEKL-2  NEKL-3  MLT-3

**I**

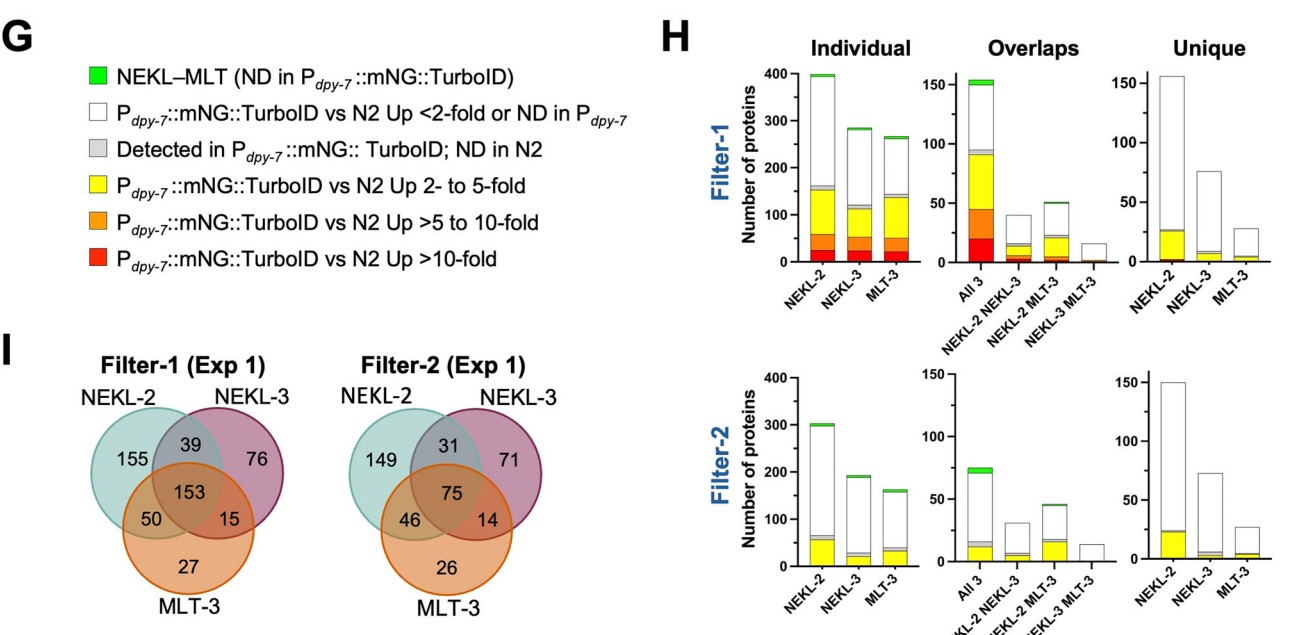

Filter-1 (Exp 1)

NEKL-2    NEKL-3

155    39    76

50    153    15

MLT-3

Filter-2 (Exp 1)

NEKL-2    NEKL-3

149    31    71

46    75    14

MLT-3

◄ **Figure 6.   Identification and filtering of non-bait-specific targets.**

(A) Diagram showing specific and non-specific biotinylation by NEKL-3::TurboID along with non-specific biotinylation by the epidermally expressed $P_{dpy-7}$::mNG::TurboID control. Red spheres indicate biotin. (B) Volcano plots of the $P_{dpy-7}$::mNG::TurboID strain versus N2 (Exp 6 and Exp 8) showing $\log_2$ fold change (FC) versus $-\log_{10}$ adjusted $P$ value. ALDO-2, with an average fold change of ~132, is indicated. (C) Scatter plots showing the correlation in the abundance of individual proteins between $P_{dpy-7}$::mNG::TurboID Exp 6 and Exp 8. $R^2$ values were derived using simple linear regression. (D) Venn diagram of proteins detected in Exp 6 and Exp 8 for $P_{dpy-7}$::mNG::TurboID that were enriched (adjusted $P \leq 0.05$) versus N2. (E) Summary of the 288 $P_{dpy-7}$::mNG::TurboID targets used to annotate potential non-bait-specific positives in our datasets. N2, the no-TurboID control; G, the $P_{dpy-7}$::mNG::TurboID strain; superscript numbers indicate Exp 6 and Exp 8. Raw data for (B–E) are available in Dataset EV11. (F) Algorithm for carrying out the Filter-1 and Filter-2 steps to annotate and eliminate false positives. The large arrow indicates that Filter-2 was sequentially applied after Filter-1. (G) Color coding used for annotating Datasets EV1–5, 6–8 and (H). (H) Bar charts showing the number and distribution of annotated proteins from Exp 1. Results from the Filter-1 and Filter-2 datasets are shown, including results for individual baits and protein overlap. (I) Venn diagram showing the extent of protein overlap between the indicated bait protein datasets using Filter-1 and Filter-2. Raw data for (H, I) are available in Dataset EV1. For additional details, see text. Source data are available online for this figure.

8 but were absent from N2 Exp 6 (N2$^8$ G$^{6/8}$; Fig. 6E). In these cases, we made use of the single available fold change and merged these proteins with the 212, leading to 261 proteins that were enriched specifically in mNG::TurboID samples (Fig. 6E; Dataset EV11). Lastly, 27 proteins were present in all replicates of mNG::TurboID (Exp 6 and Exp 8) but were missing in both N2 samples (G$^{6/8}$; Fig. 6E; Dataset EV11). Although no direct fold change or $P$ values could be ascribed to these 27 proteins, these were included in our list of potential false positives, leading to a total of 288 proteins with compelling enrichment in the epidermal mNG::TurboID strain.

Some additional points regarding non-specific TurboID controls are also worth considering. Optimally, it may be best to use a control TurboID (e.g., mNG::TurboID) that is expressed at levels close to that of the bait protein(s) (Artan and de Bono, 2022; Jiang et al, 2025). We attempted to do this by generating a $P_{nekl-3}$::mNG::TurboID construct, but failed to obtain a corresponding integrated strain. Nevertheless, because the NEKLs and MLTs are expressed at different levels, a legitimate question can be raised as to how many different controls would be optimal or practical. Additionally, it may be important to have a control TurboID that localizes within a similar region of the cell as the bait protein, such as within the nucleus or cytoplasm. This level of engineering may be extended to controlling for specific subcellular regions or organelles, although this could increase the risk of the control TurboID biotinylating some authentic targets of the bait.

We also note that in the absence of an internal TurboID control strain, it may be tempting to use data from other published sources. However, variations in experimental conditions and other factors may lead to substantial differences between studies, making this type of approach problematic. For example, our DDA–MS mNG::TurboID study (Exp 5) detected 1303 proteins versus 4359 proteins identified in a previous study by Artan and colleagues (Fig. EV5E; Dataset EV11) (Artan et al, 2021). Moreover, although 1070 (82%) of the proteins we identified overlapped with the Artan dataset, we observed only a weak correlation in the abundance of the detected proteins ($R^2 = 0.074$; Fig. EV5F; Dataset EV11). Such discrepancies may occur for multiple reasons including growth conditions, developmental stage distributions, reagents, wet-lab procedures, and the LC-MS/MS itself, which can be variable between instruments, facilities, and experimental runs. Despite this, in the absence of an internal control, annotating potential false positives based on published datasets may still be useful for prioritizing candidates or alerting others to potential false positives within datasets.

To identify, visually code, and remove potential false positives, we applied a two-step approach (Fig. 6F; Datasets EV1–8). With Filter-1, we retained proteins that had ≥2-fold enrichment relative to N2, had an adjusted $P \leq 0.05$, and were present in all the bait technical replicates. We note that for N2, average abundances were calculated from technical replicates in which the protein was detected, such that a hypothetical finding of 13, not detected (ND), and ND across three replicates would still average to 13. Nevertheless, 83% (639/772) of the bait targets that passed Filter-1 in Exp 1 were also detected in all three N2 replicates. Filter-1 also retained proteins that were present in all bait technical replicates but were absent from all corresponding N2 replicates and included NEKL-2 and MLT-4 (Figs. 2B and 3C; Datasets EV1–8). As summarized in Fig. 2B, Filter-1 led to a wide range of targets per strain (21–667) with an average of 255 (SD = 144, $n = 23$).

We next labeled Filter-1 proteins according to their fold change in the mNG::TurboID control experiments (Fig. 6E,G). Specifically, red, orange, and yellow backgrounds correspond to mNG::TurboID fold change cutoffs of >10, >5–10, and 2–5, respectively (Fig. 6E,G; Datasets EV1–8). Light gray denotes the 27 proteins detected in all mNG::TurboID datasets but not in N2, whereas a white background indicates proteins that were <twofold enriched in the mNG::TurboID dataset or were not detected in the mNG::TurboID datasets (Fig. 6E,G; Datasets EV1–8). Finally, green indicates the NEKL–MLT proteins themselves, none of which were detected in the mNG::TurboID samples (Fig. 6G).

We reasoned that the greater the fold change observed in the mNG::TurboID samples, the greater the likelihood that a given protein is a non-specific target of the bait. Filter-2 therefore removed all proteins that were enriched >fivefold in the mNG::TurboID samples (i.e., proteins with red and orange backgrounds). One example of a protein removed by Filter-2 is ALDO-2, which was enriched by 120- and 144-fold in mNG::TurboID experiments 6 and 8, respectively (Dataset EV11). Moreover, ALDO-2 was up in all five C-terminal bait–TurboID fusions relative to N2, although to a lesser extent than in the mNG::TurboID strain (Fig. 3C).

We anticipated that the yellow-labeled proteins (two- to fivefold up in mNG::TurboID) may contain a mixture of true and false positives. To further parse this category, we calculated the ratio of the fold change in the bait sample versus the mNG::TurboID control for each protein. For example, in Exp 1, EEA-1 was up 17-fold in NEKL-2::TurboID versus 2.2-fold in the mNG::TurboID control, giving a bait-to-control fold change ratio of 7.6 (Dataset EV1). Notably, NEKL-2 co-localizes with EEA-1 and

other markers of early endosomes, and loss of NEKL-2 leads to defects in EEA-1 expression, consistent with EEA-1 being a proximate target of NEKL-2::TurboID (Joseph et al, 2023; Reimann et al, 2025). In contrast, MDT-6, a subunit of the RNA polymerase mediator complex, was up 3.6-fold in MLT-3::TurboID (Exp 1) and 4.0-fold in the mNG::TurboID controls, resulting in a bait-to-control ratio of 0.88, suggesting that MDT-6 may be a non-specific target. Consistent with this, most red- and orange-labeled proteins had fold-change ratios of <1 (Datasets EV1–8). We chose a cutoff such that yellow-labeled proteins with a bait-to-control fold-change ratio of <1.5 were removed from the dataset by Filter-2, resulting in an average of 196 candidates per strain (SD = 124; $n$ = 23; range, 15–573; Fig. 2B).

Figure 6H shows a color-coded summary of results for individual baits in Exp 1 using both filters. For Exp 1, Filter-2 led to a 29% average reduction in the total number of proteins, whereas reductions in the yellow-labeled proteins ranged from 42% (NEKL-2::TurboID) to 68% (NEKL-3::TurboID; Dataset EV9). Overall, the impact of Filter-2 varied considerably between experimental samples with an average reduction of 24% but a range of 7–43% (Fig. 2B; Dataset EV9). We also examined the extent of overlap in the proteins identified by the three different baits in Exp 1 (Fig. 6H,I). As might be expected, proteins detected by all three baits contained the highest proportion of likely non-bait-specific targets (i.e., red- and orange-labeled proteins), whereas proteins unique to each bait contained the fewest. Specifically, of the 153 proteins common to all three strains based on Filter-1, fewer than half (75) remained after the Filter-2 step (Fig. 6G–I). The remaining 75 proteins include the NEKL–MLTs themselves along with candidate shared interactors. In addition, this category may contain non-bait-specific positives that our controls failed to detect.

With respect to our filtering strategy, it may be argued that a more direct approach would be to take individual intensity values from bait samples, divide by their corresponding mNG::TurboID values, and establish a cutoff based on this ratio. One complication with this strategy is that our control mNG::TurboID was expressed at substantially higher levels than our baits, potentially resulting in authentic targets being present at similar or even lower levels in the bait samples than in the mNG::TurboID control. Consistent with this, Fig. EV7A–C shows examples of outcomes from Exp 2, Exp 6, and Exp 8, including volcano plots in which the number of enriched proteins was much greater in the mNG::TurboID samples than in the bait samples. Despite this "reversed asymmetry", we were generally able to identify several true positives, including the baits and some of their immediate NEKL–MLT binding partners (Fig. EV7A–C). However, we also made the unexpected observation that numerous proteins were present at substantially reduced levels in the mNG::TurboID samples as compared with either the bait or N2 samples (e.g., SODH-1 and ACS-2; Fig. EV7A). This anomaly resulted in many proteins that appeared to be up in the bait samples when compared with mNG::TurboID but were not enriched in the bait samples relative to N2. Although we have no immediate explanation for this finding, it is possible that differences in strain backgrounds may have contributed to this outcome. Alternatively, high levels of biotinylated proteins in the mNG::TurboID lysate could have led to biotinylated proteins outcompeting ('sticky') non-biotinylated proteins for binding sites on the surface of streptavidin beads.

As a final check of our Filter-2 datasets, we determined whether the identified proteins are likely to be expressed in the *C. elegans* epidermis, as would be expected for authentic targets of the NEKLs–MLTs. To identify epidermally expressed proteins, we made use of two epidermal proteomic datasets, including prior findings using the $P_{dpy-7}$::mNG::TurboID strain, along with two epidermal transcriptomic datasets that collectively examined both larval and adult stages (Dataset EV9) (Chikina et al, 2009; Kaletsky et al, 2018; Katsanos et al, 2021; Reinke et al, 2017). Altogether, the combined "epidermal dataset" included 11,630 unique proteins or ~58% of annotated *C. elegans* genes. Encouragingly, 93–96% of the Filter-2 proteins identified by baits in Exp 1 overlapped with the epidermal dataset (Epidermal %), with a range of 73–98% across all experimental samples (Fig. 2B). Overall, 86% of the total 1868 Filter-2 proteins identified were present in at least two of the epidermal expression studies (average 2.59), indicating that a large majority of identified proteins are indeed present in the epidermis (Fig. EV9 and Dataset EV14). We also tested for a correlation between Epidermal % and the experimental metrics described above, including Carboxylase %, Up %, and TurboID %. Notably, Epidermal % correlated significantly with all three metrics (Fig. 4F) and may thereby provide an additional indicator of a positive outcome in situations where tissue-specific expression is available and relevant. This also suggests that a low Epidermal % may be caused in part by the non-specific binding of lysate proteins to streptavidin beads, which, as discussed above, may strongly impact the other metrics as well.

## Comparisons among biological replicates

Our collective experiments provided us with an opportunity to compare datasets from multiple (Filter-2) biological replicates, including five independent experiments for NEKL-2::TurboID and NEKL-3::TurboID, four for MLT-3::TurboID, and three for MLT-2::TurboID and MLT-4::TurboID. Rather strikingly, only a tiny fraction of proteins was enriched in all replicates for any given bait strain (Fig. 7A,B; Dataset EV12). For example, only three proteins were detected in all five NEKL-2::TurboID replicates, and only four proteins were detected in all five NEKL-3::TurboID replicates (Fig. 7A). Moreover, of the 1063 unique proteins identified for NEKL-3::TurboID, 856 (81%) were detected in only one of the five experiments. Likewise, of the 904 unique proteins identified for NEKL-2::TurboID, 684 (76%) were detected in just a single experiment. Nevertheless, we noted somewhat greater overlap between certain biological replicates, such as between Exp 1—Exp 3 and Exp 2—Exp 6 for NEKL-2 and NEKL-3, and between Exp 1—Exp 7 for MLT-3 (Fig. 7A; Dataset EV12).

As anticipated, proteins found in most or all biological replicates included the baits and their subcomplex partners. For example, both NEKL-3 and MLT-3 were identified in all NEKL-3::TurboID and MLT-3::TurboID samples (Fig. 7B). Likewise, NEKL-2 and MLT-2 were identified in all five NEKL-2::AID samples, with MLT-4 being identified in four of five samples. Somewhat surprisingly, despite being present at relatively low levels, MLT-2 showed the highest frequency of detection (15 of 20 samples), whereas NEKL-3 was detected in 10 of 20 samples (Fig. 7B; Datasets EV12 and 14). The high frequency of MLT-2 detection may be attributed to MLT-2 having direct physical interactions with three of the five

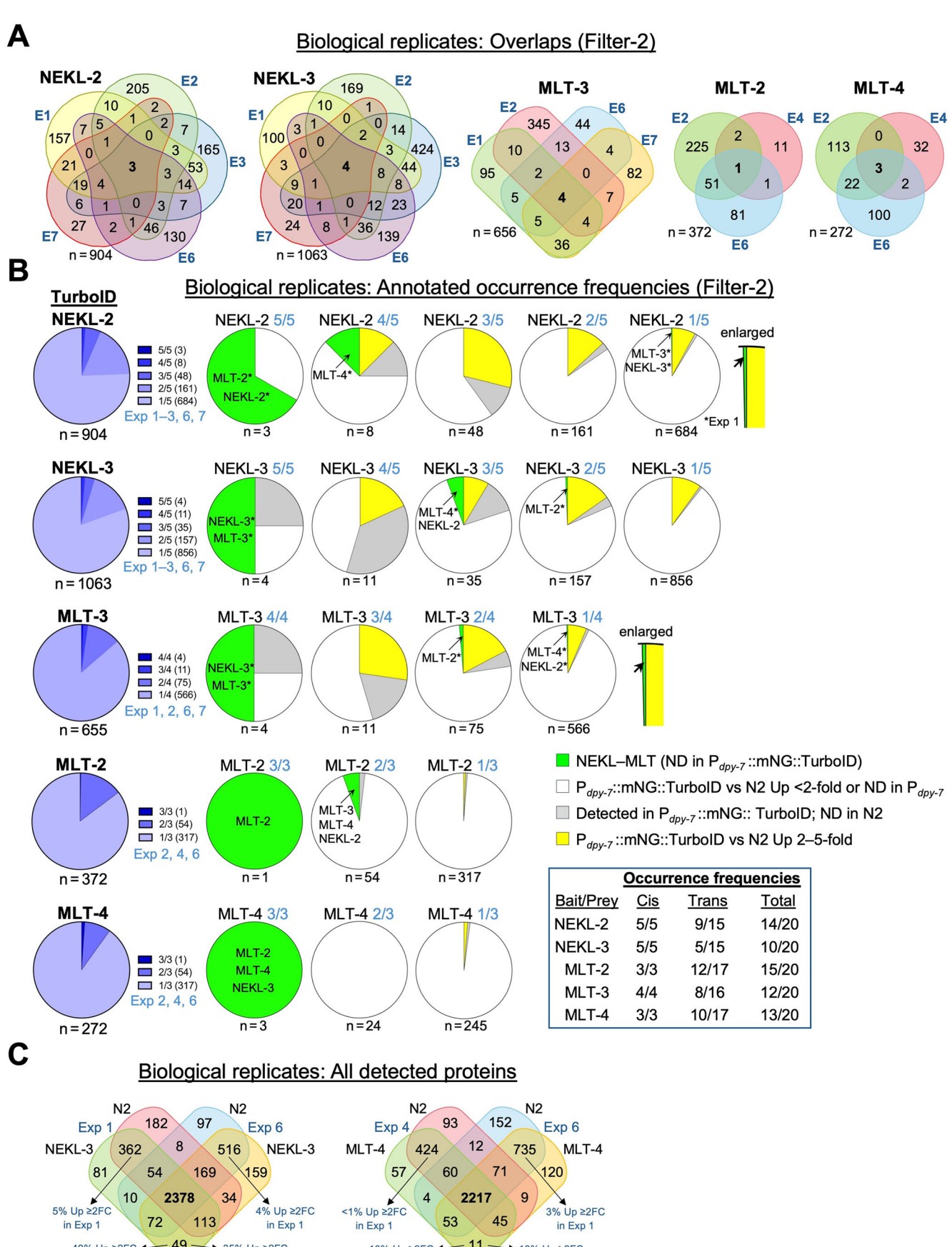

**A** Biological replicates: Overlaps (Filter-2)

**B** Biological replicates: Annotated occurrence frequencies (Filter-2)

**C** Biological replicates: All detected proteins

◀ **Figure 7.   Analysis of biological replicates.**

(A) Venn diagrams of biological replicates for the indicated baits (Filter-2 datasets). E experiment, *n* total number of unique proteins (for A–C). (B) (left column) Pie charts summarizing the proportion of proteins detected at the indicated frequencies for each bait–TurboID biological replicate. (center and right) Pie charts showing the proportion of annotated proteins at each frequency of occurrence for the indicated baits. Asterisks indicate that the NEKL–MLT protein was detected in Exp 1. Box below the key definitions summarizes the cis and trans occurrence frequencies for each NEKL–MLT protein. ND not detected. (C) Venn diagram of all proteins identified (irrespective of fold changes) in three of three replicates for the indicated experiments. Note sizeable overlaps between different strains from the same experiment, which are indicative of batch effects. Up % from select overlaps is indicated with arrows. FC fold change. Raw data are available in Dataset EV12. Source data are available online for this figure.

NEKL–MLT complex members (Fig. EV1) (Hoff et al, 2013; Leettola et al, 2014).

From the above, it might be inferred that a high frequency of occurrence is suggestive of an authentic physiological interaction. However, repeated identifications could also be consistent with non-bait-specific targets or other systematic false positives. Consistent with this, we found that combined yellow- and gray-labeled proteins (Fig. 6G) made up 39% and 28% of the proteins identified in three or more of the five biological replicates from NEKL-2::TurboID and NEKL-3::TurboID, respectively (Dataset EV12). In contrast, these classes made up only 10–12% of the proteins identified in just one or two of the replicates (Dataset EV12). Collectively, our results indicate that frequently occurring proteins are likely to include both authentic targets as well as false positives.

Conversely, it may be tempting to assume that proteins detected in only a minority of biological replicates are unlikely to be authentic targets. Our data, however, argue against this simplistic conclusion. As seen in Fig. 7B, we observed several cases in which NEKL–MLT proteins were detected in only one or two of the biological replicates. Moreover, in many cases, their identification(s) came from Exp 1, which ranked highly with respect to the quality metrics discussed above (Figs. 2B, 3A, and 7B). It may therefore be argued that more weight should be placed on the results of individual experiments that are objectively determined to be of higher quality or greater sensitivity. In practice, it will likely be necessary to apply a combination of the above principles to best evaluate any given dataset.

We also examined the extent of overlap in the full spectrum of proteins detected in the different biological replicates, irrespective of fold changes. Figure 7C (left) shows a Venn diagram of proteins detected from datasets of N2 and NEKL-3::TurboID in Exp 1 and Exp 6. Whereas 2378 proteins were common to all four experiments, 10% (362/3512) were specific to both strains in Exp 1 and 15% (562/3659) were specific to both strains in Exp 6. In contrast, only 49 proteins were specific to the NEKL-3::TurboID strain in both experiments (Fig. 7C). Likewise, when comparing N2 and MLT-4::TurboID datasets from Exp 4 and Exp 6, 14% (424/3056) and 21% (735/3489) were specific to the biological replicates, respectively (Fig. 7C). These findings are consistent with batch effects, which may be due to experimental factors including biases introduced by individual LC-MS/MS experiments (Schneider et al, 1987). As shown in Fig. 7C, however, batch effects may have a limited impact on the detection of proteins exhibiting enrichment in the bait samples. For example, only ~1–5% of the proteins associated with batch bias were up >twofold in the bait samples. In contrast, 18–49% of the proteins in the overlap between baits showed >twofold enrichment (Fig. 7C). We conclude that although batch effects likely influence the spectrum of detected proteins, they

may have a relatively minor impact on the identification of enriched targets.

## Variations on the experimental approach

Here, we discuss several experimental variations and their impacts on our studies. These included changing the location of the TurboID insertion site, depleting endogenous carboxylases to boost bait target signals, and piloting the combined use of the auxin-inducible-degron (AID) system in conjunction with TurboID. Although these can all be categorized as pilot studies, their outcomes may be informative to those designing their own proximity labeling experiments.

### N-terminal–TurboID fusions

One caveat associated with proximity labeling is the limited ability to detect distal members of a protein complex due to the relatively small biotin-labeling radius of TurboID (10–35 nm) (Cho et al, 2020b; May et al, 2020; Roux et al, 2012; Sears et al, 2019; Wu et al, 2024; Yang et al, 2024). It therefore seemed plausible that we might identify somewhat different subsets of proximal targets by tagging NEKL–MLT proteins with TurboID on their N versus C terminus. In addition, large tags, such as TurboID, can affect the ability of a protein to bind certain partners through steric interference, potentially preventing their identification. We thus obtained endogenously tagged TurboID::NEKL-2, TurboID::NEKL-3, and TurboID::MLT-3 strains that included N-terminal HA-epitope tags along with linkers separating the TurboID from the bait protein (Fig. 3D). Western blot analysis detected bands of the correct size for all three baits, although some breakdown of products was observed for the TurboID::MLT-3 samples (Exp 8, Appendix Fig. S1).

We carried out a single proximity labeling experiment (Exp 8) with four technical replicates for each strain along with the N2 control. Despite encouraging metrics including a high Carboxylase % (21–24%), Up % (64–78%) and TurboID % (71–100%), along with asymmetric volcano plots, we observed very modest enrichments for both the bait proteins and their NEKL–MLT subcomplex partners (Figs. 2B, 3D, and EV2; Dataset EV8). This included the absence of NEKL-2 from the TurboID::NEKL-2 filtered datasets, as it failed to make our twofold cutoff (1.95; $P = 0.015$). Likewise, TurboID::NEKL-2 showed lower enrichment for MLT-2 than did any of the five NEKL-2::TurboID experiments and failed to identify MLT-4 in any of the technical replicates (Fig. 2B; Dataset EV8). The diminished enrichment observed for the N-terminal TurboID fusions could be due to a combination of factors, including reduced activity of the TurboID enzyme, altered protein interactions, and reduced protein stability. We conclude that, as might be expected, the location of the TurboID tag can have a major impact on the

success of a proximity labeling experiment, although it may be difficult to anticipate the optimal insertion site prior to carrying out pilot studies.

### Endogenous carboxylase depletion

Another concern with proximity labeling is the presence of endogenously biotinylated proteins, which, depending on their relative levels, could limit the detection of low-abundance biotinylated targets of the bait proteins. As shown by Artan and colleagues (2022), depletion of the four major biotinylated carboxylases in *C. elegans* (PYC-1, PCCA-1, MCCC-1, and POD-2) can enhance the detection of certain targets, leading to a more sensitive and comprehensive screen (Artan et al, 2022). Nevertheless, we note that the percentage of carboxylases rarely exceeded 25% of the total detected protein in our experiments, suggesting that their removal might produce limited benefits (Fig. 2B; Dataset EV9).

To assess the impact of carboxylase depletion, we crossed NEKL-2::TurboID and NEKL-3::TurboID into a strain (AX7884) containing endogenously tagged ($His_{10}$) versions of the four carboxylases (gift of the de Bono lab). After lysate preparation, we depleted the endogenous carboxylases by incubating the lysates for 1 h with nickel-coated (His-binding) beads, after which the supernatant was incubated with streptavidin beads for an additional hour using our established protocol (Fig. 3E). As expected, we observed a strong reduction in the carboxylase signal on western blots probed with streptavidin (Fig. EV8A,B), which was supported by the low Carboxylase % detected by LC-MS/MS (Fig. 2B). Despite moderate-to-high values for the Up % (55% and 60% for NEKL-2::TurboID and NEKL-3::TurboID, respectively) and TurboID % (80% and 98%, respectively), we observed relatively modest enrichments for the bait and prey NEKL–MLT proteins versus the carboxylase-depleted AX7884 control strain (Figs. 2B, 3E, and EV2; Dataset EV3). Nevertheless, it is notable that Exp 3 may have detected authentic NEKL interactors that were not consistently identified in our other experiments and did show the highest percentage of epidermally expressed proteins (97% and 98%, respectively, for NEKL-2::TurboID and NEKL-3::TurboID; Fig. 2B; also see Fig. 9A). Further studies are needed to clarify the advantages of the carboxylase depletion approach, which may vary depending on the properties of individual baits and other experimental variables.

### Auxin-inducible degradation of complex members

Our previous studies have made extensive use of the AID system to degrade NEKL-2::AID and NEKL-3::AID in adult worms. This ability to rapidly downregulate the NEKLs has been particularly advantageous because the NEKLs are essential for larval development, which limits the use of genetic mutants. We hypothesized that by combining the TurboID and AID approaches, it may be possible to investigate how the removal of a bait binding partner or regulator alters the interactome of a bait protein. We constructed three strains containing AID-tagged versions of NEKL-2 or NEKL-3 in combination with NEKL-2::TurboID, NEKL-3::TurboID, or MLT-3::TurboID. The addition of auxin led to the rapid depletion of AID-tagged proteins, as expected, but did not affect the expression of the bait–TurboID proteins, which appeared to retain biotinylation activity based on corresponding streptavidin blots (Fig. EV8C). Moreover, the presence of auxin in the medium did

not appear to strongly alter predictive parameters, such as the enrichment of the bait protein (Fig. 2B), supporting the feasibility of combining these methods. One potential caveat, however, is that biotinylated proteins may endure in the cell for extended periods of time, thereby obfuscating the immediate impact of removing AID-tagged proteins. Future studies will be necessary to determine the length of time required to clear biotinylated proteins, which is likely to vary considerably among individual proteins. Alternatively, use of the AID system in combination with an inducible or activatable biotinylation enzyme could provide a path forward for these types of studies.

## Gene ontology analysis

Although not without caveats, GO analysis can be a powerful tool for identifying over-represented classes of proteins within large datasets, thereby providing insights into their molecular, cellular, and biological functions (du Plessis et al, 2011; Gaudet and Dessimoz, 2017; Gene Ontology et al, 2023). Using The Gene Ontology Resource (https://geneontology.org), we identified over-represented GO-Slim terms within the general categories (termed aspects) of Molecular Function, Cellular Component, and Biological Process. GO analysis was performed on the 23 individual (Filter-2) bait datasets as well as on combined biological replicates for each bait, for both subcomplexes, and for the full combined dataset (Fig. 8A; Dataset EV13). We note that because standard GO pipelines do not recognize duplicate values, proteins present in only one biological replicate were weighted equally with those detected in multiple samples, a limitation that may lead to the underestimation of fold enrichments for certain protein categories.

As might be expected given the observed variability among the biological replicates (Fig. 7), their corresponding GOs also exhibited limited overlap (Fig. 8B; Datasets EV12 and 13). In the case of NEKL-2, the greatest overlap was detected between Exp 1 and Exp 3 and between Exp 2 and Exp 6, corresponding to greater overlap detected at the protein level between these studies (Figs. 7A and 8B; Dataset EV13). Likewise, MLT-3 showed the most extensive overlap between Exp 1 and Exp 7 at both the protein and GO levels (Fig. 7A; Datasets EV12 and 13). Other baits, however, showed less clear correspondence between their protein overlap and GO findings (Datasets EV12 and 13).

As a control, we used the combined mNG::TurboID dataset in which red-, orange-, and a subset of yellow-labeled proteins had been removed to generate a non-specific GO dataset analogous to those generated for our bait–TurboID fusions (Dataset EV13). We then identified overlaps between the NEKL-2, NEKL-3, and mNG::TurboID combined datasets (Fig. 8C; Dataset EV13). In the case of Molecular Function and Biological Process aspects, we observed greater overlap between the GOs for NEKL-2 and NEKL-3 (54% and 58%, respectively) than between the GOs for the NEKLs and mNG::TurboID (44–46%; Dataset EV13). In addition, we determined the correlation in the fold enrichment for GO terms between NEKL-2, NEKL-3, and mNG::TurboID. For all three GO aspects, we observed the strongest correlation between the NEKL-2 and NEKL-3 datasets (Figs. 8D and EV9; Dataset EV13). These findings suggest that NEKL-2::TurboID and NEKL-3::TurboID share GO profiles to a greater extent than each does with the non-specific mNG::TurboID control.

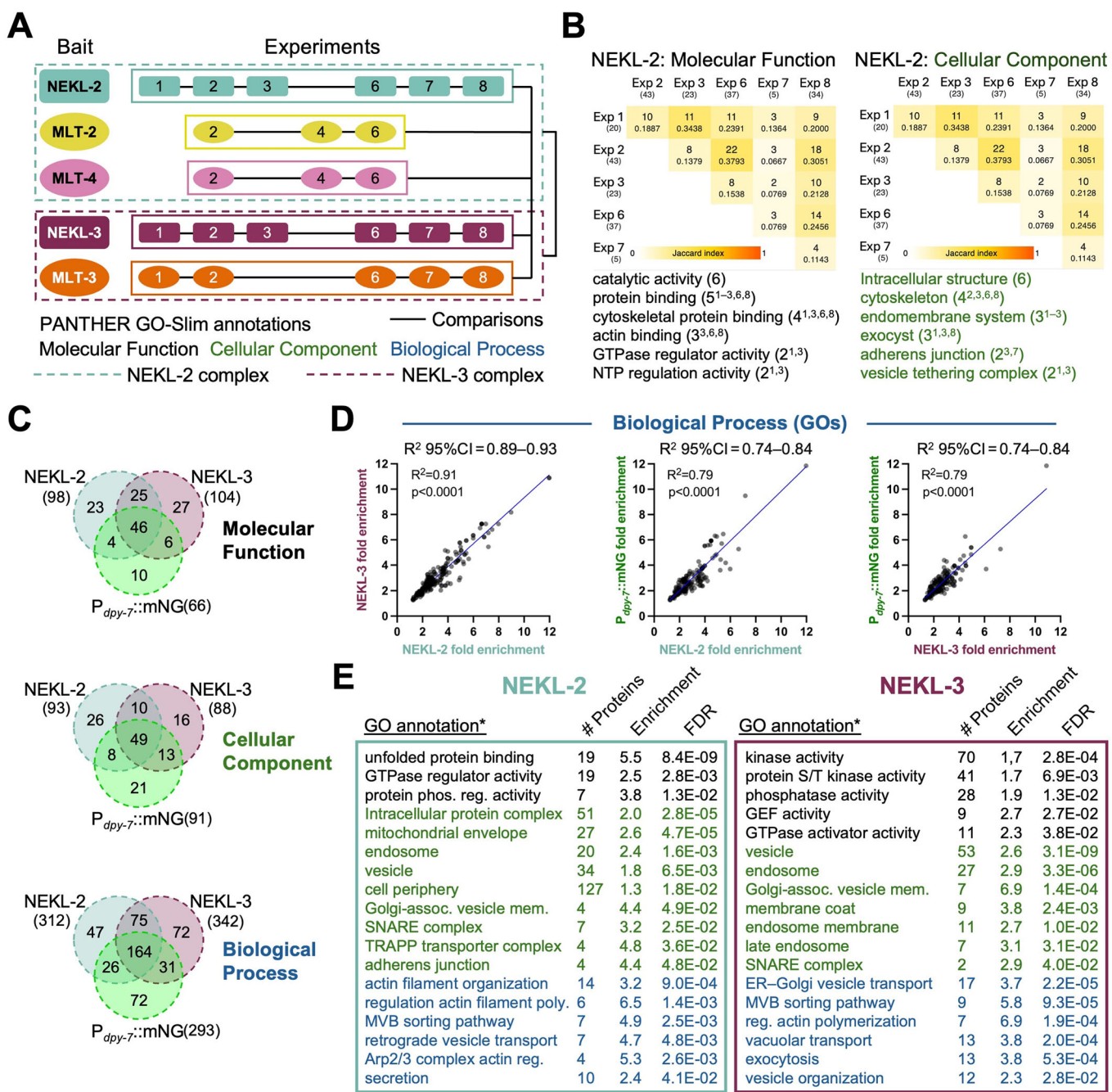

**Figure 8.   Results of gene ontology analyses.**

(A) Diagram of experiments for which gene ontology analysis was carried out using The Gene Ontology Resource (Slim subset). Solid- and dashed-line boxes indicate pooled datasets for the combined bait and subcomplex analyses, respectively. Solid black lines indicate the overlap comparisons carried out (Dataset EV13). (B) Comparison of overlaps in GOs between the six biological replicates of NEKL-2 for Molecular Function and Cellular Component aspects. Darker-shaded boxes indicate greater overlap (see Jaccard index); integers indicate the number of shared GO terms; decimals show the fraction of overlapping GOs between the replicates. Shown below are select GO terms with integers and superscripts indicating the number of experimental occurrences and the experiments in which they were identified, respectively. (C) Venn diagrams showing overlaps in GO terms from the combined NEKL-2::TurboID, NEKL-3::TurboID, and $P_{dpy-7}$::mNG::TurboID datasets for each aspect. Numbers in parentheses indicate the total number of GOs identified for each bait. (D) Scatterplots show correlations in the fold enrichment of Biological Process GOs for the combined NEKL-2::TurboID, NEKL-3::TurboID, and $P_{dpy-7}$::mNG::TurboID datasets. 95%CI, 95% confidence interval. (E) Select GO terms, color-coded by aspect, from combined NEKL-2::TurboID and combined NEKL-3::TurboID datasets that were absent from the combined $P_{dpy-7}$::mNG::TurboID GO dataset. The number of identified proteins matching the GO term, their fold enrichment in the dataset, and the false discovery rate (FDR) are shown for each term. Source data are available online for this figure.

We next identified and annotated GOs that were present in the combined bait–TurboID datasets but were absent from the mNG::TurboID dataset (Dataset EV13). Figure 8E shows a subset of such "bait-specific" GOs for NEKL-2 and NEKL-3, along with the number of corresponding proteins identified, the fold enrichment for GO terms (relative to a random dataset), and the estimated false discovery rate (FDR). Notably, of the 26 Cellular Component GOs assigned to NEKL-3 but absent from mNG::TurboID, 20 (77%) are directly linked to membrane trafficking (Fig. 8C,E; Dataset EV13). Moreover, of the 10 most well-supported Molecular Function GOs for NEKL-3 (based on the lowest FDRs), eight are associated with protein kinase activity (Dataset EV13). Likewise, the NEKL-2 bait-specific dataset contained GO terms linked to membrane trafficking and actin regulation (Fig. 8E, Dataset EV13). By contrast, the MLT–TurboID datasets showed more limited GO findings, although in the case of MLT-3 this included several GOs linked to membrane trafficking (Dataset EV13). Interestingly, MLT-3 GO analyses also indicated links to mRNA regulation and stability, consistent with several recent reports on the MLT-3 mammalian ortholog, ANKS3 (Estrada Mallarino et al, 2020; Mahuzier et al, 2024; Rothe et al, 2023; Wei et al, 2025). Overall, our GO analysis is consistent with our prior studies showing roles for the NEKLs in membrane trafficking and cytoskeletal regulation and may shed light on additional NEKL–MLT functions.

## Functional links between NEKL–MLTs and their proximity labeling datasets

In a parallel approach to our GO analysis, we manually inspected datasets for connections to proteins and pathways already linked to the NEKLs and MLTs through previous studies. Notably, proximity labeling identified six of the 18 known genetic suppressors of *nekl* molting defects in at least two independent experimental samples (range, 2–11). This included PIKI-1, a phosphatidyl inositol kinase identified in four experimental samples, and TAT-1, a membrane lipid flippase identified in two experimental samples (Fig. 9A; Dataset EV14) (Milne et al, 2025; Reimann et al, 2025). Like NEKL-2, PIKI-1 and TAT-1 both localize, in part, to early endosomes and were each identified in two of the NEKL-2::TurboID experiments. Another *nekl* suppressor, FCHO-1, is an F-BAR-containing membrane-binding/bending protein that regulates the activity of the AP2 clathrin-adapter complex (Hollopeter et al, 2014; Joseph et al, 2020). FCHO-1 was enriched 9.1-fold in NEKL-3::TurboID (Exp 3) and 2.8-fold in NEKL-2::TurboID (Exp 3; Fig. 9A; Dataset EV14). RME-1 (human EHD1), a *nekl* suppressor with roles in vesicle scission and endosomal recycling, was also identified in two experimental samples (Dataset EV14) (Grant and Caplan, 2008; Milne et al, 2025). However, it is worth noting that RME-1 was significantly enriched (adjusted $p < 0.05$) in seven additional experimental samples (~1.4- to 1.9-fold) but failed to make the twofold (Filter-1) cutoff (Datasets EV1–4, 6–8). DNBP-1, identified in six experimental samples, is the *C. elegans* ortholog of human dynamin binding protein (DNMBP/TUBA) and is predicted to function as an activator of the small GTPase and actin polymerization regulator CDC-42 (Fig. 9A; Dataset EV14) (Cestra et al, 2005; Salazar et al, 2003). Moreover, we have shown that inhibition of DNBP-1, CDC-42, and the CDC42-activated kinase ortholog SID-3 lead to the suppression of *nekl* defects (Binti et al,

2024b; Lazetic et al, 2018). Collectively, these findings indicate that our proximity labeling approach was successful in identifying proteins relevant to NEKL–MLT functions.

Previous studies by our lab and others have shown that members of the NEKL-2 subcomplex localize, in part, to cell junctions along the apical-lateral surface adjoining the epidermal hyp7 and seam-cell syncytia (Beyrent et al, 2024; Castiglioni et al, 2020; Lazetic and Fay, 2017a; Lazetic et al, 2018; Riga et al, 2021). Moreover, junctional localization of NEKL-2 is dependent on MLT-2 and MLT-4 (Lazetic and Fay, 2017a). Epithelial junctional complexes include proteins from the cadherin and catenin families, several of which were identified by our studies, including HMR-1/cadherin (seven samples), HMP-1/alpha catenin (four samples), and JAC-1/delta catenin (four samples; Fig. 9A). Moreover, 11 of the 15 experimental samples in which junctional proteins were identified involved members of the NEKL-2 complex, including MLT-2. Although the function of the NEKL-2 complex at epithelial junctions remains unclear, our proximity labeling results, together with previous co-localization studies, collectively suggest a role in cell adhesion.

Other proteins identified by our study include UNC-116, the *C. elegans* kinesin-1 heavy chain ortholog, which functions in microtubule-based motor transport, and UNC-16 (human MAPK8IP3), a cargo adapter required for protein trafficking within neurons (Byrd et al, 2001; Patel et al, 1993). UNC-16 and UNC-116 have been reported to control the transport and localization of NEKL-3 during axon extension in several neurons that lie in close juxtaposition to the epidermis (Drozd and Quinn, 2023). As for RME-1, it is worth noting that whereas UNC-16 was detected in only a single experiment of NEKL-3::TurboID (Exp 1, Filter-2), it was up 1.99 fold in NEKL-3 Exp 3 ($P = 0.011$), and both UNC-16 and UNC-116 were statistically enriched in several other experimental samples but failed to make the twofold cutoff. These observations highlight that while establishing cutoffs is essential for prioritizing results and minimizing false positives, it may also inadvertently lead to false negatives.

A final proximal interactor worth mentioning is PLK-1, the *C. elegans* ortholog of mammalian polo-like kinase 1. PLK-1 was identified in eight experiments, six of which were linked to the NEKL-2 subcomplex (Fig. 9A). Studies in mammalian cells indicate that PLK1 phosphorylates NEK9, a mammalian ortholog of NEKL-2, to control centrosome separation (Bertran et al, 2011). Although our prior studies have not implicated the NEKLs in cell cycle–related activities (Yochem et al, 2015), it remains possible that *C. elegans* PLK-1 acts through NEKLs to regulate microtubule-based processes that are not associated with the cell cycle. Consistent with this, GO analysis for the NEKL–MLTs identified terms associated with microtubule function (Dataset EV13).

## Identification of new *nekl–mlt* genetic enhancers

Beyond identifying links to established NEKL–MLT interactors, we were curious whether our proximity labeling studies could be used to identify new functional interactors. To this end we genetically tested several of the identified proteins that were detected at high frequencies and had molecular identities consistent with known NEKL functions (Fig. 9A). This included DDL-2 (human WASH2P/WASHC1), a component of the *C. elegans* WASH complex, which acts at the intersection of membrane trafficking

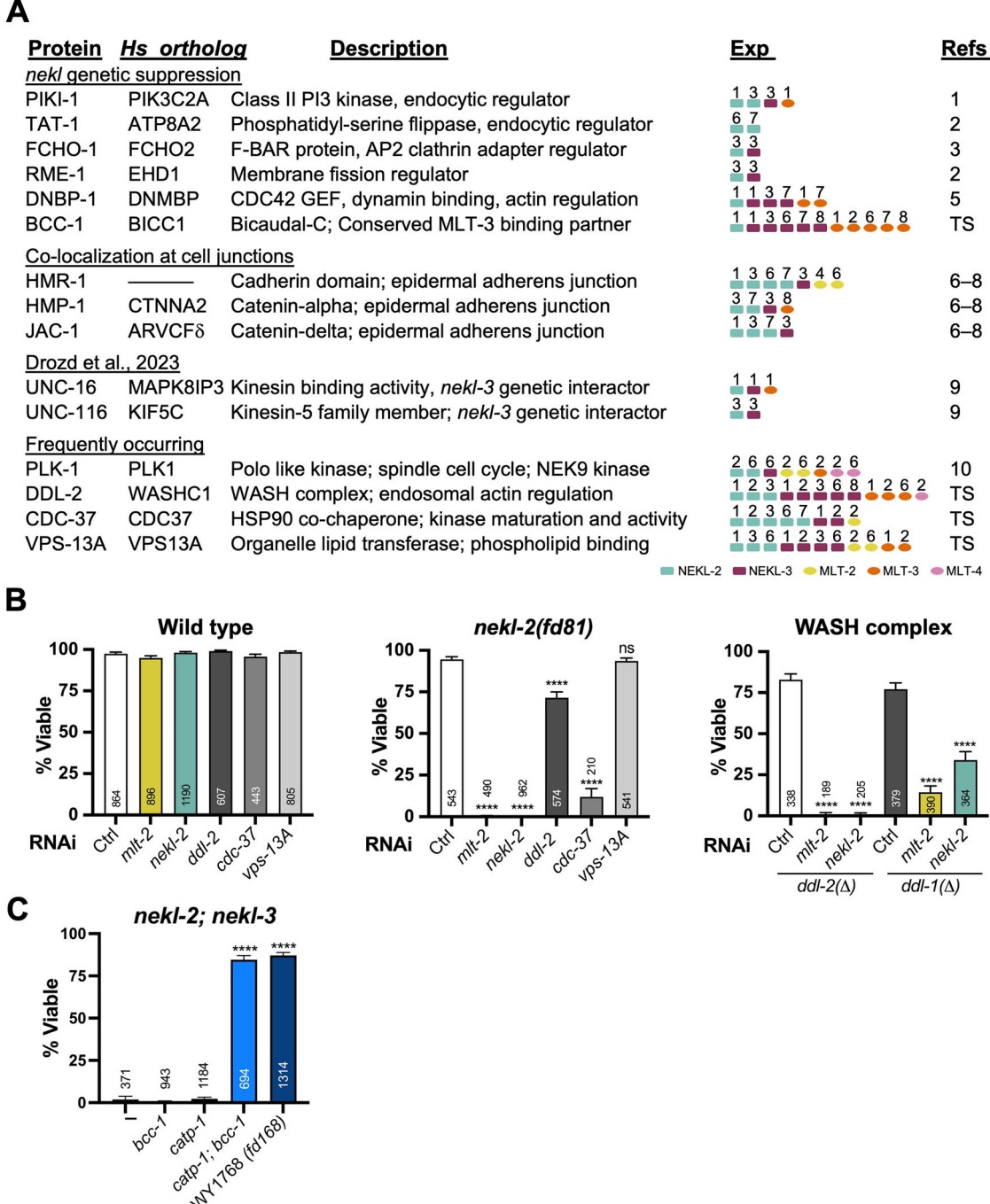

**Figure 9. Functional relevance of select proximity labeling targets.**

(A) List of NEKL–MLT interactors grouped by *nekl* genetic suppression, co-localization to cell junctions, reported interaction (Drozd and Quinn, 2023), and frequency of occurrence in the dataset. Detected experiments are indicated along with the following reference PMID numbers: (1) 40475523; (2) 39722491; (3) 32069276; (4) *to be added*; (5) 39173071; (6) 27799278; (7) 39110529; (8) 33300872; (9) 37756604; (10) 21642957. (B) Genetic enhancer assays using RNAi feeding in the indicated backgrounds. % Viable refers to the percentage of animals that reached adulthood. *ddl-1(ok2916)* and *ddl-2(ok3235)* deletion alleles were used for the analysis. Ctrl, control [*GFP(RNAi)*]. (C) Genetic suppression of *nekl-2(fd81); nekl-3(gk894345)* double mutant larval arrest with CRISPR-generated *bcc-1(fd450)* [G32S] and *catp-1(fd429)* [intron-15 splice donor (GT > AT)]. WY1768 was isolated from a *nekl-2(fd81); nekl-3(gk894345)* suppressor screen and backcrossed 5x before sequencing without sibling subtraction (Binti et al, 2024a; Joseph et al, 2018). For (B, C), error bars indicate 95% CI; *P* values were calculated using a Fisher's exact test; ****$P \leq 0.0001$. Source data are available online for this figure.

and endosomal actin regulation (Fokin and Gautreau, 2021; Simonetti and Cullen, 2019). Notably, DDL-2 was identified in 12 separate samples. VPS-13A, a protein that controls the bulk transport of lipids between organelles, was likewise identified in 11 experiments (Dziurdzik and Conibear, 2021; Kumar et al, 2018). We note that although VPS-13A expression was not detected in the epidermal datasets we tested, two other reports have suggested expression in the epidermis (McKay et al, 2003; Mounsey et al, 2002). Lastly, CDC-37, an established chaperone required for the proper folding and activity of protein kinases (Caplan et al, 2007; Verba and Agard, 2017), was identified in eight experiments, including enrichment in all five NEKL-2::TurboID experiments, where it was up 5.5- to 81-fold.

For our functional assay, we used a previously described RNAi-based enhancer approach that makes use of a CRISPR-generated aphenotypic weak reduction-of-function mutation in *nekl-2* (*fd81*) (Lazetic and Fay, 2017a). As expected, our positive RNAi feeding controls, which lead to a partial knockdown of *nekl-2* or *mlt-2*, resulted in a strong loss of viability in the *nekl-2(fd81)* background but showed little or no effect on wild-type (N2; Fig. 9B). Notably, we observed a strong reduction in viability when *cdc-37* was partially depleted by RNAi feeding in *nekl-2(fd81)* mutants but not in wild-type (Fig. 9B). Namely, ~90% of *nekl-2(fd81); cdc-37(RNAi)* worms arrested with molting defects versus <10% in our control RNAi experiments (Fig. 9B). These findings, in conjunction with our proximity labeling data, indicate that CDC-37 likely functions as a molecular chaperone for NEKL-2. In contrast, no enhancement effects were detected in parallel tests with *vsp-13A(RNAi)* (Fig. 9B).

We also observed a partial but significant decrease in the viability in *nekl-2(fd81); ddl-2(RNAi)* worms as compared with *nekl-2(fd81); control(RNAi)* as well as RNAi of *ddl-2* in the N2 background. To further explore the WASH interaction, we obtained viable deletion alleles of *ddl-2* (*ok3235*) and its complex partner, *ddl-1* (*ok2916*; human WASHC3). Whereas both deletion mutants were ~80% viable on control RNAi feeding plates, *ddl-2* mutants exhibited 100% arrest after weak knockdown of *nekl-2* or *mlt-2* by RNAi (Fig. 9B). Likewise, *ddl-1* mutants exhibited moderate-to-high levels of genetic enhancement with *nekl-2* and *mlt-2* RNAi (Fig. 9B). These results suggest that the NEKLs may function in parallel to the WASH complex to regulate actin dynamics within the endomembrane system.

## Identification of a new *nekl–mlt* genetic suppressor

Using a forward-genetic approach, we recently reported that a reduction-of-function mutation affecting the CATP-1 Na$^+$/K$^+$ cation pump (Q905Stop) was sufficient to suppress molting defects in *nekl-2(fd81); nekl-3(gk894345)* synthetically lethal mutants (Binti et al, 2024a). Moreover, whole-genome sequencing of two independently isolated suppressor strains revealed mutations affecting the *catp-1* locus [WY1744, R668C; WY1768, intron-15 splice donor (GT > AT)]. Surprisingly, neither *catp-1* mutation, when introduced into *nekl-2; nekl-3* mutants by CRISPR/Cas9, conferred genetic suppression (Fig. 9C) (Binti et al, 2024a). We hypothesized that these mutations were either non-causative or that additional mutations in these strains may cooperate with *catp-1* to promote genetic suppression.

Intriguingly, BCC-1, which was enriched in 11 experimental samples, including all five MLT-3::TurboID replicates, was identified

as a candidate causal mutation in suppressor strain WY1768 (Fig. 9A; Dataset EV14) (Binti et al, 2024a). The WY1768 mutation is predicted to alter a conserved glycine residue (G32S) within the first K-homology domain of BCC-1. We found that whereas neither *bcc-1*[G32S] nor *catp-1*[GT > AT] CRISPR-generated mutations conferred genetic suppression singly, when combined the two mutations suppressed molting defects in *nekl-2; nekl-3* mutants to levels similar to those observed in strain WY1768 (Fig. 9C). Notably, human BCC1 binds to the MLT-3 ortholog, ANKS3, and may also form a complex with the MLT-2 ortholog, ANKS6, suggesting that these physical and functional interactions are highly conserved (Rothe et al, 2023; Rothe et al, 2018). More broadly, our studies demonstrate the utility of genetic assays for establishing functional links between bait proteins and their proximal interactors.

## Closing thoughts and considerations

In this study, we have sought to outline an intuitive framework for designing proximity labeling experiments, as well as analyzing and interpreting the generated data. This includes simple metrics to aid in the evaluation of experimental outcomes, which may be particularly useful if little is known about the partners of the bait protein(s). Our approach to filtering datasets is generalizable and flexible, in that different cutoffs can be used to either increase the retention of authentic positives or reduce the occurrence of false positives. As with all filtering strategies, a balance should be struck that best serves the intended goals of the study.

Consistent with other studies, our findings argue for the incorporation of internal TurboID controls, in particular, the use of isogenic strains in which the TurboID control is expressed at similar levels to the baits of interest (Artan et al, 2021; Artan and de Bono, 2022; Jiang et al, 2025; Grismer et al, 2024; Hollstein et al, 2022; Holzer et al, 2022; Moreira et al, 2023; Nikonorova et al, 2025). In fact, this latter point brings up one weakness of our study. However, as mentioned above, this added investment in time and resources may not always be feasible or necessary to achieve experimental goals.

Another key conclusion from our study is that proximity labeling experiments often exhibit substantial variability, particularly among biological replicates. This underscores the importance of evaluating the overall quality of each experiment. For instance, based on our metrics, Exp 2 and Exp 6 appear to be of lower quality than other biological replicates. Consequently, it may be appropriate to assign less weight to the results from these experiments, including positive and negative findings. An additional source of variability in our experiments may have stemmed from the use of mixed-stage populations of *C. elegans*. On the one hand, the use of mixed stages enables a researcher to cast a wide net, which may aid in the comprehensive identification of bait-interacting proteins. Fluctuations in the distribution of life-cycle stages, however, may lead to increased variability between biological replicates. Furthermore, the use of mixed-stage populations may limit the ability to detect stage-specific interactors. As such, the decision to use synchronized versus mixed-stage populations should be made based on knowledge of the bait protein, including its developmental expression profile and stage-specific functions. We note that in the case of the NEKL–MLTs, consistent expression from late embryogenesis through adulthood informed our decision to use mixed-stage populations for our experiments.

It was also notable that some apparent NEKL interactors were consistently detected, whereas their known immediate binding partners were not. For example, HSP-90/DAF-21, the binding partner of CDC-37, was detected in only two experiments (at just above 2-fold), whereas CDC-37 was detected in eight experiments at much higher levels (Caplan et al, 2007; Stepanova et al, 1996). Likewise, DDL-2 was detected in 12 samples, whereas DDL-1 was not detected in any of our experiments. Such discrepancies may be due to differences in distances between the bait protein and its targets and may be influenced by the size of the target protein, along with the availability of exposed lysine residues on the target. Regardless, such discrepancies point out that patterns of detection may be patchy and unpredictable.

Our studies also highlight the value of DIA–MS in proximity labeling workflows, a methodology that—as noted in the introduction—has not yet seen widespread use in this context. Additionally, our data suggest that using as few as three technical replicates may be adequate, offering a cost-effective strategy for future studies. One notable caveat of our study, however, is that we did not carry out a direct comparison to test the effectiveness of DIA versus DDA–MS approaches, although such comparisons have been carried out using other experimental systems (Fernandez-Costa et al, 2020; Frohlich et al, 2022; Kitata et al, 2023; Li et al, 2020; Lou and Shui, 2024; Muller et al, 2019). In addition, our study was not designed or optimized for the detection of site-specific target phosphorylation events, which typically requires a phospho-enrichment step prior to MS (Qiu et al, 2020). Nevertheless, several recent reports indicate that DIA is applicable and effective for use in phospho-proteomic analyses (Bekker-Jensen et al, 2020; Di et al, 2023; Pham et al, 2024; Skowronek et al, 2022).

Notably, our experiments were less effective at identifying proximate interactors of MLT-2 and MLT-4 versus NEKL-3, MLT-3, and NEKL-2, which may be due to their lower levels of expression. Moreover, the NEKLs and MLTs are primarily expressed in the epidermis, which constitutes ~25% of the adult worm's volume (Froehlich et al, 2021; Lazetic and Fay, 2017a; Yochem et al, 2015). Consequently, bait proteins that are expressed in only a small fraction of cells and/or at very low levels may not be amenable to proximity labeling approaches, particularly if the bait–fusions are expressed at endogenous levels. This caveat could potentially be addressed by tagging bait proteins at both ends with TurboID or by tagging multiple members of a protein complex. Alternatively, overexpression of a bait protein may yield useful physiologically relevant findings, although the frequency of over-expression artifacts will likely be increased.

Lastly, we would emphasize that our goal in undertaking proximity labeling experiments was not to generate lists of proteins or a set of GO terms, although these may prove useful to others in the field. Rather, we are interested in identifying interaction partners and phosphorylation targets of the NEKL–MLTs (Bonham et al, 2023; Cabral et al, 2024; Guo et al, 2023; Kim et al, 2023; Matsuhisa et al, 2025; Niinae et al, 2021; Zhang et al, 2022). To this end, we have undertaken large-scale protein modeling and genetic interaction studies between the NEKL–MLTs and the candidate proteins identified by our studies. Future work will include biochemical and cell biological analyses on promising targets, with a particular focus on interactions that are likely to be conserved in mammals and may shed light on the less-explored roles of NIMA-related kinases in health and disease.

# Methods

### Reagents and tools table

| Reagent/resource | Reference or source | Identifier or catalog number |
|---|---|---|
| **Chemicals** | | |
| NaCl | Sigma | S7653-5KG |
| Sodium deoxycholate sulfate | Fisher Scientific | BP166-500 |
| Triton X-100 | RPI Research Products | 111036 |
| Tris-HCl, pH 8.0 | KD Medical | RGF-3360 |
| KCl | Fisher Scientific | P217-500 |
| $Na_2CO_3$ | Alfa Aesar | 11552 |
| Urea | EMD Milliport | 666122-500GM |
| Glycine | Sigma | G8898-500G |
| Auxin (indole-3-acetic acid) | Thermo Fisher Scientific | A10556-06 |
| Trizma base | Sigma | T6066-5KG |
| Potassium Phosphate Monobasic | Fisher Scientific | P380-212 |
| Sodium Phosphate Monobasic Anhydrous | Fisher Scientific | BP329-1 |
| Magnesium sulfate heptahydrate | Sigma | 230391-2.5KG |
| Tween 20 | Fisher Scientific | BP337-500 |
| **Reagents** | | |
| Halt Protease Phosphatase Inhibitor Cocktail | Thermo Fisher Scientific | 78442 |
| Qubit Protein Assay | Thermo Fisher Scientific | Q33212 |
| Dynabeads MyOne Streptavidin C1 beads | Thermo Fisher Scientific | 65001 |
| Immun-Blot PVFD Membrane For Protein Blotting | Bio-Rad | 1620174 |
| PBS 10X pH7.4 | KD Medical | RGF-3210 |
| 6× Laemmli SDS sample | Alfa Aesar | J61337 |
| EveryBlot Blocking Buffer | Bio-Rad | 12010020 |
| Dynabeads His-Tag Isolation and Pull-Down | Thermo Fisher Scientific | 10103D |
| SuperSignal West Pico PLUS Chemiluminescent Substrate | Thermo Fisher Scientific | 34580 |
| Restore PLUS Western Blot Stripping Buffer | Thermo Fisher Scientific | 46430 |
| Mini-PROTEAN TGX Gels (4-15%) | Bio-Rad | 4561084 |
| **Antibodies** | | |
| Streptavidin-HRP | Cell Signaling Technology | 3999S |
| β-Actin (13E5)-HRP | Cell Signaling Technology | 5125S |
| DYKDDDDK [FLAG-TAG] Antibody-HRP | GenScript | A01428 |
| **Software** | | |
| AlphaFold3 | Google | |
| ChimeraX 1.9 and AFM-LIS software | UCFS | |

| Reagent/resource | Reference or source | Identifier or catalog number |
| --- | --- | --- |
| GraphPad | Prism | |
| Excel | Microsoft | |
| **CRISPR** | | |
| Alt-R™ CRISPR-Cas9 tracrRNA, 5 nmol | IDT | 1072532 |
| Alt-R™ S.p. Cas9 Nuclease V3, 100 µg | IDT | 10008100 |
| **Other** | | |
| Huanyu MT-13K-L Mini Handheld Homogenizer | Amazon | B01LYN90BU |
| Qubit 4 Fluorometer | Thermo Fisher Scientific | Q33238 |
| MagneSphere® Technology Magnetic Separation Stands | Promega | Z5332 |
| Mini Trans-Blot Cell | Bio-Rad | 1703930 |

## Methods and protocols

### C. elegans strains

TurboID strains were designed by DSF and generated by SunyBiotech; sequence information is provided in Appendix Supplementary Methods S1. NEKL::AID and mutant strains were generated as described (Joseph et al, 2020; Lazetic and Fay, 2017a). ddl-1(ok2916) and ddl-2(ok3235) were provided by the Caenorhabditis Genetics Center (CGC) and generated by the C. elegans Gene Knockout Consortium (Consortium, 2012). AX7623 (dbIs33[P$_{dpy-7}$::TurboID::mNeonGreen::3XFLAG] was provided by M. de Bono and colleagues (Artan et al, 2021).

### Protein lysate preparation

Mixed-stage worms were grown at 25 °C on NGM plates with E. coli (OP50) using standard conditions (Branon et al, 2018; Stiernagle, 2006). Worms were collected using M9 buffer and washed twice with RIPA lysis buffer (150 mM NaCl; 0.2% SDS [wt/vol]; 0.5% sodium deoxycholate [wt/vol]; 1% Triton X-100 [vol/vol]; and 50 mM Tris-HCl, pH 8.0). Prior to lysis, Halt Protease and Phosphatase Inhibitor Cocktail (100×) (Thermo Fisher Scientific, Cat# 78442) was added to each sample to a final concentration of 1× (Binti et al, 2024b). Worm lysis was performed on ice using a handheld homogenizer (Huanyu MT-13K-L Mini Handheld Homogenizer) for 4 min. The lysates were then centrifuged at 10,000 rpm (9391 RCF) for 6 min at 4 °C. The resulting supernatants were transferred to fresh microfuge tubes, and the protein concentration of each was measured using the Qubit 4 Fluorometer (Thermo Fisher Scientific).

### Binding time optimization

Dynabeads MyOne Streptavidin C1 beads (150 µl; Thermo Fisher Scientific, Cat# 65001) were equilibrated once with RIPA lysis buffer for 2 min, and the supernatant was removed using a magnetic stand. To determine the optimal incubation time for efficient binding of biotinylated proteins, 3 mg of lysate from NEKL-3::TurboID was added to the beads, followed by gentle rotation using a rotator. Samples were incubated at room temperature for 1 h and 2 h, as well as at 4 °C for 16 h. After incubation, the beads were separated from the lysate (flow-through) using the magnetic stand. Flow-through fractions were collected and analyzed by western blotting to assess the minimum incubation time required for streptavidin beads to bind most biotinylated proteins (Fig. 1D).

### Sample preparation for LC-MS/MS

Based on the time-course experiment, a 1-h (room temperature) incubation was chosen for subsequent experiments. After bead equilibration and the 1-h incubation, extensive washing steps were carried out to eliminate non-specifically bound proteins. The washing protocol included the following sequential steps using ice-cold solutions: one wash each with RIPA lysis buffer, 1 M KCl, 0.1 M Na$_2$CO$_3$; one wash with 2 M urea in 10 mM Tris-HCl (pH 8.0); and one wash with 4 M urea in 10 mM Tris-HCl (pH 8.0). This was followed by three washes with ice-cold RIPA lysis buffer and five washes with ice-cold 1× PBS (Sanchez and Feldman, 2021). (Sanchez and Feldman 2021). Finally, 100 µl of buffer containing 8 M urea in 50 mM Tris (pH 8.0) was added and thoroughly mixed. The beads were then snap-frozen and stored at –80 °C until further processing for western blotting and on-bead digestion, followed by LC-MS/MS.

### Western blotting

For western blots, 30 µg of protein lysate and 10 µl of eluate fractions were used. To elute the biotinylated proteins from the streptavidin beads, 6× Laemmli SDS sample buffer was added to 10 µl of beads (1× final concentration). The mixture was then heated at 95 °C for 10 min. After heating, the beads were pelleted using a magnetic stand, and the supernatant was collected for western blot analysis. Laemmli sample buffer was also added to the protein lysates and similarly heated.

Proteins were separated on a 5–15% gel (Bio-Rad Mini-PROTEAN TGX Precast Gels) and transferred to a PVDF membrane for 1 h at 100 V using a Mini Trans-Blot Cell (Bio-Rad). The membrane was then blocked with EveryBlot Blocking Buffer (Bio-Rad, Cat # 12010020) for 10 min at room temperature. The membrane was then incubated with horseradish peroxidase (HRP)-conjugated primary antibodies (see below) at a 1:1000 dilution for 1 h with gentle shaking and then was washed twice with 1× TBST (Tris-buffered saline with 0.1% Tween 20) buffer and once with 1× TBS (Tris-buffered saline) buffer, each for 5 min. Finally, the PVDF membrane was developed using a chemiluminescence substrate (SuperSignal West Pico PLUS Chemiluminescent Substrate, Cat# 34577) and imaged (Binti et al, 2024b). The resulting blot was imaged, and the membrane was then washed with 1× TBS buffer for 30 s and stripped using Restore PLUS Western Blot Stripping Buffer (Thermo Fisher Scientific, Cat# 46430) for 30 min at 37 °C with shaking. The stripping buffer was discarded, and the membrane was washed again with 1× TBS for 30 s, blocked again for 10 min at room temperature, and then incubated with subsequent antibodies. This process was repeated with antibodies against FLAG, actin, and streptavidin (in order) as needed. Antibodies used in this study were Streptavidin-HRP (Cell Signaling Technology, Cat# 3999S), β-Actin (13E5)-HRP (Cell Signaling Technology, Cat# 5125S), and THE DYKDDDDK [FLAG-TAG] Antibody-HRP (GenScript, Cat# A01428).

### His-Tag carboxylase depletion

Dynabeads His-Tag Isolation and Pull-Down beads (Thermo Fisher Scientific, Cat# 10103D) were equilibrated by washing once with 1× RIPA lysis buffer as described above. A total of 3 mg of protein lysate was added to the beads and gently rotated on a spinning wheel for 1 h at room temperature. The flow-through was then separated using a magnetic stand, while ensuring that the His-tag beads remained undisturbed. The collected flow-through was immediately used for the streptavidin pull-down experiment as described above.

### Auxin treatment

A 100× stock solution of auxin (indole-3-acetic acid) (Thermo Fisher Scientific, Cat# A10556-06-Alfa Aesar) was prepared at a concentration of 0.4 M by dissolving 0.7 g of auxin in 10 ml of 100% ethanol. A working solution was made by diluting 25 µl of the stock solution in 225 µl of sterile distilled water, which was then applied to plates containing mixed-stage worms (Joseph et al, 2020). Worms were treated with auxin for 2 h at 25 °C. After this treatment, the worms were washed twice with M9 buffer and twice with RIPA lysis buffer before lysis; protein concentration was determined as described above.

### LC-MS/MS

Detailed descriptions of the LC-MS/MS methods and protein identification pipelines for each of the eight experiments are provided in Appendix Supplementary Methods S2.

### Fold-change estimates

Fold-change estimates for NEKL–MLT proteins that were missing within all N2 technical replicates for a given experiment were obtained as follows. First, we identified the 10 lowest average intensity values for the N2 sample for which positive values were obtained in each of the technical replicates (Dataset EV9, tab 1). Next, these values were averaged and used as an estimate for the intensity of the missing NEKL–MLT proteins. Finally, these values were used to compute conservative fold-change estimates using average intensity values of the NEKL–MLT proteins identified in the bait–TurboID sample ($\log_2$ scale; Dataset EV9).

### Protein modeling

Predicted protein structures were generated using AlphaFold3, and follow-up modeling was carried out using UCSF ChimeraX 1.9 and AFM-LIS software (Abramson et al, 2024; Evans et al, 2022; Jumper et al, 2021; Kim et al, 2024; Pettersen et al, 2021).

### Statistical analyses and data analyses

Statistical tests were performed with GraphPad Prism and Microsoft Excel software using established methods (Fay and Gerow, 2013). Website tools for generating Venn diagrams included https://bioinformatics.psb.ugent.be/webtools/Venn/ and https://www.biovenn.nl (Hulsen et al, 2008).

### RNAi experiments

RNAi clones (Geneservice Library and Fay Lab) corresponding to *ddl-2*, *vps-13*, *cdc-37*, *mlt-2*, and *nekl-2* were tested on strains of the indicated genotypes using previously described methods and reagents (Balasubramaniam et al, 2023; Joseph et al, 2020; Kamath and Ahringer, 2003; Lazetic et al, 2018).

### Genome editing

CRISPR/Cas9 methods were used to introduce mutations into *catp-1* [intron-15 splice donor (GT > AT)] and *bcc-1* [G32S] (Binti et al, 2024a; Farboud and Meyer, 2015; Farboud et al, 2019; Fay et al, 2021; Ghanta et al, 2021). For *bcc-1* [G32S], the following primers were used: (1) sgRNA, 5'-aattggagtcaatgataac-3'; (2) Repair template, 5'-gatggaagattcgagcagaaaattcaagtcgatcgacgaaaattggagtc-catgattactagtgatcagaagttcctctccaggaacgaatcaggaacaaaaataaagactgt-3'; Screening primers, 5'-CGTTGTCTCGAGATGAATCAG-3' and 5'-GTGGCCAATCAGCTATTAGCC-3'; Sequencing primers, 5'-CAACTAACGACCACCCATCGG-3' and 5'-CTAAAATTGTGATCTTTAATGG-3'. The edited allele contains a novel Spe I site.

## Data availability

The mass spectrometry proteomics data have been deposited to the ProteomeXchange Consortium via the PRIDE [1] partner repository with the dataset identifiers PXD069727 and PXD069753. Submission details: Project Name: Proximity Labeling of NIMA Kinase Complex Components in *C. elegans*. Project accession: PXD069727. Submission details: Project Name: Wild-type versus Pdpy-7::GFP::TurboID control. Project accession: PXD069753.

The source data of this paper are collected in the following database record: biostudies:S-SCDT-10_1038-S44318-025-00660-5.

## Peer review information

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

## Acknowledgements

We thank Amy Fluet for editing this manuscript; M de Bono and M Artan for strains and helpful advice; Roland Dunbrack, Elizabeth Conibear, and Tom Goddard for helpful discussions regarding protein modeling; Nic Blouin for bioinformatic expertise; Zachary Davis for strain construction; and Dennis Province, Rick Edmondson, Stephanie Byrum, and Sam Mackintosh at the IDeA National Resource for Quantitative Proteomics at the University of Arkansas for Medical Biosciences (UAMS). Some strains were provided by the CGC, which is funded by the NIH Office of Research Infrastructure Programs (P40 OD010440). This research was supported by the National Institutes of Health NIGMS P20 GM103432 and R35 GM136236.

## Author contributions

**David S Fay**: Conceptualization; Resources; Data curation; Formal analysis; Supervision; Funding acquisition; Validation; Investigation; Visualization; Methodology; Writing—original draft; Project administration; Writing—review and editing. **Boopathi Balasubramaniam**: Data curation; Validation; Investigation; Methodology. **Sean M Harrington**: Software; Formal analysis. **Philip T Edeen**: Data curation; Validation; Investigation; Methodology.

Source data underlying figure panels in this paper may have individual authorship assigned. Where available, figure panel/source data authorship is listed in the following database record: biostudies:S-SCDT-10_1038-S44318-025-00660-5.

## Disclosure and competing interests statement

The authors declare no competing interests.

# Expanded View Figures

**Figure EV1.  AlphaFold modeling of NEKL–MLT interactions.**

(A–E) AlphaFold3 modeling and ChimeraX rendering of predicted (**A**) NEKL-3–MLT-3, (**B**) MLT-2–MLT-4–NEKL-2, (**C**) MLT-2–MLT-3, (**D**) NEKL-2–NEKL-3, and (**E**) MLT-3–NEKL-3–NEKL-2–MLT-2–MLT-4 protein complexes. Black lines between proteins indicate high-confidence Predicted Aligned Error (PAE) pseudobonds (PAE ≤ 6 Å; interatomic-distance ≤5 Å). For clarity, highly unstructured portions of the peptides (B-factor ≤30) were hidden. (**A–E**) Show protein-backbone ribbon diagrams (colored by peptide) along with single-color (green) PAE diagrams indicating regions of high structural confidence (low PAE) within and between the indicated proteins. Dashed-line cyan boxes correspond to regions where the two proteins are predicted for form a coherent structural unit. (**A–D**) Additionally show protein ribbon diagrams colored by PAE domain along with corresponding color-matched PAE diagrams. Ribbon diagrams in (**C–E**) also show partially transparent surface renderings. Inter-chain predicted Template Modeling (ipTM) values, and Local Interaction Scores (LIS) are shown for the indicated proteins modeled as heterodimers by AlphaFold3. Note that the five-protein complex (**E**) predicts very similar protein–protein interfaces as the subcomplex models, consistent with the hypothesis that the two subcomplexes can physically interact.

▶

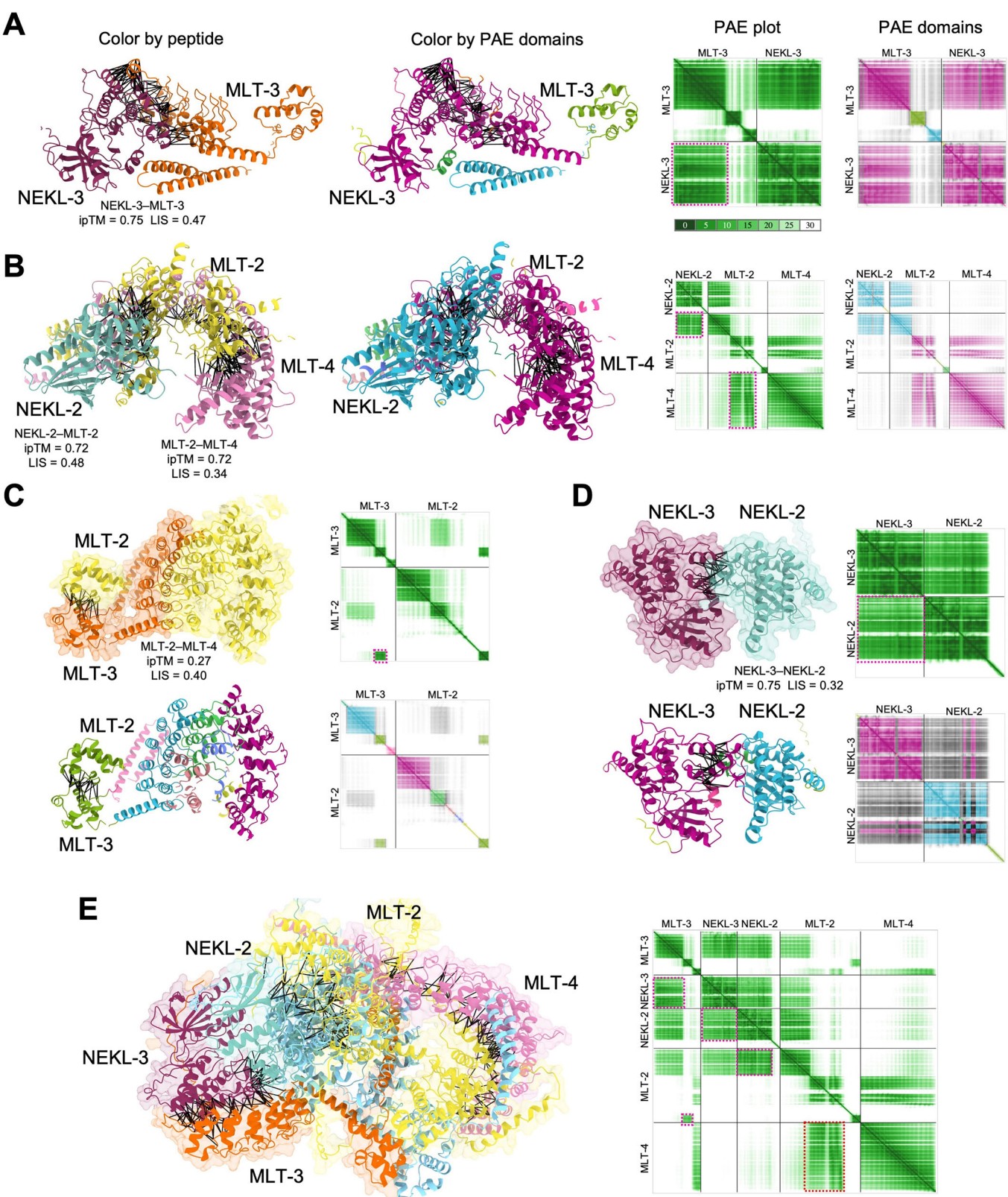

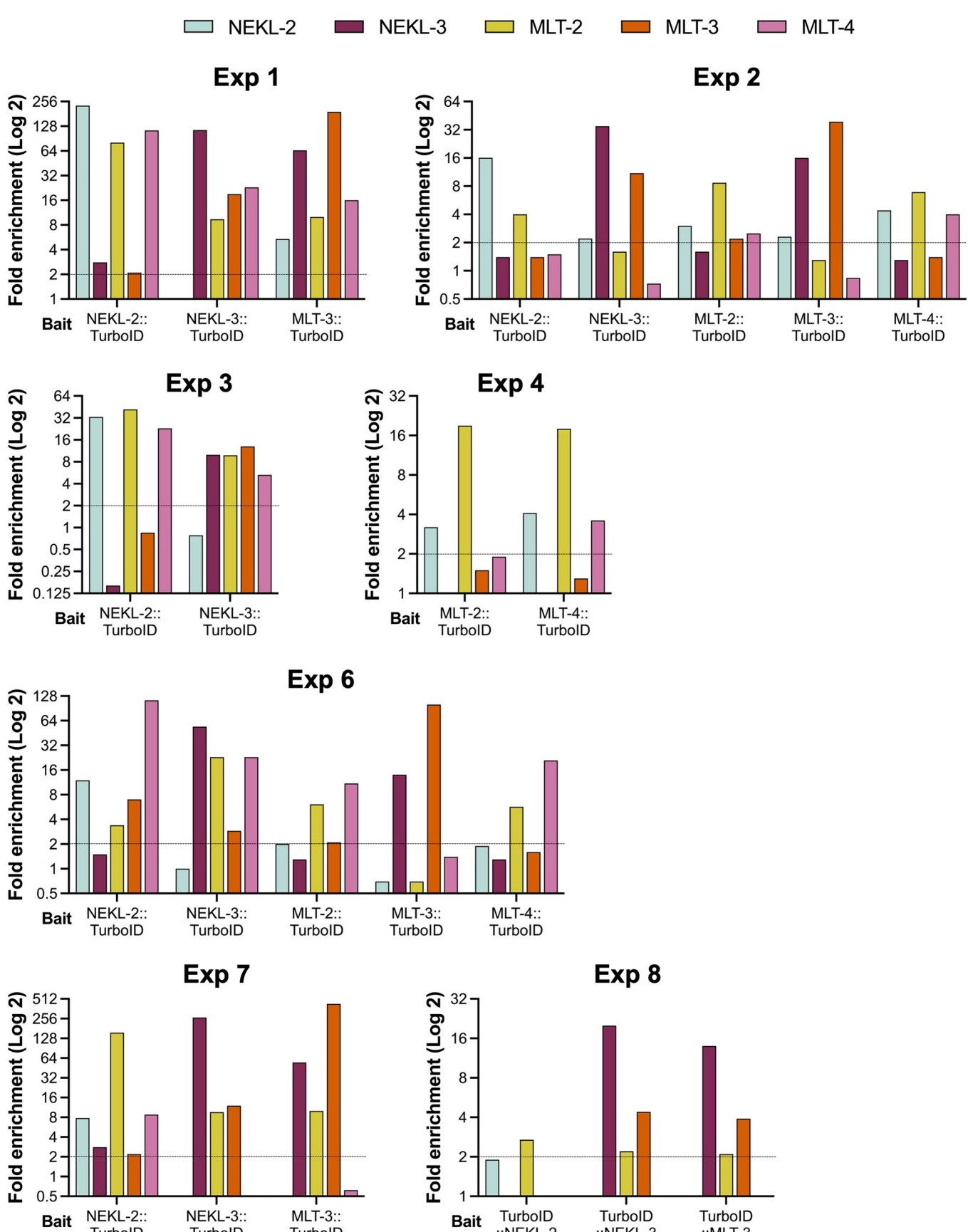

**Figure EV2. Fold changes of NEKL-MLT bait and prey proteins.**

Bar graphs of fold changes shown in Fig. 2B and Datasets EV1–8 for Exp 1–4 and 6–8.

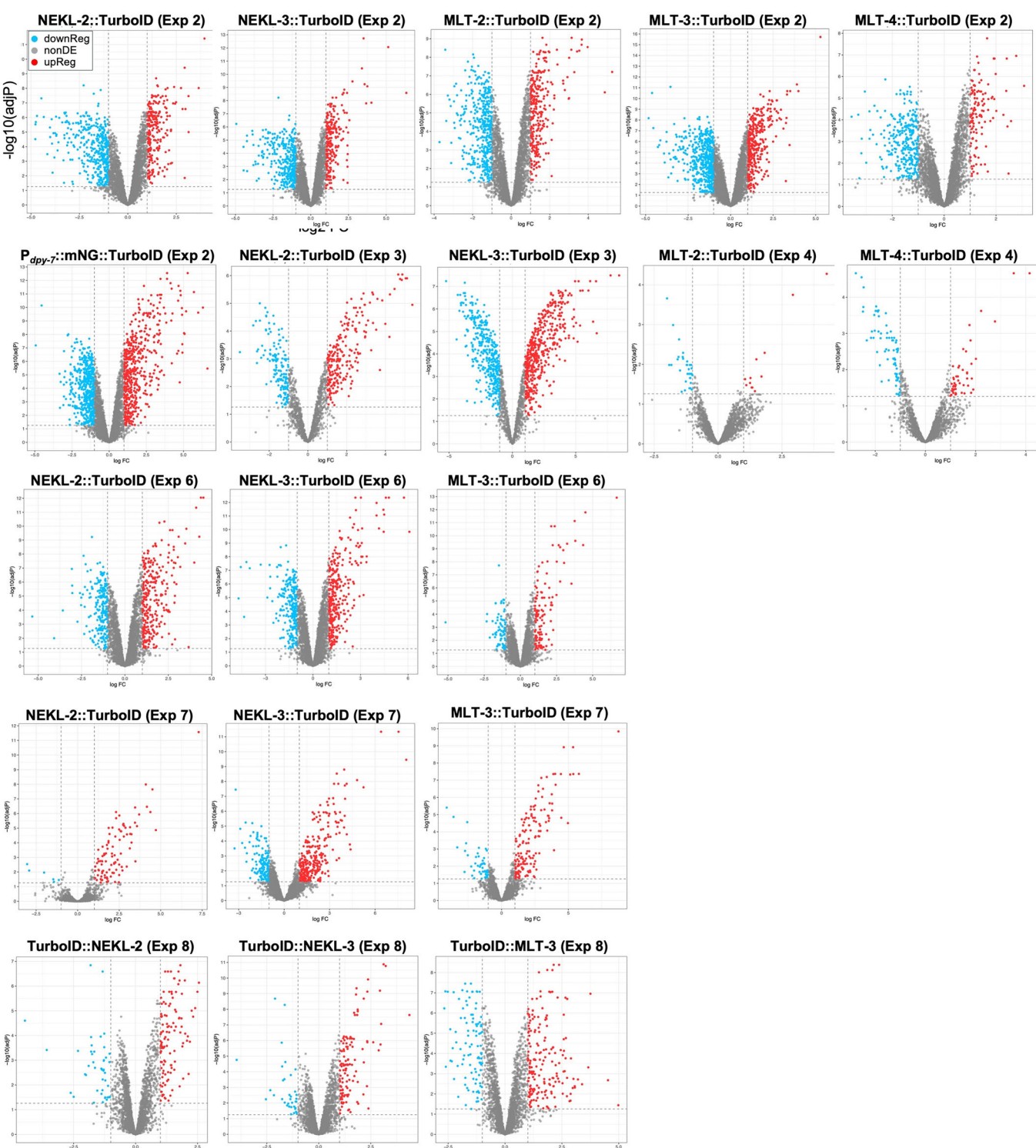

**Figure EV3. Volcano plots.**

Volcano plots of bait–TurboID strains versus N2 control showing the $\log_2$ fold change (FC) versus $-\log_{10}$ adjusted $P$ value. Red and blue dots indicate proteins that were up and down, respectively, in the bait strain relative to the N2 strain; select identified bait and prey proteins are indicated by arrows. NonDE, not differentially expressed. Red and blue dots indicate proteins that were up in the bait and N2 strains, respectively. Additional volcano plots are shown in Figs. 3A, 6B, EV5, and EV7. Data corresponding to volcano plots are available in Datasets EV1–8.

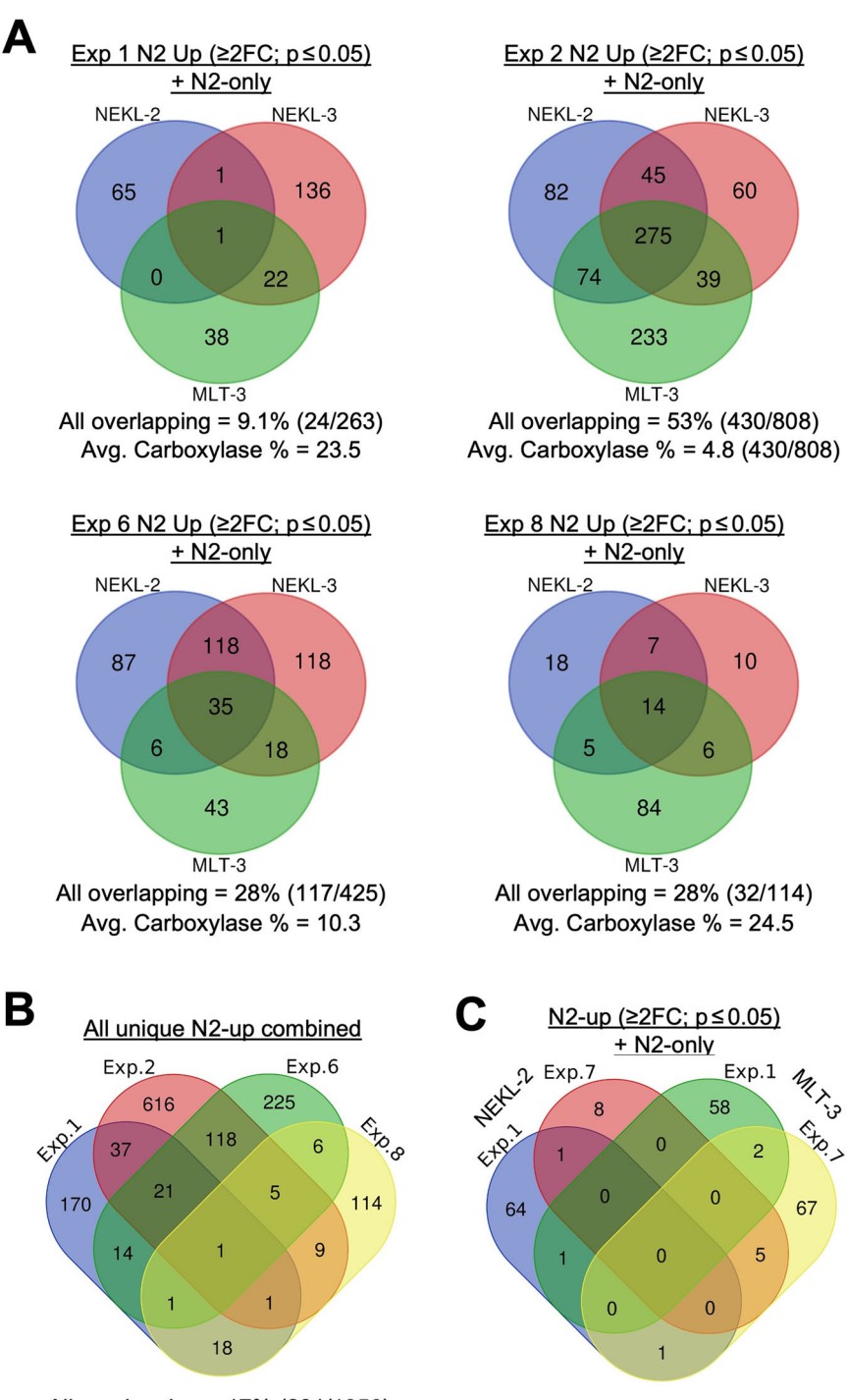

**Figure EV4. Analysis of N2-enriched proteins.**

(A) Venn diagrams showing overlap of proteins enriched (≥ 2-fold (FC); adjusted $P ≤ 0.05$) in N2 samples (N2 Up) versus NEKL-2::TurboID, NEKL-3::TurboID, and MLT-3::TurboID bait samples from Exp 1, Exp 2, Exp 6, and Exp 8. N2-only indicates proteins that were present in all technical replicates of the N2 sample but were absent in all technical replicates of the corresponding bait–TurboID; Also see Fig. 2B. Note variability in the extent of overlap among experiments. (B) Venn diagram comparing overlaps among the combined N2 Up proteins from Exp 1, Exp 2, Exp 6, and Exp 8. Note the relative lack of consistently overlapping proteins among experiments. (C) Venn diagram showing overlaps of N2-enriched proteins from NEKL-2::TurboID and MLT-3::TurboID (Exp 1 and Exp 7). Little overlap was observed between comparable biological replicates of NEKL-2::TurboID and MLT-3::TurboID as well as between the two bait proteins within the same biological replicate.

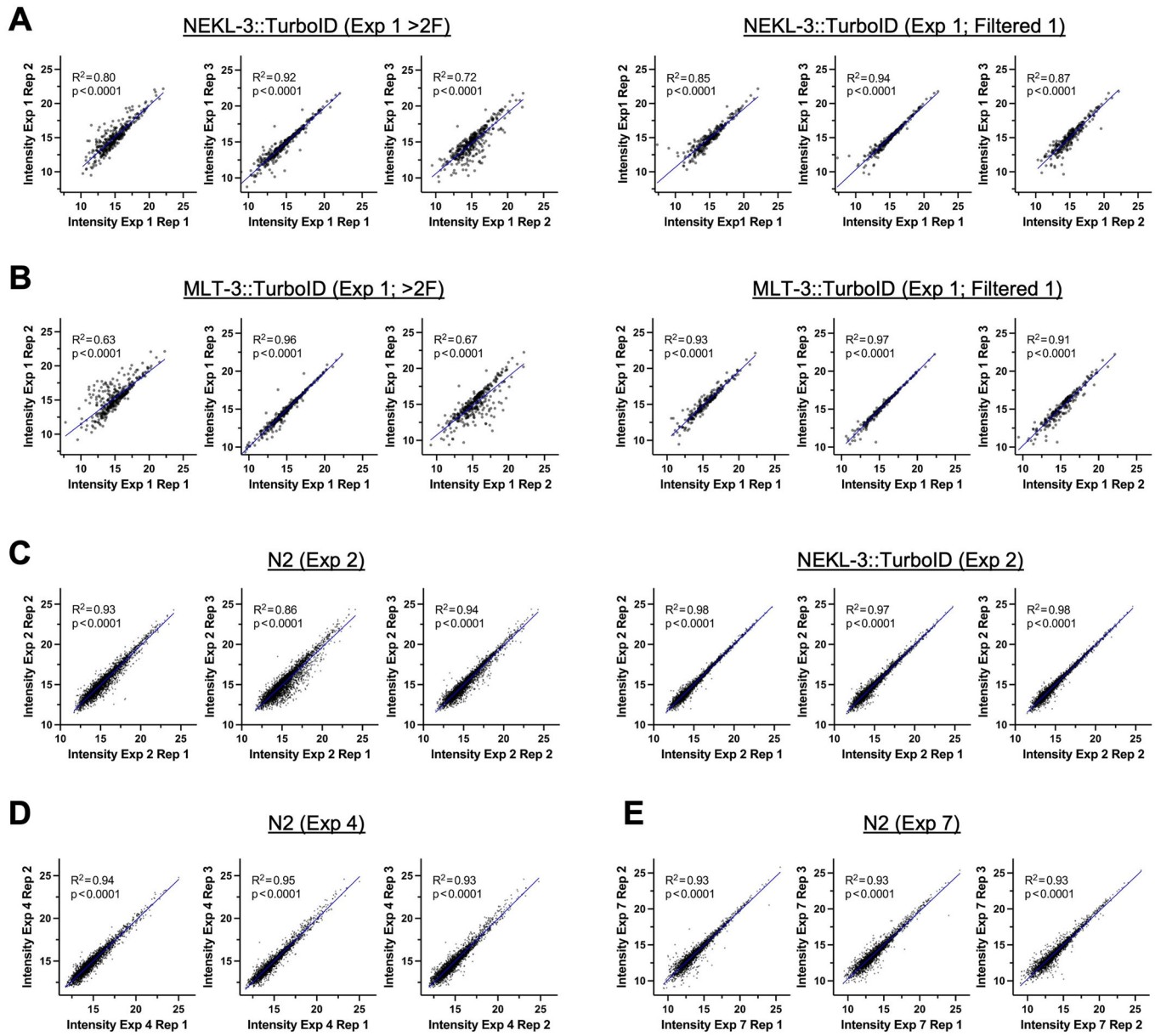

**Figure EV5. Additional analysis of technical replicates.**

(A–E) Scatter plot diagrams showing correlations of individual protein intensities (abundance) between the indicated replicate pairs. $R^2$ values were derived using simple linear regression; raw data are available in Dataset EV10. Also see Fig. 5.

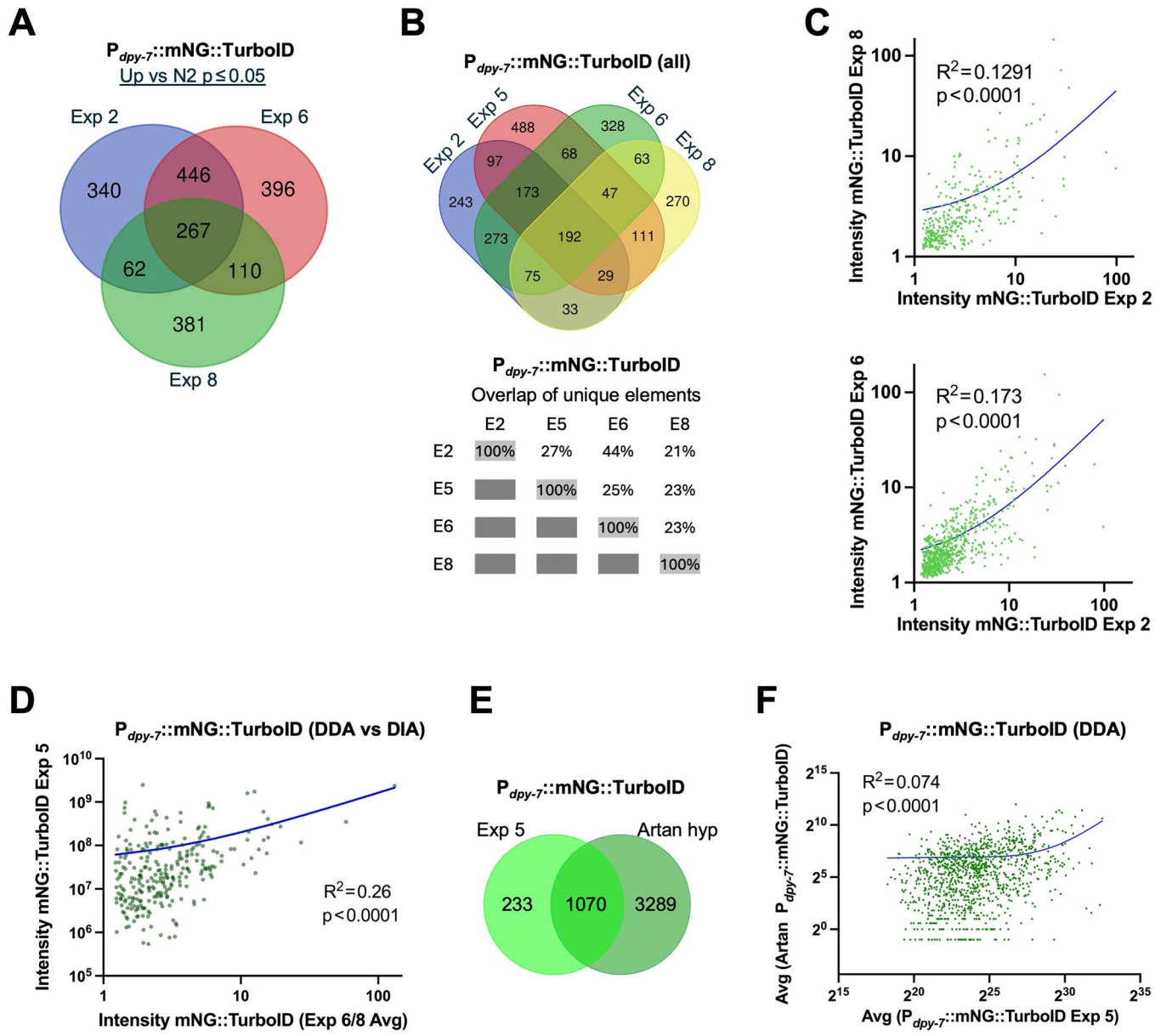

Figure EV6. Additional analysis of P$_{dpy-7}$::mNG::TurboID datasets.

(A) Venn diagram overlap of proteins significantly enriched (based on adjusted *P* values) in the three DIA–MS P$_{dpy-7}$::mNG::TurboID experiments. (B) Venn diagram and quantification of pairwise overlaps for the four P$_{dpy-7}$::mNG::TurboID experiments. Proteins included were detected in all technical replicates. Note that although Exp 2 and Exp 6 showed the greatest overlap in detected proteins, the correlation in protein levels was strongest between Exp 6 and Exp 8 (also see Fig. 6C). (C) Scatter plots showing correlations of protein intensities (abundance) for the mNG::TurboID control strain from indicated experimental pairs. *R²* values were derived using simple linear regression; raw data are available in Dataset EV11. The curved regression lines are due to the conversion of axes to a log scale. The markedly lower *R²* values, relative to Exp 6 vs Exp 8, are due in large part to discrepancies in intensity values at the high end of the scale. (D) Scatter plot showing weak correlation between Exp 5 (DDA–MS) and averaged data from Exp 6 and Exp 8 (DIA–MS). (E) Overlap and (F) correlation between Exp 5 and DDA–MS experiment from Artan and colleagues (PMID: 35665632). Note that although the proteins detected in Exp 5 overlapped extensively with a subset of those previously reported, the correlation in their expression levels was weak.

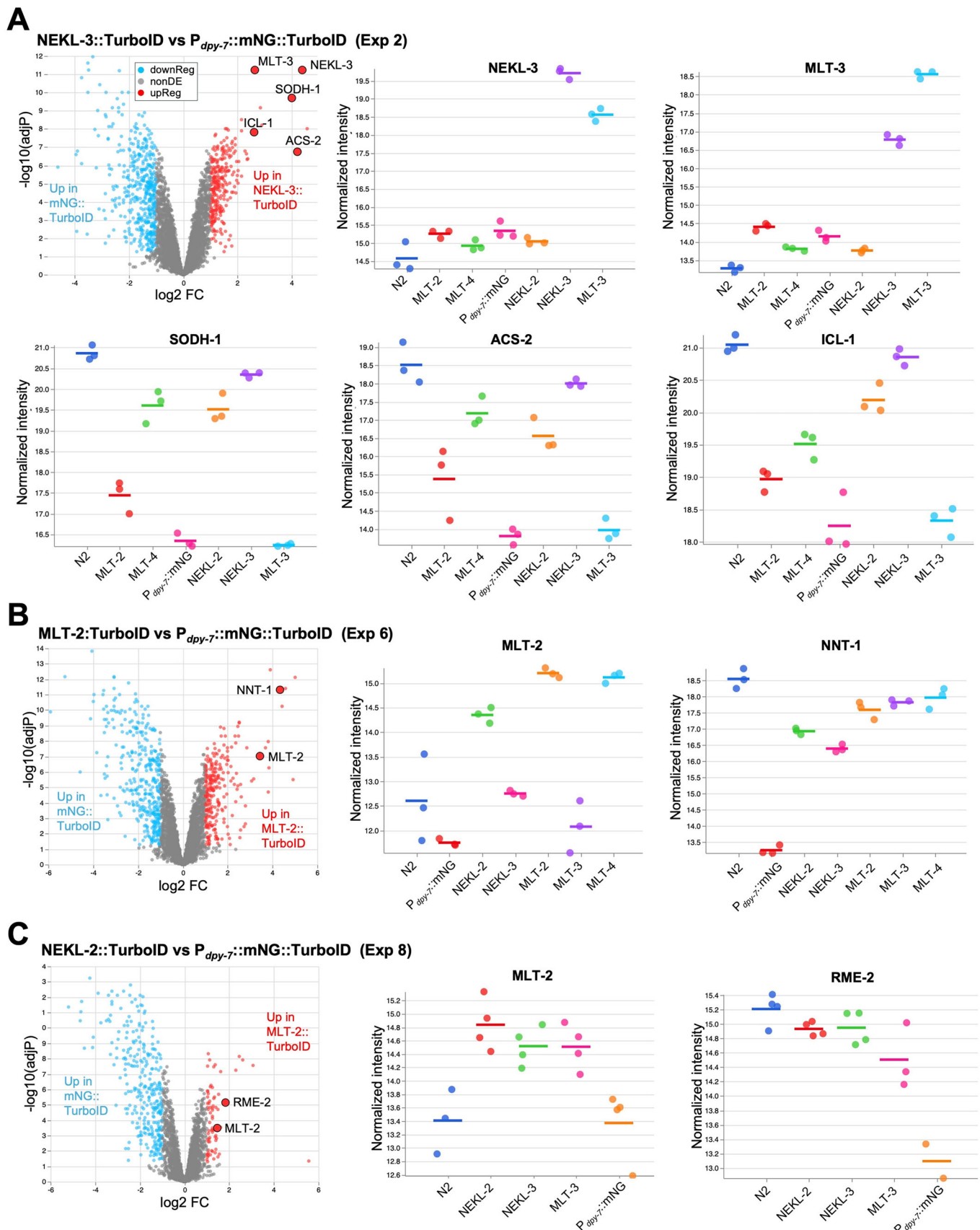

◀ **Figure EV7. P$_{dpy-7}$::mNG::TurboID down proteins.**

(A–C) Volcano plots of P$_{dpy-7}$::mNG::TurboID versus NEKL-3::TurboID (**A**), MLT-2::TurboID (**B**), and NEKL-2 TurboID (**C**) indicating the log$_2$ fold change (FC) versus –log$_{10}$ adjusted P values. Enlarged red dots indicate proteins shown in the accompanying intensity plots. Intensity plots (log$_2$ scale) showing the abundance of the indicated bait and prey proteins; individual dots represent values from the three technical replicates. For SODH-1, ACS-2, ICL-1, NNT-1, and RME-2, levels in P$_{dpy-7}$::mNG::TurboID were markedly lower than in the N2 or most bait–TurboID strains.

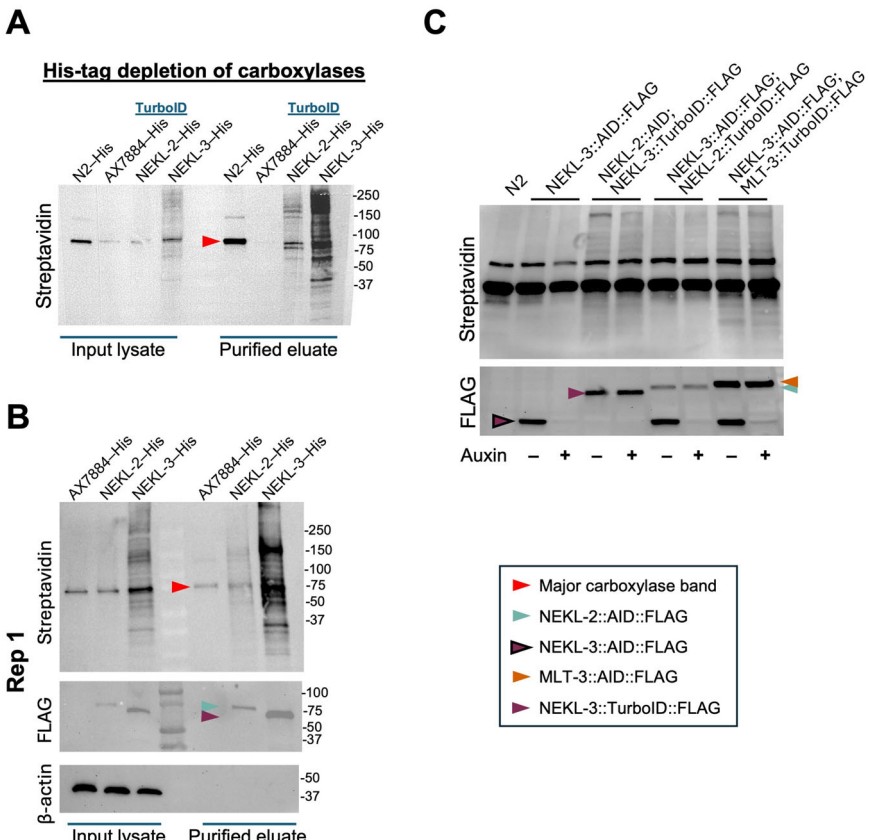

**Figure EV8. Methodological variations.**

(A–C) Western blots with specific proteins indicated by the key. (A) Streptavidin-probed western blot showing a reduction in the major carboxylase band in strains that underwent carboxylase depletion (AX7884, NEKL-2, and NEKL-3). –His, lysates were incubated with nickel-coated (His-binding) beads to remove the endogenous His-tagged versions of PYC-1, PCCA-1, MCCC-1, and POD-2. N2 does not encode His-tagged carboxylases, accounting for the stronger band in this strain. Increased protein concentration in the eluate lanes facilitates the visualization of the bait-induced biotinylated bands. (B) Western blots from Exp 3, replicate 1 Exp showing biotinylated products (streptavidin), bait proteins (FLAG), and loading control (β-actin). (C) Western blot of lysates from the indicated strains after incubation in the presence or absence of auxin, which leads to the degradation of the AID-tagged proteins. Note that the NEKL-2::AID strain contained a defective FLAG tag and was thus not detectable.

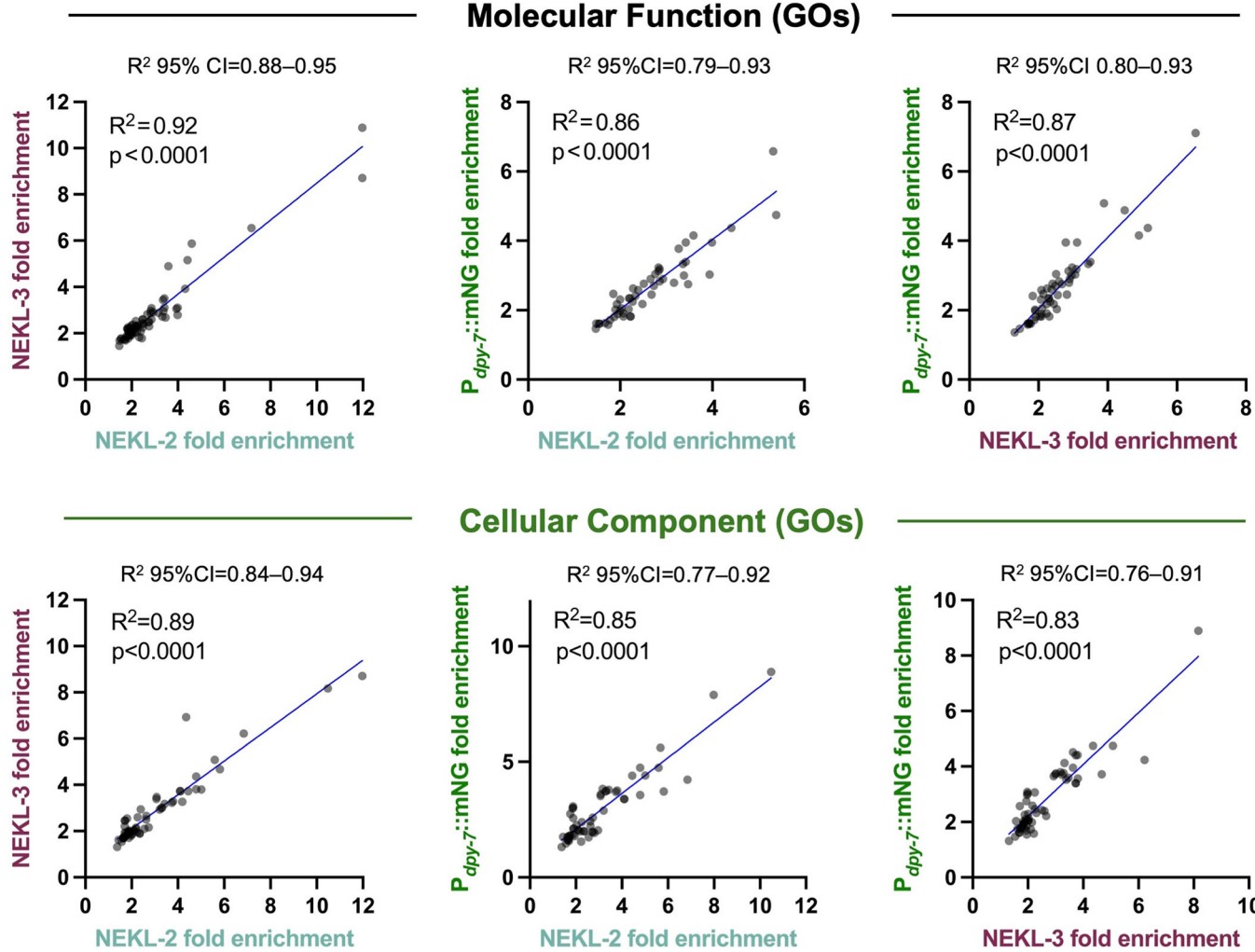

**Figure EV9. Correlations of GO terms.**

Scatterplots showing correlations in fold enrichments of Molecular Function and Cellular Component GOs for the combined NEKL-2::TurboID, NEKL-3::TurboID, and P$_{dpy-7}$::mNG::TurboID datasets. 95%CI, 95% confidence interval. Although correlation was greatest between NEKL-2–NEKL-3, this was not statistically significant based on CIs.

