## [Peer Review File · The EMBO Journal]

Integrating endogenous TurboID and data-independent acquisition mass spectrometry for *in vivo* proximity labeling

David Fay, Boopathi Balasubramaniam, Sean Harrington, and Philp Edeen

Corresponding author: David Fay (davidfay@uwyo.edu)

Review Timeline:

Transferred from Review Commons:	11th Aug 25
Editorial Decision:	25th Sep 25
Editor's Correspondence:	1st Oct 25
Revision Received:	7th Oct 25
Accepted:	7th Nov 25

Editor: Daniel Klimmeck

Transaction Report:

This manuscript was transferred to The EMBO Journal following peer review at Review Commons.

Review #1**1. Evidence, reproducibility and clarity:****Evidence, reproducibility and clarity (Required)**

Proximity labeling has become a powerful tool for defining protein interaction networks and has been utilized in a growing number of multicellular model systems. However, while such an approach can efficiently generate a list of potential interactors, knowledge of the most appropriate controls and standardized metrics to judge the quality of the data are lacking. The study by Fay systematically investigates these questions using the *C. elegans* NIMA kinase family members NEKL-2 and NEKL-2 and their known binding partners MLT-2, MLT-3 and MLT-4. The authors perform eight TurboID experiments each with multiple NEKL and MLT proteins and explore general metrics for assessing experimental outcomes as well as how each of the individual metrics correlates with one another. They also compare technical and biological replicates, explore strategies for identifying false positives and investigate a number of variations in the experimental approach, such as the use of N- versus C-terminal tags, depletion of endogenous biotinylated proteins, combining auxin-inducible degradation, and the use of gene ontology analysis to identify physiological interactors. Finally, the authors validate their findings by demonstrating that a number of the candidate identified functionally interact with NEKL-2 or components of the WASH complex.

Overall this is an outstanding study that will be of great interest to those interested in using proximity labeling to identify interactors of their favorite protein. The experiments are well executed and the data presented in a mostly clear manner. I really like this study (particularly because I plan to do a proximity labeling study of my own), but I did come away less than impressed with some of the analysis. This is a data-dense manuscript, and it appears to me that the authors tried to cover so much ground that in some cases very little insight was provided. For instance, the authors promote the use of data independent acquisition (DIA) as compared to the more commonly used data dependent acquisition (DDA). However the authors do not provide any analysis to indicate one approach is better than the other. Likewise the combined use of auxin-induced degradation and proximity labeling is explored but there is very little to take away from these experiments. Despite these issues, I am very enthusiastic about the study as a whole. Below I list major and minor concerns.

****Major concerns****

1. My biggest issue with the manuscript is that a lot is made of the use of data independent acquisition (DIA) as compared to the more commonly used data dependent acquisition (DDA). The authors perform experiments using DIA and DDA approaches but do not directly compare the outcomes. As a result there is really no way to know if one approach is better than the other. I would suggest the authors either perform the necessary analysis to compare the two approaches or tone down their promotion of DIA.

2. Line 75, The authors promote the use of data-independent acquisition (DIA) without defining what this approach is and how it differs from the more conventional data-dependent acquisition. As a non-mass spectroscopist, I found myself with lots of question concerning DIA, what it is and how it differs from DDA. I think it would really be helpful to expand the description of DIA and its comparison with DDA in the introduction.

****Minor concerns:****

Line 92 typo. I believe the authors meant to say NEKL-2-MLT-2-MLT-4.

Line 169. Is exogenous the correct word to use here? It suggests that you are talking about non-worm proteins, but I know you are not.

Line 177 typo (D) should be (C).

Figure 1C: Lucky Charms may sue you for infringement of their trademarked marshmallow treats.

Figure 1D The NEKL-2::TurboID band is indicated with a green triangle in the figure but the figure legend states that green triangles indicate mNG::TurboID control. I know this triangle is a shade off the triangle that indicates mNG::TurboID but it's really hard to see the difference. All of the differently colored triangles in panel F are unnecessary. I would either just pick one color for all non-control bait proteins or better yet, only use a triangle to point to bands that are not obvious. For instance I don't need the triangles that point to NEKL-2 -3 and -4 fusion proteins. These are just distracting.

Line: 316: Conceivably, another factor that could contribute to the counterintuitive upregulation of some proteins in the N2 samples is related to the fusion proteins that are being expressed in the TurboID lines. A partially functional bait protein (one with a level of activity similar to nekl-2(fd81) that may not result in an obvious phenotype) could directly or indirectly affect gene expression leading to lower levels of a subset of proteins in the TurboID samples. The same could be said for fusion proteins with a gain-of-function effect.

Fig 3 B-E. I am a little confused how the data in these graphs is normalized. For instance, I would have expected that for NEKL-3 in panel B, that the normalized (log₂) intensity value in N2 be set at 0 as it is for NEKL-2. Maybe I just don't have enough information on how these plots were generated.

Figure 6C legend is not correct.

Line 575: Figure reference should be Fig. S5G. The authors should check to make sure all references to supplemental figures include correct panel information.

Line 576. The authors reference a study by Artan and colleagues and report a weak correlation between their study and that of Artan. They reference figure S4 but it should be Fig S5H.

Line 652. The authors note that numerous proteins were present at substantially reduced levels in the mNG::TurboID samples and suggest that sticky proteins may have been outcompeted or otherwise excluded from beads incubated with the mNG::TurboID lysates. Why would sticky proteins only be a problem in these samples? The reasoning is not clear to me.

Line 745: The term "bait overlaps" is a bit vague. Ultimately, I figured out what it meant but it was not immediately obvious.

S7B Fig. Why is actin missing from the eluate?

Line 873: The authors state the extent of overlap in GO terms between the various experiments and provide percentages. I tried to extract this information from Figure 8C and came up with different values. For instance, in the case of Molecular Function, they state that they observed a 54% overlap between NEKL-2 and NEKL-3 but in the Venn diagram in Figure 8C I see that the NEKL-2 and NEKL-3 experiments had 71 (25+46) GO terms in common. Out of 98 GO terms for NEKL-2 or 104 for NEKL-3 the percentage I got is closer to 72. Am I analyzing this correctly?

2. Significance:

Significance (Required)

Overall this is an outstanding study that will be of great interest to those interested in using proximity labeling to identify interactors of their favorite protein. The experiments are well executed and the data presented in a mostly clear manner. I really like this study (particularly because I plan to do a proximity labeling study of my own), but I did come away less than impressed with some of the analysis. This is a data-dense manuscript, and it appears to me that the authors tried to cover so much ground that in some cases very little insight was provided. For instance, the authors promote the use of data independent acquisition (DIA) as compared to the more commonly used data dependent acquisition (DDA). However the authors do not provide any analysis to indicate one approach is better than the other. Likewise the combined use of auxin-induced degradation and proximity labeling is explored but there is very little to take away from these experiments. Despite these issues, I am very enthusiastic about the study as a whole.

3. How much time do you estimate the authors will need to complete the suggested revisions:

Estimated time to Complete Revisions (Required)

(Decision Recommendation)

Between 1 and 3 months

4. Review Commons values the work of reviewers and encourages them to get credit for their work. Select 'Yes' below to register your reviewing activity at Web of Science Reviewer Recognition Service (formerly Publons); note that the content of your review will not be visible on Web of Science.

Yes

Review #2

1. Evidence, reproducibility and clarity:

Evidence, reproducibility and clarity (Required)

This study expanded the use of data-independent acquisition-mass spectrometry (DIA-MS) in TurboID proximity-labeling proteomics to identify novel interactors of NEKL-2, NEKL-3, MLT-2, MLT-3, and MLT-4 complexes in *C. elegans*. The authors described several useful metrics to evaluate the quality of TurboID experiments, such as using the percentage of upregulated genes, the percentage of proteins present only in bait-TurboID experiments as

compared to N2 controls, and the percentage of endogenously biotinylated carboxylases as internal controls. Further, the authors introduced methodological variability across 23 TurboID experiments and evaluated any improvement to the resulting data, such as N-terminally tagging bait proteins with TurboID, depleting endogenous carboxylases, and auxin-inducible degradation of known complex members. Finally, this study identified the kinase folding chaperone CDC-37 and the WASH complex component DDL-2 as novel interactors with the NEKL-MLT complexes through an RNAi-based enhancer approach following their identification by TurboID.

****Major comments:****

The key conclusions are convincing, and the work is rigorous. The work provides a clear roadmap to reproducing the data. The experiments are adequately replicated, and statistical analysis is adequate. We only have minor comments.

****Minor comments:****

- In the western blot in Fig 1 why does the mNG::Turbo have two bands?
- Fig 2B is difficult to parse as a reader. Columns labeled "Upreg," "Downreg," "TurboID only," "N2 only," "Filter-1," "Filter-2," and "Epi %" could be moved to Supplemental. Fold change vs N2 could be represented as a bar chart, allowing for trends between fold change and the metrics Upreg %, Turbo %, and Carboxylase % to be seen more clearly. Further, rows headed "Carboxylase depletion," "DDA," and "Auxin treated" could be presented as separate panels to better match the distinct points made in the text.
- Line 179: *in vivo* should be italicized
- Lines 215-217: The comparison between Western blot expression levels and prior fluorescent reporter levels is unclear. Could be reformatted to make it clearer that relative expression of the different NEKL-MLTs in this study is consistent with prior data.
- Lines 267-268: The final line of the passage is unclear and can be removed.
- Lines 311-313: This study is able to use the recovery of bait and known interactor proteins as internal controls to determine the quality of each experiment, but this may not always be the case for other users' experiments. The authors should comment on how Upreg %, a value influenced by many factors, can actually be used as a quality check when a bait protein has no known interactors.
- Line 702: There is a [new REF] that should be removed
- The approach used mixed stage animals, but some genes oscillate or are transiently expressed. Please discuss cost-benefit of mixed stage vs syncing.
- Authors were working on hypodermally expressed proteins. It would be valuable to

discuss what tissues are amenable to TurboID. Ie are the cases where there are few cells (anchor cell, glial sockets, etc) that it will be extremely challenging to perform this technique

- Authors mention approaches such as nanobodies, split Turbo. Based on their experiences it would be valuable to add Discussion on strengths and weaknesses of these approaches to guide folks considering TurboID and DIA-MS experiments in *C. elegans*

2. Significance:

Significance (Required)

- Advance in technique: This study expands the use cases of data-independent acquisition MS method (DIA-MS) in *C. elegans*, which fragments all ions independent of the initial MS1 data. The benefits of this approach include better reproducibility across technical replicates and better recovery of low abundance peptides, which are critical for advancing our ability to capture weak and transient interactions.

- The use of DIA-MS in this study has improved our understanding of the partners of these NEKL-MLTs in membrane trafficking, molting, and cell adhesion within the epidermis.

- In many papers, TurboID seems very trivial but this paper clearly highlights the limitations and will be an invaluable resource for labs that want to get proximity labeling established in their labs

3. How much time do you estimate the authors will need to complete the suggested revisions:

Estimated time to Complete Revisions (Required)

(Decision Recommendation)

Less than 1 month

4. Review Commons values the work of reviewers and encourages them to get credit for their work. Select 'Yes' below to register your reviewing activity at Web of Science Reviewer Recognition Service (formerly Publons); note that the content of your review will not be visible on Web of Science.

Yes

Review #3

1. Evidence, reproducibility and clarity:

Evidence, reproducibility and clarity (Required)

Summary:

Fay and colleagues perform a series of proximity labeling experiments in *C. elegans* followed by thorough and rational analysis of the resulting biotinylated proteins identified by LC-MS/MS. The overall goals of the study are to evaluate different techniques and provide practical guidance on how to achieve success. The major takeaways are that integration of data-independent acquisition (DIA) along with comparison of endogenously tagged TurboID alleles to soluble TurboID expressed in the same tissue results in improved detection of bona-fide interactors and reduced numbers of false-positives.

Major comments:

Overall the claims are solid and conclusions supported. The data and methods are substantial to enable reproducibility in other labs. The experiments have been repeated multiple times with particular attention to statistical analysis. I have no major concerns with the manuscript and focus primarily on improving the accessibility of this important contribution to the scientific community. As such, I suggest that the authors:

1. Provide more explanation of and rationale for using DIA. This is not yet a standard technique and most basic biomedical scientists will be unaware of the jargon. As I expect many labs in the *C. elegans* community and beyond will be interested in the guidance provided in this manuscript, the introduction offers a great opportunity to bring the reader up to speed, as opposed to sending them to the complicated proteomics analysis literature.
2. Provide a better overview of the various protocols tested (Experiments 1-8). Maybe at the beginning of the results, and maybe with an accompanying schematic. As currently written, it is difficult to figure out details regarding how the experiments vary and why.
3. As to be expected, expression of TurboID tags at endogenous levels via low abundance proteins in a complex multicellular system results in somewhat weak signals that flirt with the limit of detection. Perhaps by combining tagged alleles within the same complex (NEKL-3/MLT-3 or NEKL-2/MLT-2/MLT-4) the signals could be boosted? Tandem tags, either on one end or multiple ends of proteins might help as well. As the authors point out, a benefit of tagging the two NEKL-MLT complexes is that there are strong loss-of-function phenotypes (lethal molting defects) to help evaluate whether a tagging strategy results in a non-functional complex. THESE EXPERIMENTS ARE OPTIONAL and might simply be discussed at the authors discretion.

****Minor Comments:****

1. Figure 3A is cropped on the right.
2. Better define [new REF] on line 702.

****Referee cross-comments****

Overall, I am in agreement with, and supportive of, the other reviewers' comments.

2. Significance:

Significance (Required)

Proximity labeling is often proposed as a technique to determine interaction networks of proteins in vivo, but in practice it remains challenging for most labs to execute a successful experiment, especially within the context of multicellular model organisms. Fay and colleagues provide a much needed roadmap for how to best approach proximity labeling experiments in *C. elegans* that will likely apply to other model systems.

They establish a rigorous approach by choosing to endogenously tag components of two essential NEKL-MLT complexes required for *C. elegans* molting. These complexes are relatively low abundance as they are only expressed in a single cell type, the hyp7 epidermal syncytium. In addition, as inactivation of any member of the complexes results in molting defects, they have a powerful selection for functional tags. Thus, they have set a high bar for themselves in order to discern whether a given variation on the experimental approach results in improved detection of interactors and fewer false positives.

Potential areas for improvement include lowering the expression level of the skin-specific soluble TurboID used to determine non-specific biotinylation events. This control results in much higher levels of biotinylation compared to the TurboID-tagged NEKL-MLT alleles and likely affects their analysis, which they openly admit. In addition, to reduce the high level of background biotinylation signals generated by endogenous carboxylases, they adopt a depletion strategy pioneered by other researchers but this does not offer major improvements in detection of specific signals. The source of these conflicting results remains to be determined. It is also curious that auxin-inducible degradation of components of the NEKL-MLT complexes did not robustly alter the resulting biotinylating capacity of other members. This approach should be evaluated in subsequent studies. Finally, as mentioned in Major Comment #3 (above), it would be interesting to see if

combining TurboID tags within the same complex might improve signal-to-background ratios.

This manuscript represents a methodological advance that will likely become an oft-cited reference for members of the *C. elegans* community and a springboard for other basic biomedical scientists wanting to adapt rigorous proximity labeling techniques to their system. I am a cell biologist that uses a variety of genetic, molecular and biochemical approaches, mostly centered around *C. elegans*. I have used LC/MS-MS in our studies but have relatively little expertise in evaluating all aspects of proteomic pipelines.

3. How much time do you estimate the authors will need to complete the suggested revisions:

Estimated time to Complete Revisions (Required)

(Decision Recommendation)

Less than 1 month

4. Review Commons values the work of reviewers and encourages them to get credit for their work. Select 'Yes' below to register your reviewing activity at Web of Science Reviewer Recognition Service (formerly Publons); note that the content of your review will not be visible on Web of Science.

No

Review #4

1. Evidence, reproducibility and clarity:

Evidence, reproducibility and clarity (Required)

Fay et al. describe an extensive proximity labeling BioID study in *C. elegans* with TurboID and DIA-LCMS analysis. They chose the NEKL-2/3 kinases and their known interactors MLT-2/3/4 as TurboID-fused bait proteins (C- and partially N-terminal fusions encoded from CRISPR-mediated genome edited genes). With eight biological replicates (and three to four technical replicates each) and with the unmodified wildtype or mNeonGreen-TurboID expressing worms as controls, a comprehensive dataset was generated. Although starting from quite different abundances of the bait-fusions within the cell lysates all bait proteins and known complex-binding partners were convincingly enriched with capturing

streptavidin beads after only one hour of incubation with the lysate. This confirms the general applicability of TurboID-BioID approach in *C. elegans*. The BioID method typically gives rise to large proteomics datasets (up to more than thousand proteins identified after biotin capture) with several tens to hundreds enriched proteins (against negative control strains) as potential proteins that localize proximal to the bait-TurboID protein. However, substantial variations of candidates between biological replicates are frequently observed in BioID experiments. The authors scrutinized their dataset towards indicative metrics, filters and cutoffs in order to separate high-confidence from low-confidence candidates. With the workflow applied the authors melt down the number of candidates to 15 proteins that were grouped in four functional groups reasonably associated to NEKL-MLT function.

Successful BioID experiments depend on reliable enrichment quantification with mass spectrometry using control cell lines that require a carefully bait-tailored design. Those must adequately express TurboID controls matching the abundance of the bait-TurboID fusion protein and its biotinylation activity. After affinity capture, sample preparation and LCMS data acquisition there is no silver bullet towards the identification true bait neighbors. Fay et al. elaborately describe their considerations and workflow towards high-confidence candidates. The workflow considered (i) data analysis with Volcano plots to account for statistical reproducibility of biological replicates against negative controls, (ii) fraction of proteins only detected in the positive or negative controls thus evading the fold-enrichment quantification approach, (iii) evaluation of variations in carboxylase enrichment as a measure for variations in the general biotin capture quality between experiments, (iv) an assessment of technical reproducibility with scatter plots and Venn diagrams, (v) exclusion of potentially false positives, e.g. promiscuously biotinylated non-proximal proteins, through comparisons with control worms expressing a non-localized mNeonGreen-TurboID fusion protein, (vi) batch effects, (vii) the impact of endogenous biotinylated carboxylases through depletion, (viii) gene ontology analysis of enriched proteins, (ix) weighing data according to the quality of individual experiments according to the afore mentioned metrics, and finally (x) genetic interaction studies to functionally associate high-confidence candidates with the bait.

****Major comments:****

Fay et al. present a solid, clear and comprehensive BioID-based proteomics study that takes into account and discusses decisive aspects for the (re)production and analysis of high-quality TurboID-based mass spectrometry data. Claims and conclusions are generally well and sufficiently supported by the presented data and illustrated with figures (throughout the text as well as with plenty of supplementary data). However, although the

authors claim to seek for substrates of the kinase complex they drew no further attention to the phosphorylation status of the captured proteins. Haven't the MS data been analyzed in this respect? Information regarding this issue would enhance the manuscript. Data generation and method description appear reproducible for readers. Also, the statistical analyses appear adequate. The authors should also consider to deposit their MS raw and analysis data in a public repository (e.g. PRIDE) for future reviewing processes and as reference data for readers and followers.

****Minor comments:****

The authors should combine supplementary data files to reduce the number of single files readers have to deal with.

The authors should avoid the term "upregulation" or "increased biotinylation" when capture enrichment is meant.

2. Significance:

Significance (Required)

The manuscript presents a robust BioID proteomics screening for co-localizing proteins of NEKL-2/3 kinases and their known interactors MLT-2/3/4. The ongoing validation of their functional interactions and whether the protein candidates reflect phosphorylation substrates or else remains elusive and is announced for upcoming manuscripts. The knowledge gain in terms of molecular mechanisms with NEKL-2/3 MLT-2/3/4 involvement in *C. elegans* is therefore limited to a table of - promising - interacting candidates that have to be studied further. Information about the phosphorylation status of the captured proteins from the MS data are not given. However, knowing the protein candidates will be of interest for groups working with these complexes (or the identified potentially interacting proteins) either in *C. elegans* or any other organism. Also, in-depth proteomics screenings with novel approaches such as BioID have to be established for individual organisms. For *C. elegans* there is only one prior BioID publication (Holzer et al. 2022). Many of the aspects discussed here have also been addressed earlier for BioIDs in other organisms and are not principally new. However, the presented study can be of conceptual interest for labs delving into or entangled with the BioID method in *C. elegans* or other organisms. The study addresses especially proteomics groups working on protein-protein interactions using proximity labeling/MS approaches. Basic consideration and thoughts for the experimental design and MS data analysis are given in detail and can serve as another guideline for future studies.

3. How much time do you estimate the authors will need to complete the suggested revisions:

Estimated time to Complete Revisions (Required)

(Decision Recommendation)

Less than 1 month

No

Full Revision

Manuscript number: RC-2025-03083

Corresponding author(s): David Fay

General Statements [optional]

This section is optional. Insert here any general statements you wish to make about the goal of the study or about the reviews.

We greatly appreciate the input of the four reviewers, all of whom carried out a careful reading of our manuscript, provided useful suggestions for improvements, and were enthusiastic about the study including its thoroughness and utility to the field. Because the reviewers required no additional experiments, we were able to address their comments in writing.

However, in response to a comment from reviewer #4 we decided to add an additional new biological finding to our study given that our functional validation of proximity labeling targets was not extensive. Namely, we now show that a missense mutation affecting BCC-1, one of the top NEKL–MLT interactors identified by our proximity labeling screen, is a causative mutation (together with *catp-1*) in a strain isolated through a forward genetic screen for suppressors of *nekl* molting defects (new Fig 9C). This finding, combined with our genetic enhancer tests, further strengthens the functional relevance of proteins identified through our proximity labeling approach and highlights the synergy of proteomics combined with classical genetics.

Positive statements from reviewers include:

Reviewer #1: Overall, this is an outstanding study that will be of great interest to those interested in using proximity labeling to identify interactors of their favorite protein. The experiments are well executed and the data presented in a mostly clear manner.

Reviewer #2: The key conclusions are convincing, and the work is rigorous. The work provides a clear roadmap to reproducing the data. The experiments are adequately replicated, and statistical analysis is adequate... In many papers, TurboID seems very trivial but this paper clearly highlights the limitations and will be an invaluable resource for labs that want to get proximity labeling established in their labs.

*Reviewer #3: Overall, the claims are solid and conclusions supported. The data and methods are substantial to enable reproducibility in other labs. The experiments have been repeated multiple times with particular attention to statistical analysis. ...This manuscript represents a methodological advance that will likely become an oft-cited reference for members of the *C. elegans* community and a springboard for other basic biomedical scientists wanting to adapt rigorous proximity labeling techniques to their system.*

Reviewer #4: Fay et al. present a solid, clear and comprehensive BioID-based proteomics study that takes into account and discusses decisive aspects for the (re)production and analysis of

high-quality TurboID-based mass spectrometry data. Claims and conclusions are generally well and sufficiently supported by the presented data and illustrated with figures (throughout the text as well as with plenty of supplementary data)... Basic consideration and thoughts for the experimental design and MS data analysis are given in detail and can serve as another guideline for future studies.

Based on these reviews and comments, we believe that our manuscript is suitable for publication in a high-impact journal.

1. Point-by-point description of the revisions

This section is mandatory. Please insert a point-by-point reply describing the revisions that were already carried out and included in the transferred manuscript.

Reviewer #1 (Evidence, reproducibility and clarity (Required)):

Proximity labeling has become a powerful tool for defining protein interaction networks and has been utilized in a growing number of multicellular model systems. However, while such an approach can efficiently generate a list of potential interactors, knowledge of the most appropriate controls and standardized metrics to judge the quality of the data are lacking. The study by Fay systematically investigates these questions using the *C. elegans* NIMA kinase family members NEKL-2 and NEKL-2 and their known binding partners MLT-2, MLT-3 and MLT-4. The authors perform eight TurboID experiments each with multiple NEKL and MLT proteins and explore general metrics for assessing experimental outcomes as well as how each of the individual metrics correlates with one another. They also compare technical and biological replicates, explore strategies for identifying false positives and investigate a number of variations in the experimental approach, such as the use of N- versus C-terminal tags, depletion of endogenous biotinylated proteins, combining auxin-inducible degradation, and the use of gene ontology analysis to identify physiological interactors. Finally, the authors validate their findings by demonstrating that a number of the candidate identified functionally interact with NEKL-2 or components of the WASH complex.

Overall this is an outstanding study that will be of great interest to those interested in using proximity labeling to identify interactors of their favorite protein. The experiments are well executed and the data presented in a mostly clear manner. I really like this study (particularly because I plan to do a proximity labeling study of my own), but I did come away less than impressed with some of the analysis. This is a data-dense manuscript, and it appears to me that the authors tried to cover so much ground that in some cases very little insight was provided. For instance, the authors promote the use of data independent acquisition (DIA) as compared to the more commonly used data dependent acquisition (DDA). However the authors do not provide any analysis to indicate one approach is better than the other. Likewise the combined use of auxin-induced degradation and proximity labeling is explored but there is very little to

Full Revision

take away from these experiments. Despite these issues, I am very enthusiastic about the study as a whole. Below I list major and minor concerns.

Major concerns

1. My biggest issue with the manuscript is that a lot is made of the use of data independent acquisition (DIA) as compared to the more commonly used data dependent acquisition (DDA). The authors perform experiments using DIA and DDA approaches but do not directly compare the outcomes. As a result there is really no way to know if one approach is better than the other. I would suggest the authors either perform the necessary analysis to compare the two approaches or tone down their promotion of DIA.

We agree and have scaled back any statements comparing DDA to DIA as our manuscript did not address this directly. We also now point out this caveat in our closing thoughts section, while referencing other studies that compared the two (lines 926-929). Our main point was to convey that DIA worked well for our proximity labeling studies but has seen little use by the model organism field. Surprising (to us), DIA was also considerably less expensive than DDA options.

2. Line 75, The authors promote the use of data-independent acquisition (DIA) without defining what this approach is and how it differs from the more conventional data-dependent acquisition. As a non-mass spectroscopist, I found myself with lots of question concerning DIA, what it is and how it differs from DDA. I think it would really be helpful to expand the description of DIA and its comparison with DDA in the introduction.

As non-mass-spectroscopists ourselves, we understand the reviewer's point. Because the paper is quite long, we were trying to avoid non-essential information. We have now added some information to explain some of the key differences between DDA and DIA. We have also included references for readers who may want to learn more. (lines 77-80)

Minor concerns:

Line 92 typo. I believe the authors meant to say NEKL-2-MLT-2-MLT-4.

Corrected. (line 95)

Line 169. Is exogenous the correct word to use here? It suggests that you are talking about non-worm proteins, but I know you are not.

Corrected. Changed to "Moreover, the detection of biotinylated proteins may be difficult if the bait-TurboID fusion is expressed at low levels..." (line 181).

Line 177 typo (D) should be (C).

Corrected. (line 1122)

Figure 1C: Lucky Charms may sue you for infringement of their trademarked marshmallow treats.

Thank you for picking up on this. The authors accept full responsibility for any resulting lawsuits.

Figure 1D. The NEKL-2::TurboID band is indicated with a green triangle in the figure but the

figure legend states that green triangles indicate mNG::TurboID control. I know this triangle is a shade off the triangle that indicates mNG::TurboID but it's really hard to see the difference. All of the differently colored triangles in panel F are unnecessary. I would either just pick one color for all non-control bait proteins or better yet, only use a triangle to point to bands that are not obvious. For instance I don't need the triangles that point to NEKL-2 -3 and -4 fusion proteins. These are just distracting.

We understand the reviewer's point. We colored the triangles to match the colors used for the proteins in the figures. We have now added "bright green triangles with white outlines" (Fig 1 legend) to indicate the $P_{dpy-7}::mNG::TurboID$ control" and changed triangles in the corresponding figures. Although we would be fine with removing or changing the triangles, we think that they may aid somewhat with clarity.

Line: 316: Conceivably, another factor that could contribute to the counterintuitive upregulation of some proteins in the N2 samples is related to the fusion proteins that are being expressed in the TurboID lines. A partially functional bait protein (one with a level of activity similar to nekl-2(fd81) that may not result in an obvious phenotype) could directly or indirectly affect gene expression leading to lower levels of a subset of proteins in the TurboID samples. The same could be said for fusion proteins with a gain-of-function effect.

This is an interesting idea, and we tested this possibility by looking for consistent overlap between N2-up proteins between biological replates of individual bait proteins. We now include a representative Venn diagram in S3C Fig to highlight this comparison. In summary, although we cannot rule out this possibility, our analysis did not support the widespread occurrence of this effect in our study. We also made certain that our statement regarding N2 up proteins was not too definitive. (lines 285-288)

Fig 3 B-E. I am a little confused how the data in these graphs is normalized. For instance, I would have expected that for NEKL-3 in panel B, that the normalized (log2) intensity value in N2 be set at 0 as it is for NEKL-2. Maybe I just don't have enough information on how these plots were generated.

The difference is that in the N2 sample, NEKL-3 was detected but NEKL-2 was not. The numbers themselves are assigned by the Spectronaut software used to quantify the DIA results but are not meaningful beyond indicating relative amounts (intensity values) of a given protein within an individual biological experiment. We've added some lines to the figure legend to make this clearer. (lines 1165-1169)

Figure 6C legend is not correct.

Corrected. (line 1214)

Line 575: Figure reference should be Fig. S5G. The authors should check to make sure all references to supplemental figures include correct panel information.

Corrected. (line 464) In addition, we have now gone through the manuscript and added panel numbers references where applicable. Note that the addition of a new supplemental file has

shifted the numbering.

Line 576. The authors reference a study by Artan and colleagues and report a weak correlation between their study and that of Artan. They reference figure S4 but it should be Fig S5H.

Apologies and many thanks to the reviewer for catching these errors. (line 464)

Line 652. The authors note that numerous proteins were present at substantially reduced levels in the mNG::TurboID samples and suggest that sticky proteins may have been outcompeted or otherwise excluded from beads incubated with the mNG::TurboID lysates. Why would sticky proteins only be a problem in these samples? The reasoning is not clear to me.

The idea was that in the sample with very high levels of biotinylated proteins (mNG::TurboID), the surface of the beads might become saturated with high-affinity biotinylated proteins. This could prevent or out complete the binding of random proteins that are not biotinylated but nevertheless have some affinity to the beads ("sticky" proteins). We have reworded this section to make this clearer. (lines 546-550)

Line 745: The term "bait overlaps" is a bit vague. Ultimately, I figured out what it meant but it was not immediately obvious.

We have changed this to "overlap between baits" and made this section clearer. (line 624-628)

S7B Fig. Why is actin missing from the eluate?

In S7B we refer to the purified eluate as the "eluate", which may have caused some confusion. In other sections of the manuscript, we refer to the bead-bound proteins as the "purified eluate" (Figs 1 and 5). For the purified eluate a portion of the streptavidin beads are boiled in sample buffer to elute the bound proteins before running a western. Actin would not be expected in these samples because it's (presumably) not biotinylated in our samples and doesn't detectably bind the beads. This result was seen in all relevant westerns in S1 Data. For consistency, however, we've gone through all our files to make sure we consistently use the term "purified eluate" versus "eluate", which is less specific.

Line 873: The authors state the extent of overlap in GO terms between the various experiments and provide percentages. I tried to extract this information from Figure 8C and came up with different values. For instance, in the case of Molecular Function, they state that they observed a 54% overlap between NEKL-2 and NEKL-3 but in the Venn diagram in Figure 8C I see that the NEKL-2 and NEKL-3 experiments had 71 (25+46) GO terms in common. Out of 98 GO terms for NEKL-2 or 104 for NEKL-3 the percentage I got is closer to 72. Am I analyzing this correctly?

Thanks for checking this. We believe our method for calculating the percent overlap is correct. In the case of NEKL-2/NEKL-3 overlap for Molecular Function, there are 131 total unique terms, of which 71 overlap, giving a 54% overlap. In the case of NEKL-2/NEKL-3 overlap for Biological Process, however, we made an error in arithmetic (415 unique, 239 overlap), such that the correct percentage is 58%, which we have corrected in the text.

Reviewer #1 (Significance (Required)):

Overall this is an outstanding study that will be of great interest to those interested in using proximity labeling to identify interactors of their favorite protein. The experiments are well executed and the data presented in a mostly clear manner. I really like this study (particularly because I plan to do a proximity labeling study of my own), but I did come away less than impressed with some of the analysis. This is a data-dense manuscript, and it appears to me that the authors tried to cover so much ground that in some cases very little insight was provided. For instance, the authors promote the use of data independent acquisition (DIA) as compared to the more commonly used data dependent acquisition (DDA). However the authors do not provide any analysis to indicate one approach is better than the other. Likewise the combined use of auxin-induced degradation and proximity labeling is explored but there is very little to take away from these experiments. Despite these issues, I am very enthusiastic about the study as a whole.

Reviewer #2 (Evidence, reproducibility and clarity (Required)):

This study expanded the use of data-independent acquisition-mass spectrometry (DIA-MS) in TurboID proximity-labeling proteomics to identify novel interactors of NEKL-2, NEKL-3, MLT-2, MLT-3, and MLT-4 complexes in *C. elegans*. The authors described several useful metrics to evaluate the quality of TurboID experiments, such as using the percentage of upregulated genes, the percentage of proteins present only in bait-TurboID experiments as compared to N2 controls, and the percentage of endogenously biotinylated carboxylases as internal controls. Further, the authors introduced methodological variability across 23 TurboID experiments and evaluated any improvement to the resulting data, such as N-terminally tagging bait proteins with TurboID, depleting endogenous carboxylases, and auxin-inducible degradation of known complex members. Finally, this study identified the kinase folding chaperone CDC-37 and the WASH complex component DDL-2 as novel interactors with the NEKL-MLT complexes through an RNAi-based enhancer approach following their identification by TurboID.

Major comments:

The key conclusions are convincing, and the work is rigorous. The work provides a clear roadmap to reproducing the data. The experiments are adequately replicated, and statistical analysis is adequate. We only have minor comments.

Minor comments:

- In the western blot in Fig 1 why does the mNG::Turbo have two bands?

Thank you for pointing this out. To our knowledge this is a breakdown product that was especially prevalent in replicate 3 (also see S1 Data), which we chose to show because all the NEKL-MLTs were clearly visible in this western. The expected size of the mNeonGreen::TurboID (including linker and tags) is ~68 kDa and our blots are roughly consistent those of Artan et al., (2001). This lower band was not evident in Exp 8. We have now included a statement in the figure legend to indicate that the upper band is the full-length protein whereas the lower band is

likely to be a breakdown product (lines 1141-1142).

•Fig 2B is difficult to parse as a reader. Columns labeled "Upreg," "Downreg," "TurboID only," "N2 only," "Filter-1," "Filter-2," and "Epi %" could be moved to Supplemental. Fold change vs N2 could be represented as a bar chart, allowing for trends between fold change and the metrics Upreg %, Turbo %, and Carboxylase % to be seen more clearly. Further, rows headed "Carboxylase depletion," "DDA," and "Auxin treated" could be presented as separate panels to better match the distinct points made in the text.

After serious consideration we have made several changes including the addition of S2 Fig, which may provide readers with a better visual representation of the bait and prey fold changes observed in all our experiments. However, we feel that the detailed data embedded in Fig 2 is the most concise and accurate means by which to convey our full results and is key to our methodological conclusions. As such we did not want to relegate this information to a supplemental table. We note that this figure was not found to be problematic by other reviewers, although we do understand the points made by this reviewer.

•Line 179: *in vivo* should be italicized

Because journals differ in their stylistic practices, we are currently waiting before doing our final formatting. We did keep our use of Latin phrases consistently non-italicized in the draft.

•Lines 215-217: The comparison between Western blot expression levels and prior fluorescent reporter levels is unclear. Could be reformatted to make it clearer that relative expression of the different NEKL-MLTs in this study is consistent with prior data.

We reformatted this sentence to improve clarity. (lines 205-207)

•Lines 267-268: The final line of the passage is unclear and can be removed.

This sentence has been removed.

•Lines 311-313: This study is able to use the recovery of bait and known interactor proteins as internal controls to determine the quality of each experiment, but this may not always be the case for other users' experiments. The authors should comment on how Upreg %, a value influenced by many factors, can actually be used as a quality check when a bait protein has no known interactors.

We have added language to highlight this point. (lines 344-348)

•Line 702: There is a [new REF] that should be removed

As described above, we have now included this finding on *bcc-1* as part of this manuscript (Fig 9C).

•The approach used mixed stage animals, but some genes oscillate or are transiently expressed. Please discuss cost-benefit of mixed stage vs syncing.

This is an important point. We have added a discussion on the benefits and drawbacks of using mixed stages to the discussion. (lines 901-911)

- Authors were working on hypodermally expressed proteins. It would be valuable to discuss what tissues are amenable to TurboID. Ie are the cases where there are few cells (anchor cell, glial sockets, etc) that it will be extremely challenging to perform this technique
We agree that certain tissues/proteins will not be amenable to proximity labeling. We believe that we have addressed this point together with the above comment throughout the manuscript and now on lines 936-940.

- Authors mention approaches such as nanobodies, split Turbo. Based on their experiences it would be valuable to add Discussion on strengths and weaknesses of these approaches to guide folks considering TurboID and DIA-MS experiments in C. elegans
Because we have not tested these methods, we feel that we cannot provide a great deal of insight into these alternate approaches. We mention and reference these methods in the introduction so that readers are aware of them.

Reviewer #2 (Significance (Required)):

- Advance in technique: This study expands the use cases of data-independent acquisition MS method (DIA-MS) in C. elegans, which fragments all ions independent of the initial MS1 data. The benefits of this approach include better reproducibility across technical replicates and better recovery of low abundance peptides, which are critical for advancing our ability to capture weak and transient interactions.

- The use of DIA-MS in this study has improved our understanding of the partners of these NEKL-MLTs in membrane trafficking, molting, and cell adhesion within the epidermis.

- In many papers, TurboID seems very trivial but this paper clearly highlights the limitations and will be an invaluable resource for labs that want to get proximity labeling established in their labs.

Reviewer #3 (Evidence, reproducibility and clarity (Required)):

Summary:

Fay and colleagues perform a series of proximity labeling experiments in C. elegans followed by thorough and rational analysis of the resulting biotinylated proteins identified by LC-MS/MS. The overall goals of the study are to evaluate different techniques and provide practical guidance on how to achieve success. The major takeaways are that integration of data-independent acquisition (DIA) along with comparison of endogenously tagged TurboID alleles to soluble TurboID expressed in the same tissue results in improved detection of bona-fide interactors and reduced numbers of false-positives.

Major comments:

Overall the claims are solid and conclusions supported. The data and methods are substantial to enable reproducibility in other labs. The experiments have been repeated multiple times with particular attention to statistical analysis. I have no major concerns with the manuscript and focus primarily on improving the accessibility of this important contribution to the scientific community. As such, I suggest that the authors:

1) Provide more explanation of and rationale for using DIA. This is not yet a standard technique and most basic biomedical scientists will be unaware of the jargon. As I expect many labs in the *C. elegans* community and beyond will be interested in the guidance provided in this manuscript, the introduction offers a great opportunity to bring the reader up to speed, as opposed to sending them to the complicated proteomics analysis literature.

We have added some additional context (lines 77-80) as well as new references. We note that getting into the technical differences between DIA and DDA, beyond what we briefly mention, would take a substantial amount of space, may not be of interest to many readers, and can be found through standard internet and (sigh) AI-based searches.

2) Provide a better overview of the various protocols tested (Experiments 1-8). Maybe at the beginning of the results, and maybe with an accompanying schematic. As currently written, it is difficult to figure out details regarding how the experiments vary and why.

We have now added a short paragraph to better inform the reader at the front end regarding the major experiments. (lines 139-146).

3) As to be expected, expression of TurboID tags at endogenous levels via low abundance proteins in a complex multicellular system results in somewhat weak signals that flirt with the limit of detection. Perhaps by combining tagged alleles within the same complex (NEKL-3/MLT-3 or NEKL-2/MLT-2/MLT-4) the signals could be boosted? Tandem tags, either on one end or multiple ends of proteins might help as well. As the authors point out, a benefit of tagging the two NEKL-MLT complexes is that there are strong loss-of-function phenotypes (lethal molting defects) to help evaluate whether a tagging strategy results in a non-functional complex. THESE EXPERIMENTS ARE OPTIONAL and might simply be discussed at the authors discretion.

These are interesting ideas that we have now incorporated into our discussion. (lines 936-940)

Minor Comments:

1) Figure 3A is cropped on the right.

Thank you for catching this. Corrected.

2) Better define [new REF] on line 702.

We have added new results (Fig 9C), obviating the need for this reference.

Referee cross-comments

Full Revision

Overall, I am in agreement with, and supportive of, the other reviewers' comments.

Reviewer #3 (Significance (Required)):

Significance:

Proximity labeling is often proposed as a technique to determine interaction networks of proteins in vivo, but in practice it remains challenging for most labs to execute a successful experiment, especially within the context of multicellular model organisms. **Fay and colleagues provide a much needed roadmap for how to best approach proximity labeling experiments in C. elegans that will likely apply to other model systems.**

They establish a rigorous approach by choosing to endogenously tag components of two essential NEKL-MLT complexes required for C. elegans molting. These complexes are relatively low abundance as they are only expressed in a single cell type, the hyp7 epidermal syncytium. In addition, as inactivation of any member of the complexes results in molting defects, they have a powerful selection for functional tags. Thus, they have set a high bar for themselves in order to discern whether a given variation on the experimental approach results in improved detection of interactors and fewer false positives.

Potential areas for improvement include lowering the expression level of the skin-specific soluble TurboID used to determine non-specific biotinylation events. This control results in much higher levels of biotinylation compared to the TurboID-tagged NEKL-MLT alleles and likely affects their analysis, which they openly admit. In addition, to reduce the high level of background biotinylation signals generated by endogenous carboxylases, they adopt a depletion strategy pioneered by other researchers but this does not offer major improvements in detection of specific signals. The source of these conflicting results remains to be determined. It is also curious that auxin-inducible degradation of components of the NEKL-MLT complexes did not robustly alter the resulting biotinylation capacity of other members. This approach should be evaluated in subsequent studies. Finally, as mentioned in Major Comment #3 (above), it would be interesting to see if combining TurboID tags within the same complex might improve signal-to-background ratios.

This manuscript represents a methodological advance that will likely become an oft-cited reference for members of the C. elegans community and a springboard for other basic biomedical scientists wanting to adapt rigorous proximity labeling techniques to their system. I am a cell biologist that uses a variety of genetic, molecular and biochemical approaches, mostly centered around C. elegans. I have used LC/MS-MS in our studies but have relatively little expertise in evaluating all aspects of proteomic pipelines.

Reviewer #4 (Evidence, reproducibility and clarity (Required)):

Fay et al. describe an extensive proximity labeling BioID study in C. elegans with TurboID and

DIA-LCMS analysis. They chose the NEKL-2/3 kinases and their known interactors MLT-2/3/4 as TurboID-fused bait proteins (C- and partially N-terminal fusions encoded from CRISPR-mediated genome edited genes). With eight biological replicates (and three to four technical replicates each) and with the unmodified wildtype or mNeonGreen-TurboID expressing worms as controls, a comprehensive dataset was generated. Although starting from quite different abundances of the bait-fusions within the cell lysates all bait proteins and known complex-binding partners were convincingly enriched with capturing streptavidin beads after only one hour of incubation with the lysate. This confirms the general applicability of TurboID-BioID approach in *C. elegans*. The BioID method typically gives rise to large proteomics datasets (up to more than thousand proteins identified after biotin capture) with several tens to hundreds enriched proteins (against negative control strains) as potential proteins that localize proximal to the bait-TurboID protein. However, substantial variations of candidates between biological replicates are frequently observed in BioID experiments. The authors scrutinized their dataset towards indicative metrics, filters and cutoffs in order to separate high-confidence from low-confidence candidates. With the workflow applied the authors melt down the number of candidates to 15 proteins that were grouped in four functional groups reasonably associated to NEKL-MLT function.

Successful BioID experiments depend on reliable enrichment quantification with mass spectrometry using control cell lines that require a carefully bait-tailored design. Those must adequately express TurboID controls matching the abundance of the bait-TurboID fusion protein and its biotinylation activity. After affinity capture, sample preparation and LCMS data acquisition there is no silver bullet towards the identification true bait neighbors. Fay et al. elaborately describe their considerations and workflow towards high-confidence candidates. The workflow considered (i) data analysis with Volcano plots to account for statistical reproducibility of biological replicates against negative controls, (ii) fraction of proteins only detected in the positive or negative controls thus evading the fold-enrichment quantification approach, (iii) evaluation of variations in carboxylase enrichment as a measure for variations in the general biotin capture quality between experiments, (iv) an assessment of technical reproducibility with scatter plots and Venn diagrams, (v) exclusion of potentially false positives, e.g. promiscuously biotinylated non-proximal proteins, through comparisons with control worms expressing a non-localized mNeonGreen-TurboID fusion protein, (vi) batch effects, (vii) the impact of endogenous biotinylated carboxylases through depletion, (viii) gene ontology analysis of enriched proteins, (ix) weighing data according to the quality of individual experiments according to the aforementioned metrics, and finally (x) genetic interaction studies to functionally associate high-confidence candidates with the bait.

Major comments:

Fay et al. present a solid, clear and comprehensive BioID-based proteomics study that takes into account and discusses decisive aspects for the (re)production and analysis of high-quality TurboID-based mass spectrometry data. Claims and conclusions are generally well and sufficiently supported by the presented data and illustrated with figures (throughout the text as

well as with plenty of supplementary data). However, although the authors claim to seek for substrates of the kinase complex they drew no further attention to the phosphorylation status of the captured proteins. Haven't the MS data been analyzed in this respect? Information regarding this issue would enhance the manuscript. Data generation and method description appear reproducible for readers. Also, the statistical analyses appear adequate. The authors should also consider to deposit their MS raw and analysis data in a public repository (e.g. PRIDE) for future reviewing processes and as reference data for readers and followers. Our raw MS data have been deposited by the Arkansas Proteomics Facility. I have followed up to ensure that they are publicly available.

Minor comments:

The authors should combine supplementary data files to reduce the number of single files readers have to deal with.

We have combined these files as suggested.

The authors should avoid the term "upregulation" or "increased biotinylation" when capture enrichment is meant.

We agree with reviewer's point. We now use the terms enriched versus reduced or up versus down, depending on the context, and clearly define these terms. These changes have been incorporated throughout the manuscript.

Reviewer #4 (Significance (Required)):

The manuscript presents a robust BioID proteomics screening for co-localizing proteins of NEKL-2/3 kinases and their known interactors MLT-2/3/4. The ongoing validation of their functional interactions and whether the protein candidates reflect phosphorylation substrates or else remains elusive and is announced for upcoming manuscripts. The knowledge gain in terms of molecular mechanisms with NEKL-2/3 MLT-2/3/4 involvement in *C. elegans* is therefore limited to a table of - promising - interacting candidates that have to be studied further. Information about the phosphorylation status of the captured proteins from the MS data are not given. However, knowing the protein candidates will be of interest for groups working with these complexes (or the identified potentially interacting proteins) either in *C. elegans* or any other organism. Also, in-depth proteomics screenings with novel approaches such as BioID have to be established for individual organisms. For *C. elegans* there is only one prior BioID publication (Holzer et al. 2022). Many of the aspects discussed here have also been addressed earlier for BioIDs in other organisms and are not principally new. However, the presented study can be of conceptual interest for labs delving into or entangled with the BioID method in *C. elegans* or other organisms. The study addresses especially proteomics groups working on protein-protein interactions using proximity labeling/MS approaches. Basic consideration and thoughts for the experimental design and MS data analysis are given in detail and can serve as another guideline for future studies.

Dear Dr Fay,

Thank you for submitting your revised manuscript (EMBOJ-2025-122132-T) to The EMBO Journal, as well for your patience with our response. Your amended study was sent back to the referees for their scientific reassessment, and we have received detailed re-reports from two of them, which I enclose below. Please note that while referees #1 and #4 were unfortunately not able to reassess your work, we have now editorially evaluated your response to these experts and found the issues raised to be addressed satisfactorily. As you will see, the other experts state that the work has been substantially enhanced by the revisions and they are now broadly in favour of publication, pending minor revision.

Thus, we are pleased to inform you that your manuscript has been accepted in principle for publication in The EMBO Journal as a Methods Resource article.

Please carefully consider the remaining minor points raised by the referees, revising data presentation and annotation.

Also, we now need you to take care of a number of minor issues related to formatting and data annotation, i.p. with respect to a methods article, which I will share shortly in a separate message, together with additional changes and requests by our production team including for Source Data provision.

Please contact me at any time if you have additional questions related.

As you might have noted from our webpage, every paper at the EMBO Journal now includes a 'Synopsis', displayed on the html and freely accessible to all readers. The synopsis includes a 'model' figure as well as 2-5 one-short-sentence bullet points that summarize the article. I would appreciate if you could provide this figure and the bullet points.

Thank you for giving us the chance to consider your manuscript for The EMBO Journal. I look forward to your final revision.

Again, please contact me at any time if you need any help or have further questions.

Best regards,

Daniel Klimmeck

Revision to The EMBO Journal should be submitted online within 90 days, unless an extension has been requested and approved by the editor; please click on the link below to submit the revision online before 24th Dec 2025:

Referee #2:

The authors have improved an already strong paper. In particular the addition of the bcc-1 data in 9C is very nice. The only minor comment is that in the author list the last author is "Eden" but on the lab's papers and websites it seems like it should be "Edeen"?

Referee #3:

The Authors have responded to all Reviewers concerns, except potentially the question regarding whether the phosphorylation status of the capture proteins was analyzed (see

Reviewer #4 Major Comments).

I expect the manuscript will find significant traction among the readers of The EMBO Journal and exert a positive impact upon multiple areas of basic and biomedical research.

Rev_Com_number: RC-2025-03083

New_manu_number: EMBOJ-2025-122132-T

Corr_author: Fay

Title: Proximity Labeling of NIMA Kinase Complex Components in *C. elegans*

Dear Dr Fay,

Further to below, please find encoded the list of mentioned formatting and data presentation changes required for the final revised version of your manuscript and related files.

Please let us know any time if you have questions related, or need additional input.

We look forward to the submission of your final, revised manuscript.

Best regards,

Daniel Klimmeck

>> Please add up to five keywords to your study.

>> Please provide the main manuscript text as .docx file.

>> Provide a completed Author Checklist.

>> Add a 'Disclosure and Competing Interests Statement' section and place after Acknowledgements.

>> Section order should be corrected as follows: title page with complete author information, abstract, keywords, introduction, results, discussion, methods, data availability section, acknowledgements, disclosure and competing interests statement, references, main figure legends, tables, expanded figure legends.

>> Funding: please integrate the funding information into the Acknowledgements.

>> Please delete the list of supplementary information from the manuscript text file.

>> References: adjust the reference format to EMBO Journal format, 10 authors et al, and place References after the Disclosure and competing interests statement, before figure legends. Remove the DOIs and PubMed IDs (except for unpublished works). In the text of the manuscript, a reference should be cited by author and year of publication; no more than two authors may be cited per reference; 'et al' should be used if there are more than two authors (i.e. Smith & Jones, 2003; Smith et al, 2000).

>> Please recheck references for the bioRxiv entries: Zhe Yang et al. (2024), Gömörýova et al. (2025), Shi et al. (2025), Reimann et al. (2025), Grismer et al. (2024), Wei et al. (2025), Mahuzier et al. (2024), Evans et al. (2025), Evans et al. (2025), Kim et al. (2024), and update the citation if in the meantime published as regular article.

>> Add a Reagents and Tools table to the Methods section, as a separate file using the existing template in the Guide For Authors, listing key reagents, experimental models, software and relevant equipment.

Overall, we classify your study as a 'Methods' article. Accordingly, Please describe the new original methods developed using a step-by-step protocol format with bullet points and notes on "tricky steps" in the procedure. We strongly recommend this 'protocol' format for novel methodologies and procedures that will be relevant for readers to use in their own studies. Using the protocol format is optional unless otherwise specified by the editor. The protocol format does not need to be applied throughout the Methods section.

Please compare also our Guide-to-Authors instructions:

<https://www.embopress.org/page/journal/14602075/authorguide#structuredmethods>

>> Appendix file with ToC: integrate the three PDF files with S1 Data (western blots), S1 Information (sequences) into an

Appendix .pdf file. Rename the suppl. tables "Appendix Table S1" etc., and the suppl. figures "Appendix Figure S1". Please add a table of contents on the first page, including page numbers.

>> Move all Suppl Methods to the main manuscript.

>> Figures in separate files: please rename the supplementary figures "Figure EV1" - "Figure EV9" and upload them as individual, high resolution figure files.

>> Dataset EV legends: Please rename the files S1 - S14 "Dataset EV1" - "Dataset EV14", and remove S15 File and distribute the information to the corresponding files, added in a separate tab/worksheet. Please add the S2 Data to the corresponding source data folders for Fig 4 and Fig 9.

>> Please provide source data for the study as to the separate request e-mail. Source data should be uploaded as one (zipped) file per figure.

>> Data availability section: please add a Data availability section to the manuscript text, stating ' No data amenable to large data repository deposition were generated in this study.'.

>> Consider additional changes and comments from our production team as indicated below:

- DAS

Please note that the data availability statement is not provided in the manuscript.

- Figure legends:

1. Please note that the exact p values are not provided in the legends of figures 3B-E; 5B-D; 6C; 9B
2. Please indicate the statistical test used for data analysis in the legends of figures 3B-E; 4B-F; 5B-D; 6B, 9B
3. Please note that information related to n is missing in the legend of figure 6B
4. Please note that the error bars are not defined in the legends of figures 9B, C

Referee #2:

The authors have improved an already strong paper. In particular the addition of the bcc-1 data in 9C is very nice. The only minor comment is that in the author list the last author is "Eden" but on the lab's papers and websites it seems like it should be "Edeen"?

Corrected.

Referee #3:

The Authors have responded to all Reviewers concerns, except potentially the question regarding whether the phosphorylation status of the capture proteins was analyzed (see Reviewer #4 Major Comments). I expect the manuscript will find significant traction among the readers of The EMBO Journal and exert a positive impact upon multiple areas of basic and biomedical research.

Apologies for missing this point. We have now added a few sentences to the final section of our manuscript as follows:

“Additionally, our study was not designed or optimized for the detection of site-specific target phosphorylation events, which typically requires a phospho-enrichment step prior to MS (Qiu *et al*, 2020). Nevertheless, several recent reports indicate that DIA is applicable and effective for use in phospho-proteomic analyses (Bekker-Jensen *et al*, 2020; Di *et al*, 2023; Pham *et al*, 2024; Skowronek *et al*, 2022).”

Dear Dr Fay,

Thank you for submitting the revised version of your manuscript. I have now evaluated your amended manuscript and concluded that the remaining minor concerns have been sufficiently addressed.

I am thus pleased to inform you that your manuscript has been accepted for publication in the EMBO Journal.

Kind regards,

Daniel Klimmeck

Daniel Klimmeck, PhD
Senior Editor
The EMBO Journal
EMBO
Postfach 1022-40
Meyerhofstrasse 1
D-69117 Heidelberg
contact@embojournal.org

Please note that it is The EMBO Journal policy for the transcript of the editorial process (containing referee reports and your response letters) to be published as an online supplement to each paper. If you should prefer removal of any referee-only figures included in the point-by-point response(s), e.g. because they may still be used for future publication or because they have been reproduced from published work by others, please do let us know immediately via response email.

More information is available here: https://www.embopress.org/transparent-process#Review_Process

Rev_Com_number: RC-2025-03083

New_manu_number: EMBOJ-2025-122132R

Corr_author: Fay

Title: Proximity Labeling of NIMA Kinase Complex Components in *C. elegans*